# Task Weighting in Meta-learning with Trajectory Optimisation

**Cuong Nguyen**                                                      *cuong.nguyen@adelaide.edu.au*
*Australian Institute for Machine Learning*
*The University of Adelaide*

**Thanh-Toan Do**                                                      *toan.do@monash.edu*
*Department of Data Science and AI*
*Monash University*

**Gustavo Carneiro**                                                   *g.carneiro@surrey.ac.uk*
*Centre for Vision, Speech and Signal Processing*
*University of Surrey*

**Reviewed on OpenReview:** *https://openreview.net/forum?id=SSkTBUyJip*

## Abstract

Developing meta-learning algorithms that are un-biased toward a subset of training tasks often requires hand-designed criteria to weight tasks, potentially resulting in sub-optimal solutions. In this paper, we introduce a new principled and fully-automated task-weighting algorithm for meta-learning methods. Specifically, we frame the task-weighting problem as a trajectory optimization problem, where the weights of tasks within a mini-batch are treated as an action, and the meta-parameter of interest is viewed as the system state. Such a modelling allows us to employ the iterative linear quadratic regulator to determine the optimal task weights. We theoretically show that the proposed algorithm converges to an $\epsilon_0$-stationary point, and empirically demonstrate that the proposed approach out-performs common hand-engineering weighting methods on two few-shot learning benchmarks.

## 1 Introduction

Meta-learning has been studied from the early 1990s (Schmidhuber, 1987; Naik & Mammone, 1992; Thrun & Pratt, 1998) and recently gained a renewed interest with the use of deep learning methods that achieves remarkable state-of-art results in several few-shot learning benchmarks (Vinyals et al., 2016; Finn et al., 2017; Snell et al., 2017; Nichol et al., 2018; Ravi & Beatson, 2019; Allen et al., 2019; Khodak et al., 2019; Baik et al., 2020; Flennerhag et al., 2020). However, the majority of existing meta-learning algorithms simply minimise the average loss evaluated on validation subsets of training tasks, implicitly assuming that testing tasks come from the same distribution (over family of tasks) as training tasks (analogous to class-imbalance in single-task learning). This assumption is hardly true in practice, and potentially biases the trained meta-learning models toward tasks observed more frequently during training, and consequently, resulting in a large variation of performance when evaluating on different subsets of testing tasks as shown in (Dhillon et al., 2019, Figure 1) and (Nguyen et al., 2021, Figure 1).

One way to address such issue is to exploit the diversity of training tasks, so that the trained meta-learning models can generalise to a wider range of testing tasks. In fact, various studies in task relatedness or task similarity have shown that learning from certain tasks may facilitate the generalisation of meta-learning models (Thrun & O'Sullivan, 1996; Zamir et al., 2018; Achille et al., 2019; Nguyen et al., 2021). This suggests the design of a re-weighting mechanism to diversify the contribution of each training task when training a meta-learning model of interest. Existing re-weighting methods mostly rely on either hand-crafted criteria to determine those weights (Collins et al., 2020), or additional validation tasks to learn the re-weighting factors

of interest (Xu et al., 2021). Such ad-hoc development of re-weighting mechanisms motivates us to design a more principled approach to re-weight tasks for meta-learning. We note that there are also studies learning to balance the contribution of each training task, e.g., learning to balance (L2B) (Lee et al., 2020). However, L2B focuses on the task-adaptation step (also known as inner loop), while our interest is to explicitly weight the contribution of each task at the meta-learning step (also known as outer-loop).

In this paper, we present a new principled and fully-automated task-weighting algorithm, called **t**rajectory **o**ptimisation based task **w**eighting for meta-learning (TOW). In particular, we regard the iterative update of the meta-parameter of interest as a trajectory, where the meta-parameter is the state and the task weighting vector is the corresponding action. This formulation casts the task-weighting problem in meta-learning as a trajectory optimisation problem, with the gradient-based update for the meta-parameter as the state transition dynamics and the weighted validation loss as the cost function. We then employ the techniques of trajectory optimisation to determine the optimal action (i.e., weighting vector) w.r.t. the cost (i.e., weighted validation loss) to train the meta-parameter. We note that TOW is not a meta-learning method, but a task weighting framework that can be integrated into existing meta-learning algorithms to circumvent the problematic assumption about the even distribution of training tasks. Our contributions can be summarised as follows:

- We propose to cast the task-weighting problem in meta-learning to a finite-horizon discrete-time trajectory optimisation with its state denoted by the meta-parameter and action denoted by the re-weight factors of tasks, and solve such problem using the iterative linear quadratic regulator.

- We prove that under the conditions of boundedness and smoothness of the loss function used, TOW converges to a particular $\epsilon_0$-stationary point.

We also demonstrate TOW's functionality by incorporating it into two common meta-learning algorithms, namely MAML (Finn et al., 2017) and Prototypical Networks (Snell et al., 2017), and showing that TOW enables meta-learning methods to converge with fewer number of iterations and achieves higher prediction accuracy than some common task re-weighting mechanisms in the literature.

## 2 Background

### 2.1 Trajectory optimisation

Given a continuous $D$-dimensional vector $\mathbf{x}$ as a state and an $M$-dimensional vector $\mathbf{u}$ as an action, the objective of a trajectory optimisation is to find an optimal sequence of actions $\{\mathbf{u}_t^*\}_{t=1}^T$ as follows:

$$\{\mathbf{u}_t^*\}_{t=1}^T = \arg \min_{\{\mathbf{u}_t\}_{t=1}^T} \sum_{t=1}^T c(\mathbf{x}_t, \mathbf{u}_t) \quad \text{s.t. } \mathbf{x}_{t+1} = f(\mathbf{x}_t, \mathbf{u}_t), \tag{1}$$

where: $c(\mathbf{x}_t, \mathbf{u}_t)$ and $f(\mathbf{x}_t, \mathbf{u}_t)$ are the cost function and the state-transition dynamics evaluated at time step $t$, respectively. These functions are assumed to be twice differentiable. In addition, the initial state $\mathbf{x}_1$ is assumed to be given, and *trajectory optimisation* means finding the optimal sequence of actions $\{\mathbf{u}_t^*\}_{t=1}^T$ for a particular $\mathbf{x}_1$, not for all possible initial states.

In trajectory optimisation, the finite-horizon discrete-time problem shown in (1) can be solved approximately by iterative methods, such as differential dynamic programming (DDP) (Jacobson & Mayne, 1970) or iterative linear quadratic regulator (iLQR) (Todorov & Li, 2005; Tassa et al., 2012). These methods rely on a local approximation of the state-transition dynamics and cost function using Taylor series about a nominal trajectory $\{\hat{\mathbf{x}}_t, \hat{\mathbf{u}}_t\}_{t=1}^T$. In DDP, both the state-transition dynamics and cost function are approximated to the second order of their Taylor series, while in iLQR – a "simplified" version of DDP, the state-transition dynamics is approximated up to the first order. In a loose sense, DDP is analogy to the Newton's method, while iLQR is analogous to a quasi-Newton's method. In this paper, we employ iLQR to solve for the task weights of interest due to its efficiency in terms of computation compared to DDP.

The main idea of iLQR is to cast a general non-linear trajectory optimisation problem shown in (1) to a linear quadratic problem (LQP) by linearising the state-transition dynamics and quadraticising the cost function about a nominal trajectory $\{(\hat{\mathbf{x}}_t, \hat{\mathbf{u}}_t)\}_{t=1}^{T}$ via Taylor's series. In particular:

$$f(\mathbf{x}_t, \mathbf{u}_t) \approx f(\hat{\mathbf{x}}_t, \hat{\mathbf{u}}_t) + \underbrace{\boldsymbol{\nabla}_{\mathbf{x}} f(\hat{\mathbf{x}}_t, \hat{\mathbf{u}}_t)}_{\mathbf{F}_{\mathbf{x}_t}}(\mathbf{x}_t - \hat{\mathbf{x}}_t) + \underbrace{\boldsymbol{\nabla}_{\mathbf{u}} f(\hat{\mathbf{x}}_t, \hat{\mathbf{u}}_t)}_{\mathbf{F}_{\mathbf{u}_t}}(\mathbf{u}_t - \hat{\mathbf{u}}_t), \tag{2}$$

where $\mathbf{F}_{\mathbf{x}_t}$ and $\mathbf{F}_{\mathbf{u}_t}$ denote the first gradients of the transition dynamics $f(\mathbf{x}_t, \mathbf{u}_t)$ w.r.t. $\mathbf{x}_t$ and $\mathbf{u}_t$, respectively. Similarly, one can obtain the Taylor's series for the cost function $c(\mathbf{x}_t, \mathbf{u}_t)$ with $\mathbf{c}_{\mathbf{x}_t}$ and $\mathbf{c}_{\mathbf{u}_t}$ as the first gradients of the cost function, and $\mathbf{C}_{\mathbf{x}_t, \mathbf{x}_t}, \mathbf{C}_{\mathbf{x}_t, \mathbf{u}_t}, \mathbf{C}_{\mathbf{u}_t, \mathbf{x}_t}$ and $\mathbf{C}_{\mathbf{u}_t, \mathbf{u}_t}$ as the second gradients w.r.t. the variables specified in their corresponding subscripts (see Appendix A for more details).

The approximate LQP can then be solved exactly by the linear quadratic regulator (LQR) (Anderson & Moore, 2007). Subsequently, the newly obtained trajectory is used as the nominal trajectory for the next iteration. This process is repeated until the cost function is converged. The detailed derivation of iLQR can be found in Appendix B. Further details of iLQR can be referred to (Todorov & Li, 2005; Tassa et al., 2012). To our best knowledge, there are no previous works that provide a proof on the convergence of iLQR. Therefore, for a complete analysis, we provide the proof of convergence for iLQR adopted from DDP (Yakowitz & Rutherford, 1984) in Appendix C.

## 2.2 Meta-learning

We follow the *task environment* (Baxter, 2000), where tasks are i.i.d. sampled from an unknown distribution $p(\mathcal{T})$ over a family of tasks, to formulate the meta-learning problem. Each task $\mathcal{T}_i$ is associated with two data subsets: training (or support) subset $\mathcal{S}_i^{(s)} = \{(\mathbf{s}_{ij}^{(s)}, y_{ij}^{(s)})\}_{j=1}^{m_i^{(s)}}$, where $\mathbf{s}_{ij}^{(s)}$ denotes a training input and $y_{ij}^{(s)}$ denotes the corresponding training label, $i \in \{1, \ldots, M\}$, and validation (or query) subset $\mathcal{S}_i^{(q)}$ which is similarly defined. For training tasks $\{\mathcal{T}_i\}_{i=1}^{M}$, both data subsets have labels, while for testing tasks $\mathcal{T}_{M+1}$, only the data in $\mathcal{S}_{M+1}^{(s)}$ is labelled. The aim is to learn a $D$-dimensional meta-parameter $\mathbf{x}$ shared across all tasks, so that $\mathbf{x}$ can be efficiently fine-tuned on $\mathcal{S}_i^{(s)}$ to produce a task-specific parameter for predicting the unlabelled data in $\mathcal{S}_i^{(q)}$. One of the simplest forms of meta-learning is analogous to an extension of hyper-parameter optimisation in single-task learning, where the shared meta-parameter $\mathbf{x}$ is learnt from many tasks. The objective of meta-learning can generally be expressed as:

$$\mathbf{x}^* = \arg\min_{\mathbf{x}} \mathbf{u}^{\top} \mathsf{L}_i(\mathbf{x}) \quad \text{with}: \mathsf{L}_i(\mathbf{x}) = \frac{1}{m_i^{(q)}} \sum_{j=1}^{m_i^{(q)}} \ell\left(\mathbf{s}_{ij}^{(q)}, y_{ij}^{(q)}; \phi_i(\mathbf{x})\right)$$

$$\text{s.t.}: \ \phi_i^*(\mathbf{x}) \in \arg\min_{\phi_i} \frac{1}{m_i^{(s)}} \sum_{k=1}^{m_i^{(s)}} \left[\ell\left(\mathbf{s}_{ik}^{(s)}, y_{ik}^{(s)}; \phi_i(\mathbf{x})\right)\right], \forall i \in \{1, \ldots, M\}$$

$$\mathbf{u} \in \mathcal{U}, \tag{3}$$

where $\ell(.)$ is the loss function that is non-negative and twice differentiable, $\mathbf{u}$ is an $M$-dimensional vector that re-weights the influence of $M$ training tasks, $\mathcal{U}$ is the set of feasible weights defined by some weighting criterion, and $\phi(\mathbf{x})$ is the parameter fine-tuned on task $\mathcal{T}_i$ (also known as task-specific parameter).

The task adaptation in the lower-level optimisation of (3) varies depending on the type of meta-learning algorithm used. For example, it can be a gradient descent update with $\mathbf{x}$ being the initialisation of the neural network of interest (Finn et al., 2017). In metric-based meta-learning (Snell et al., 2017), that task adaptation step is slightly different where the class prototypes of training data are embedded into a latent space by the meta-model, and the validation loss is based on the distance between the class prototypes to each data-point in $\mathcal{S}_i^{(q)}$. There are also other extensions of (3) using probabilistic approaches (Yoon et al., 2018; Ravi & Beatson, 2019; Nguyen et al., 2020). Nevertheless, our approach proposed in Section 3 can generally be integrated into any of these meta-learning algorithms. In addition, our attention is on weighting tasks, corresponding to the upper-level in equation (3). As a result, we exclude the discussion of the lower-level optimisation for $\phi^*(x)$ in the rest of this paper to simplify the analysis.

**Remark 1.** *The task-weighting problem in* (3) *is carried out at the meta level (also known as "outer-loop"). It is, therefore, different from some recent meta-learning methods (Khodak et al., 2019; Baik et al., 2020; Flennerhag et al., 2020; Lee et al., 2020) that design different learning strategies for $\phi_i(\mathbf{x})$ at the task adaptation step (also known as "inner-loop") to estimate the meta-parameters with a uniform weighting $\mathcal{U} = \{1/M\}^M$.*

For the feasible weighting set $\mathcal{U}$, the most widely-used weighting criterion is **uniform**: $\mathcal{U} = \{1/M\}^M$, which resembles the conventional meta-learning. Another popular criterion is to select **difficult** tasks – tasks that have the largest validation losses – for training to optimise the performance on the worst-case scenarios (Collins et al., 2020). However, such difficult tasks may not always be preferred when outliers and noise are present. That leads to another weighting approach which favours the **most familiar** data-points in single-task learning (Kumar et al., 2010; Bengio et al., 2009; Wang et al., 2017) – often referred as *curriculum learning*. The two latter task-weighting approaches can be considered as the "exploration" and "exploitation" strategies used in reinforcement learning (RL), respectively. Similar to the exploration and exploitation dilemma in RL, we hypothesise that the optimality for task weighting is formed by a balance between these two approaches. In the following section, we propose a principled approach to automate re-weighting tasks through an optimisation on a sequence of many mini-batches rather than relying on manually-designed criteria as the previous studies.

## 3 Methodology

### 3.1 Task-weighting as a trajectory optimisation

The upper-level optimisation in (3) can also be written as: $\mathbf{x}^* = f(\mathbf{x}, \mathbf{u})$, where $f$ is a function denoting an optimiser, e.g., stochastic gradient descent (SGD) (Robbins & Monro, 1951). Such an update can be considered as a state-transition dynamics where the meta-parameter $\mathbf{x}$ is the state and the weighting vector $\mathbf{u}$ is the action. In addition, the set of feasible re-weighting vectors, $\mathcal{U}$, can be written as a result of an optimisation that minimises some cost function $c$, i.e.,:

$$\mathcal{U} = \left\{ \{\mathbf{u}_t^*\}_{t=1}^T : \{\mathbf{u}_t^*\}_{t=1}^T = \arg\min_{\{\mathbf{u}\}_{t=1}^T} \sum_{t=1}^T c(\mathbf{x}_t, \mathbf{u}_t) \quad \text{s.t.: } \mathbf{x}_{t+1} = f(\mathbf{x}_t, \mathbf{u}_t) \text{ and } \mathbf{x}_1 \text{is given} \right\}, \quad (4)$$

where $\mathbf{x}_1$ is the initialisation of the meta-learning parameter, and the subscript denotes the time step.

Since our interest is the convergence speed and the generalisation of the learnt meta-model, we define the cost function as an un-discounted sum of uniformly-weighted validation losses of a sequence of $T$ mini-batches, each has $M$ tasks, plus a penalisation on the action $\mathbf{u}$, i.e., a Gaussian prior with mean $\mu_u$ and precision $\beta_u$. For the state-transition dynamics $f$, it is assumed to be the SGD to simplify the analysis. Such assumptions result in the following state-transition dynamics and cost function:

$$f(\mathbf{x}_t, \mathbf{u}_t) = \mathbf{x}_t - \alpha \boldsymbol{\nabla}_{\mathbf{x}_t} \left[ \mathbf{u}_t^\top \mathsf{L}(\mathbf{x}_t) \right] \quad \text{and} \quad c(\mathbf{x}_t, \mathbf{u}_t) = \mathbf{1}_M^\top \mathsf{L}(\mathbf{x}_t) + {}^{\beta_u}/{}_2 \|\mathbf{u}_t - \mu_u \mathbf{1}_M\|^2, \quad (5)$$

where: $\mathsf{L}(\mathbf{x})$ is an $M$-dimensional vector containing $M$ validation losses $\mathsf{L}_i(\mathbf{x})$ defined in (3), $\mathbf{1}_M$ is an $M$-dimensional vector with all elements equal to 1 and $\|.\|$ denotes the $\ell_2$-norm.

Note that the action $\mathbf{u}_t$ is not necessarily normalised to 1, as this constraint may not be effective in certain scenarios. For example, consider a mini-batch consisting entirely of familiar tasks and another consisting entirely of unfamiliar tasks. Our hypothesis is to assign small weights to the familiar tasks in the former mini-batch and large weights to the unfamiliar tasks in the latter mini-batch to encourage diverse learning. However, normalizing $\mathbf{u}_t$ to 1 would be disadvantageous as it would make the contribution of tasks in both mini-batches equal, thereby further biasing the meta-learning model towards familiar tasks in the first mini-batch. To address this, we allow the weights to be automatically determined by the optimisation in equation (4) with a Gaussian prior. Nonetheless, one can choose to normalize $\mathbf{u}_t$ to 1 by simply replacing it with softmax($\mathbf{u}_t$).

In general, the solution in (4) cannot be solved exactly, but approximately using iterative methods such as DDP or iLQR. Given the state-transition dynamics $f(\mathbf{x}_t)$ follows the formulation of a first-order gradient-

based optimiser (see Eq. (5)), $f(\mathbf{x}_t)$ consists of the first gradient of the weighted loss $\mathbf{u}_t^\top \mathsf{L}(\mathbf{x}_t)$ w.r.t. $\mathbf{x}_t$. Hence, applying DDP will result in an intractable solution since DDP requires the second gradient of $f(\mathbf{x}_t)$, corresponding to the third gradient of the weighted loss $\mathbf{u}_t^\top \mathsf{L}(\mathbf{x}_t)$. In contrast, iLQR needs only the first gradient of $f(\mathbf{x}_t)$, which corresponds to the second gradient of the weighted loss $\mathbf{u}_t^\top \mathsf{L}(\mathbf{x}_t)$. Although this means that iLQR no longer exhibits the quadratic convergence rate as DDP, in the context of meta-learning, the significant reduction in computation out-weights the speed of convergence for the task weighting vector $\mathbf{u}$. In this paper, we use iLQR to solve for the re-weighting vectors $\{\mathbf{u}_t^*\}_{t=1}^T$ in (4).

### 3.2 Practical task-weighting method based on trajectory optimisation

The approximation using Taylor's series on the state-transition dynamics and cost function is shown in Appendix A. This approximation leads to the calculation of two Hessian matrices: $\mathbf{F}_{\mathbf{x}_t}$ in the transition dynamics, and $\mathbf{C}_{\mathbf{x}_t,\mathbf{x}_t}$ in the cost function (see their definitions in Eq. (2)). In addition, while performing recursive backward iLQR (see Algorithm 2 in Appendix D), we need to calculate another intermediate matrix of the *cost-to-go* in (28) (please refer to Appendix B), denoted as $\mathbf{V}_t$, which is analogous to an inverse Hessian matrix in Newton's method. Naively calculating these Hessian matrices comes at the quadratic complexity $\mathcal{O}(D^2)$ ($D$ is the dimension of $\mathbf{x}$) in terms of running time and storage, resulting in an intractable solution for large-scaled models. To address such issue, the two Hessian matrices $\mathbf{F}_{\mathbf{x}_t}$ and $\mathbf{C}_{\mathbf{x}_t,\mathbf{x}_t}$ may be approximated by their diagonals which can be efficiently computed using the Hutchinson's method (Bekas et al., 2007). However, as the size of the model increases, using a few samples from the uniform Rademacher distribution produces noisy estimations of the Hessian diagonals, resulting in a poor approximation (Yao et al., 2021). Instead of calculating the Hessian diagonals, we use the Gauss-Newton diagonals as replacements. As the Gauss-Newton matrix is known to be a good approximation of the Hessian matrix (Martens, 2010; Botev et al., 2017), this, therefore, results in a good approximation for the Hessian operator. In addition, Gauss-Newton diagonals can be efficiently calculated using a single backward pass (Dangel et al., 2020). For the matrix $\mathbf{V}_t$, we approximate it by its diagonal matrix. Since matrix $\mathbf{V}_t$ is analogous to the inverse Hessian matrix in Newton's method, approximating it by its diagonal means performing Newton's method separately for each coordinate, which holds when the diagonal of $\mathbf{V}_t$ is dominant. We also provide some additional results using full matrix $\mathbf{V}_t$ in Appendix H. In general, we do not observe any significant difference in terms of accuracy evaluated on the validation set between the diagonal approximation and the one with full Gauss-Newton matrix. Nevertheless, these approximation increases the tractability of our proposed method, allowing to implement the proposed method for very large models, such as deep neural networks. The complete algorithm of the proposed task-weighting meta-learning approach is outlined in Algorithm 1.

To simplify the implementation and convergence analysis, we select the nominal actions that coincide with the uniform weighting, meaning that: $\hat{\mathbf{u}}_{ti} = 1/M, \forall t \in \{1, \ldots, T\}, i \in \{1, \ldots, M\}$. In addition, we constrain that all elements of the weighting vector or action $\mathbf{u}$ are non-negative since each task would either contribute more or less or even not contribute to the learning for $\mathbf{x}$. This constraint is incorporated into the stopping condition for iLQR shown in step 16 of Algorithm 1. If there is at least one element $\mathbf{u}_{ti}, t \in \{1, \ldots, T\}, i \in \{1, \ldots, M\}$ being negative, the backtracking line search will iterate one more time with $\varepsilon$ decaying toward 0, forcing $\mathbf{u}_t$ to stay close to the nominal $\hat{\mathbf{u}}_t$. Thus, in the worst-case, $\varepsilon$ is reduced to 0, making $\mathbf{u}_t$ coincide with $\hat{\mathbf{u}}_t$, which is the uniform weighting. We also provide a complexity analysis of Algorithm 1 in Appendix E.

## 4 Convergence analysis

We prove that the training process using TOW to weight tasks converges to an $\epsilon_0$-stationary point where $\epsilon_0$ is greater than some positive constant. Before analysing the convergence of TOW, we state a lemma bounding the norm of the weighting vector (or action) $\mathbf{u}_t$ obtained from iLQR:

**Lemma 1.** *If $\mathbf{u}_t$ is a stationary action of a nominal action $\hat{\mathbf{u}}_t$ obtained from iLQR, then:*

$$\exists \delta > 0 : \|\mathbf{u}_t - \hat{\mathbf{u}}_t\| \leq \delta.$$

*Proof.* Please refer to Appendix F.2.1 for the detailed proof. □

---

**Algorithm 1** Task-weighting for meta-learning

---

1: **procedure** TOW TRAINING($\mathbf{x}_1, n_{\text{iLQR}}$)
2:     ▷ $\mathbf{x}_1$: *initial meta-parameter*                                             ◁
3:     ▷ $n_{\text{iLQR}}$: *number of iterations in iLQR*                                ◁
4:     **while x** is not converged **do**
5:        get $T$ mini-batches, each consists of $M$ tasks
6:        $\hat{\mathbf{u}}_t \leftarrow {}^1\!/_M \mathbf{1}_M, \forall t \in \{1, \ldots, T\}$
7:        $\hat{\mathbf{x}}_{t+1} \leftarrow f(\hat{\mathbf{x}}_t, \hat{\mathbf{u}}_t), \forall t \in \{1, \ldots, T\}$
8:        **for** $n = 1, \ldots, n_{\text{iLQR}}$ **do**
9:           $\{\mathbf{K}_t, \mathbf{k}_t\}_{t=1}^T, \theta_1 \leftarrow \text{ILQRBACKWARD}(\{\hat{\mathbf{x}}_t, \hat{\mathbf{u}}_t\}_{t=1}^T)$         ▷ *Algorithm 2 (Appendix D)*
10:           $\varepsilon = 2$
11:           **repeat**                                         ▷ *Backtracking line search*
12:              $\varepsilon \leftarrow \frac{1}{2}\varepsilon$
13:              **for** $t = 1 : T$ **do**                           ▷ *iLQR forward pass*
14:                 $\mathbf{u}_t = \mathbf{K}_t(\mathbf{x}_t - \hat{\mathbf{x}}_t) + \varepsilon \mathbf{k}_t + \hat{\mathbf{u}}_t$
15:                 $\mathbf{x}_{t+1} = f(\mathbf{x}_t, \mathbf{u}_t)$
16:           **until** $J(\mathbf{u}_{1:N}) - J(\hat{\mathbf{u}}_{1:N}) \leq \frac{1}{2}\varepsilon\theta_1$ and $\mathbf{u}_{ti} \geq 0$          ▷ *Eq. (40)*
17:           $\{\hat{\mathbf{x}}_t\}_{t=1}^T \leftarrow \{\mathbf{x}_t\}_{t=1}^T$                       ▷ *Update nominal state*
18:           $\{\hat{\mathbf{u}}_t\}_{t=1}^T \leftarrow \{\mathbf{u}_t\}_{t=1}^T$
19:        $\mathbf{x}_1 \leftarrow \mathbf{x}_T$                                   ▷ *Update meta-parameter*
20:     **return** $\mathbf{x}_1$

---

### 4.1 Assumptions on the boundedness and smoothness of the loss function and its gradients

To analyse the convergence of a general non-convex function, one typically assumes that the loss function, its first derivative and second derivative are bounded and Lipschitz-continuous (Collins et al., 2020; Fallah et al., 2020) as shown in the following Assumptions 1 to 3, respectively.

#### 4.1.1 Boundedness and smoothness of the loss function

**Assumption 1.** *The loss function of interest, $\ell$, mentioned in* (3) *is B-bounded and L-Lipschitz.*

Formally, Assumption 1 means that the loss function $\ell$ has the following properties:

- Boundedness: $\exists B > 0 : \forall \mathbf{x} \in \mathbb{R}^D, |\ell(\mathbf{s}_{ij}, y_{ij}; \mathbf{x})| \leq B$,

- Lipschitz continuity: $\exists L > 0 : \forall \widetilde{\mathbf{x}}, \overline{\mathbf{x}} \in \mathbb{R}^D, |\ell(\mathbf{s}_{ij}, y_{ij}; \widetilde{\mathbf{x}}) - \ell(\mathbf{s}_{ij}, y_{ij}; \overline{\mathbf{x}})| \leq L\|\widetilde{\mathbf{x}} - \overline{\mathbf{x}}\|$.

The boundedness assumption is to bound the second moment of the loss function, while the Lipschitz-continuity assumption of the loss function $\ell$ implies that the gradient norm of $\ell$ w.r.t. $\mathbf{x}$ is bounded above (see Lemma 14 in Appendix F.4):

$$\|\boldsymbol{\nabla}_{\mathbf{x}}\ell(\mathbf{s}, y; \mathbf{x})\| \leq L. \tag{6}$$

#### 4.1.2 Smoothness of the Jacobian and Hessian

We then state the assumptions about the gradient and Hessian of the loss function $\ell$ as follows:

**Assumption 2.** *The gradient of the loss function $\ell(\mathbf{s}, y; \mathbf{x})$ w.r.t. $\mathbf{x}$ is S-Lipschitz.*

Assumption 2 means that:

$$\exists S > 0 : \forall \widetilde{\mathbf{x}}, \overline{\mathbf{x}} \in \mathbb{R}^D, \|\boldsymbol{\nabla}_{\mathbf{x}}\ell(\mathbf{s}_{ij}, y_{ij}; \widetilde{\mathbf{x}}) - \boldsymbol{\nabla}_{\mathbf{x}}\ell(\mathbf{s}_{ij}, y_{ij}; \overline{\mathbf{x}})\| \leq S\|\widetilde{\mathbf{x}} - \overline{\mathbf{x}}\|.$$

**Assumption 3.** *The Hessian matrix $\boldsymbol{\nabla}_{\mathbf{x}}^2\ell(\mathbf{s}, y; \mathbf{x})$ is $\rho$-Lipschitz.*

Assumption 3 implies that:

$$\exists \rho > 0 : \forall \widetilde{\mathbf{x}}, \overline{\mathbf{x}} \in \mathbb{R}^D, \left\| \boldsymbol{\nabla}_{\mathbf{x}}^2 \ell(\mathbf{s}_{ij}, y_{ij}; \widetilde{\mathbf{x}}) - \boldsymbol{\nabla}_{\mathbf{x}}^2 \ell(\mathbf{s}_{ij}, y_{ij}; \overline{\mathbf{x}}) \right\| \le \rho \| \widetilde{\mathbf{x}} - \overline{\mathbf{x}} \|.$$

These assumptions are used to bound the gradient of the "true" validation loss of task $\mathcal{T}_i$ (see Lemma 12 in Appendix F.2.3), which is defined as follows:

$$\bar{\mathsf{L}}_i(\mathbf{x}) = \mathbb{E}_{\mathcal{D}_i^{(q)}} \left[ \ell \left( \mathbf{s}_{ij}^{(q)}, y_{ij}^{(q)}; \phi(\mathbf{x}) \right) \right], \tag{7}$$

where $\mathbb{E}_{\mathcal{D}_i^{(q)}}$ indicates the expectation over all data pairs $\{(\mathbf{s}_{ij}^{(q)}, y_{ij}^{(q)})\}_{j=1}^{+\infty}$ sampled from the true (validation) probability distribution $\mathcal{D}_i^{(q)}$.

Given the above assumptions and lemmas, the convergence of TOW can be shown in Theorem 2. We also provide two examples of loss functions satisfying Assumptions 1 to 3 in Appendix I.

## 4.2 The convergence of the proposed method

**Theorem 2** (Main result). *If Assumptions 1 to 3 hold, the learning rate $\alpha < {2}/{\widetilde{S}(\delta\sqrt{M}+1)}$, and $\mathbf{z}$ is randomly sampled from $\{\mathbf{x}_t\}_{t=1}^{T_{\text{iter}}}$ returned by Algorithm 1, then:*

$$\mathbb{E}_{\mathbf{z} \sim \{\mathbf{x}_t\}_{t=1}^{T_{\text{iter}}}} \left[ \mathbb{E}_{\left(\mathcal{D}_{1:t}^{(q)}\right)^M} \left[ \left\| \boldsymbol{\nabla}_{\mathbf{z}} \mathbf{u}_t^\top \bar{\mathsf{L}}_{1:M}(\mathbf{z}) \right\|^2 \right] \right] \le \epsilon_0 + \frac{\kappa}{T_{\text{iter}}},$$

*where:*

$$\epsilon_0 = \frac{4\delta B\sqrt{M} + \alpha^2 \widetilde{\sigma}^2 \widetilde{S}\left(\delta\sqrt{M}+1\right)}{\alpha\left[2 - \alpha\widetilde{S}\left(\delta\sqrt{M}+1\right)\right]} > 0, \quad \kappa = \frac{2\mathbf{u}_1^\top \bar{\mathsf{L}}_{1:M}(\mathbf{x}_1)}{\alpha\left[2 - \alpha\widetilde{S}\left(\delta\sqrt{M}+1\right)\right]},$$

*with $T_{\text{iter}}$ as the number of gradient-update for the meta-parameter, and $\mathbb{E}_{\left(\mathcal{D}_{1:t}^{(q)}\right)^M}$ as the expectation taken over all query data sampled from $t$ mini-batches.*

*Proof.* Please refer to Appendix F.3 for the detailed proof. □

Theorem 2 shows that the expectation of squared gradient norm of the weighted validation loss is upper-bounded by a monotonically reducing function w.r.t. the number of iterations $T_{\text{iter}}$. This implies that Algorithm 1 converges in expectation to an $\epsilon_0$-stationary point.

**Remark 2.** *The result in Theorem 2 agrees with previous studies on task-weighting for meta-learning, e.g., (Collins et al., 2020, Ineq. (75)) where the gradient norm is bounded above by some positive constant. The tightness of the bound in Theorem 2 mostly depends on how small the value of $\epsilon_0$ is. In fact, we can observe that $\lim_{\delta \to 0} \epsilon_0 = 0$. Thus, to ensure that $\epsilon_0$ is small, $\delta$ needs to be small. We can make $\delta$ small by imposing a strong prior of $\mathbf{u}$, i.e., setting a large value of $\beta_u$ in Eq. (5), e.g., $\beta_u = 10$ in our experiments presented in Section 5. In addition, the integrated the backtracking line search in Algorithm 1 forces $\mathbf{u}$ to stay close to the uniform weighting $\hat{\mathbf{u}}$, making $\delta$ very small. Another factor contributes to the small value of $\epsilon_0$ is the inverse of the number of tasks in a mini-batch, ${1}/{M}$, as seen in (2). In practice, $M$ cannot be too large and often in the range of 5 to 10 to fit into the memory of a processing unit, e.g., GPU. And since we can impose constraints to make $\delta$ tiny, we can guarantee to obtain a small $\epsilon_0$ to make the bound in Theorem 2 tight.*

## 5 Experiments

### 5.1 N-way k-shot classification

In this section, we evaluate the performance of the proposed trajectory optimisation task weighting (TOW) approach with three baselines: one with uniform weighting, denoted as *uniform*, one with higher weights on

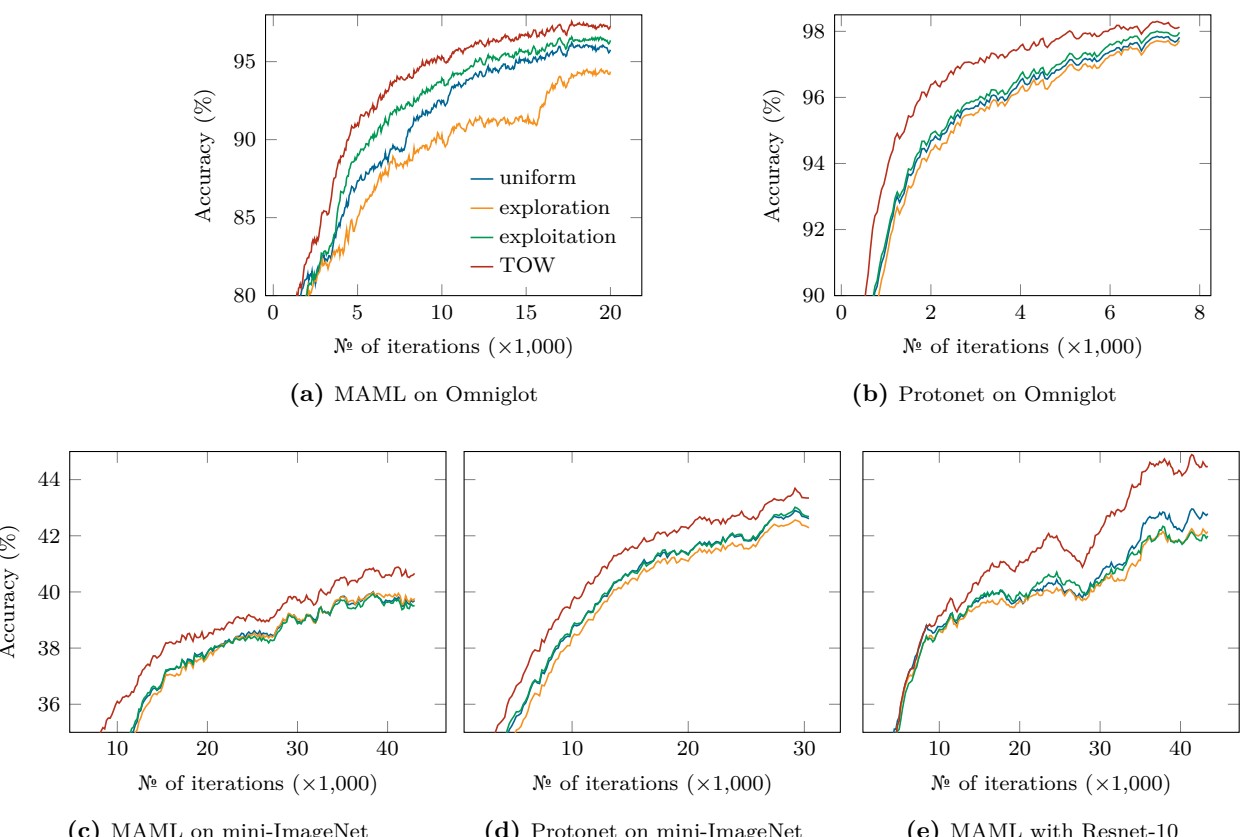

**Figure 1:** Validation accuracy exponential moving average (with smoothing factor 0.1) of different task-weighting strategies evaluated on: (a) and (b) Omniglot, and (c), (d) and (e) mini-ImageNet.

difficult tasks (or tasks with higher losses), denoted as *exploration*, and the other one with higher weights on easier tasks (or tasks with lower losses), denoted as *exploitation*. The experiments are based on *n*-way *k*-shot classification setting used in few-shot learning with tasks formed from Omniglot (Lake et al., 2015) and mini-ImageNet (Vinyals et al., 2016) – the two most widely-used datasets in meta-learning.

**Baselines** Naively implementing the two baselines, *exploration* and *exploitation*, will easily lead to trivial solutions where only the task with largest or smallest loss within a mini-batch is selected. Thus, only one task in each mini-batch is used for learning, and consequently, making the learning noisy and unstable. We, therefore, introduce a prior, denoted as $p(\mathbf{u})$, as a regularisation to prevent many tasks within the same mini-batch from being discarded. Formally:

$$\mathbf{u}^* = \begin{cases} \arg\min_{\mathbf{u}} -\mathbf{u}^\top \boldsymbol{\ell}(\mathbf{x}) - \ln p(\mathbf{u}) & \text{for } \textit{exploration} \\ \arg\min_{\mathbf{u}} \mathbf{u}^\top \boldsymbol{\ell}(\mathbf{x}) - \ln p(\mathbf{u}) & \text{for } \textit{exploitation}. \end{cases} \tag{8}$$

In general, the prior $p(\mathbf{u})$ can be any distribution that has support in $(0, +\infty)$ such as Beta, Gamma or Cauchy distribution. For simplicity, $p(\mathbf{u})$ is selected as a Dirichlet distribution with a concentration $\kappa > 1$ to constrain the weight vector within a probability simplex. One can then use a non-linear optimisation solver to solve (8) to obtain an optimal $\mathbf{u}^*$ for one of the two baselines. In the implementation, we use Sequential Least SQuares Programming (SLSQP) to obtain $\mathbf{u}^*$. Note that the definition of the *exploration* baseline above resembles TR-MAML (Collins et al., 2020), but is applicable for common few-shot learning benchmarks where the number of tasks is large. Similarly, the *exploitation* is an analogy to robust Bayesian data re-weighting (Wang et al., 2017) or *curriculum learning* in single-task learning.

**Table 1:** The classification accuracy averaged on 1,000 random testing tasks generated from Omniglot and 600 tasks from mini-ImageNet with 95 percent confident interval; the bold numbers denote their statistically differences from the ones in the same column.

| | Weighting method | Omniglot | Mini-ImageNet | |
| --- | --- | --- | --- | --- |
| | | | 4 layer CNN | Resnet-10 |
| MAML | Uniform | 94.86 ±0.43 | 48.70 ±1.84[1] | 49.12 ±0.76 |
| | Exploration | 92.64 ±0.52 | 48.80 ±0.72 | 48.72 ±0.74 |
| | Exploitation | 95.34 ±0.42 | 49.22 ±0.74 | 48.44 ±0.76 |
| | TOW | 95.94 ±0.40 | 51.55 ±0.75 | **52.32 ±0.80** |
| Protonet | Uniform | 95.21 ±0.37 | 49.42 ±0.78[2] | _ |
| | Exploration | 94.57 ±0.38 | 48.56 ±0.77 | _ |
| | Exploitation | 95.78 ±0.40 | 48.39 ±0.79 | _ |
| | TOW | **96.84 ±0.37** | **51.05 ±0.80** | _ |

**Table 2:** Running time (in GPU-hour) of different task-weighting methods based on MAML.

| | Omniglot | mini-ImageNet | |
| --- | --- | --- | --- |
| | | CNN | Resnet-10 |
| Exploration | 1.55 | 5.24 | 7.18 |
| Exploitation | 1.55 | 5.24 | 7.18 |
| Uniform | 1.35 | 5.03 | 7.16 |
| TOW | 7.50 | 38.12 | 67.78 |

**Datasets**   Omniglot and mini-ImageNet are used in the evaluation. For Omniglot, we follow the original train-test split (Lake et al., 2015) by using 30 alphabets for training and 20 alphabets for testing. We also utilise the hierarchy structure of alphabet-character to form finer-grained classification tasks to make the classification more difficult than the random train-test split. For mini-ImageNet, we follow the standard train-test split that uses 64 classes for training, 16 classes for validation and 20 for testing (Ravi & Larochelle, 2017) in our evaluation.

**Models used**   The base model used across the experiments is the 4 CNN module network that is widely used in few-shot image classification (Vinyals et al., 2016; Finn et al., 2017). Two common meta-learning algorithms considered in this section include MAML (Finn et al., 2017) and Prototypical Networks (Snell et al., 2017). In MAML, the flattened features are passed to a linear fully-connected layer to classify, while in Prototypical Networks, the classification is based on Euclidean distances to the prototypes of each class.

**Hyper-parameters**   Please refer to Appendix G for the details of hyper-parameters used.

**Results**   Figures 1a to 1d plot the testing accuracy evaluated on 100 validation tasks drawn from Omniglot and mini-ImageNet following the 5-way 1-shot setting. The validation accuracy curves along the training process show that TOW can achieve higher performance comparing to the three baselines on various datasets and meta-learning methods. We also carry out an experiment using Resnet-10 (He et al., 2016) on mini-ImageNet to demonstrate the scalability of TOW. The results on Resnet-10 in Figure 1e shows a a similar observation that TOW out-performs other task-weighting methods. We note that the validation accuracy curves of Resnet-10 fluctuates due to our injected dropout to regularise the network from overfitting since it is known that larger networks, such as Resnet-10 or Resnet-18, severely overfit in the few-shot setting (Nguyen et al., 2020). For the evaluation on testing sets, we follow the standard setting in few-shot learning by measuring the prediction accuracy on 1,000 and 600 testing tasks formed from Omniglot and mini-ImageNet,

---

[1]reported in (Finn et al., 2017)
[2]reported in (Snell et al., 2017)

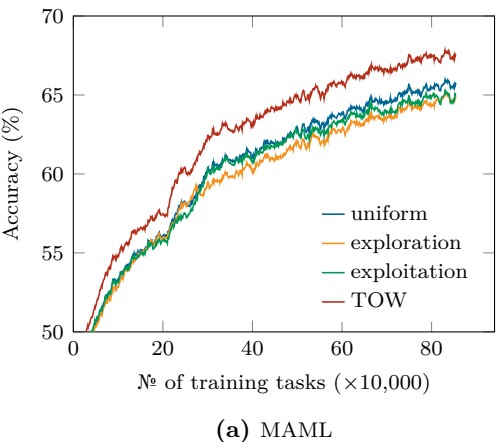

**(a)** MAML

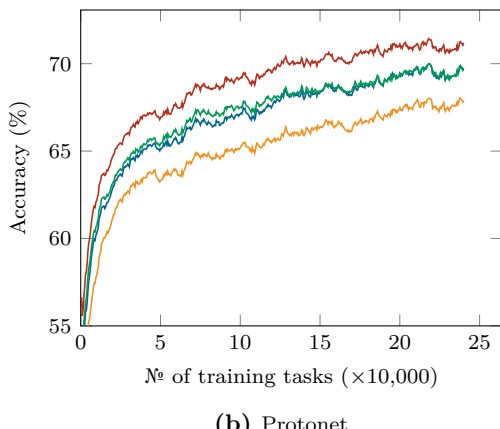

**(b)** Protonet

**Figure 2:** Exponential moving average with smoothing factor 0.1 of the prediction accuracy evaluated on validation tasks formed from the any-shot setting of mini-ImageNet dataset mentioned in Section 5.2 where the base model is a 4-module CNN.

respectively (Vinyals et al., 2016; Finn et al., 2017). The results in Table 1 show that TOW can be at least 2 percent more accurate than the best baseline among Uniform, Exploration, and Exploitation. Note that there a difference between the results shown in Figure 1 and Table 1 due to their differences in terms of (i) the tasks form: one from validation set, while the other from testing set, and (ii) the number of tasks evaluated. Despite the promising results, the downside of TOW is the overhead caused by approximating the cost and state-transition dynamics over $T$ mini-batches of tasks to determine the locally-optimal $\{\mathbf{u}_t^*\}_{t=1}^T$. As shown in Table 2, TOW is about 7 to 9 times slower than the three baselines. We also provide a visualisation of the weights $\mathbf{u}_t$ in Section 5.3.

### 5.2 Any-shot classification

We also follow the *realistic task distribution* (Lee et al., 2020) to evaluate further the performance of TOW. The new setting is mostly similar to $N$-way $k$-shot, except $k$ is not fixed and might be different for each class within a task. Specifically, with a probability of 0.5, the number of shots for each class is sampled from a uniform distribution: $k \sim \mathrm{Uniform}(1, 50)$ to simulate class imbalance. With the other 0.5 probability, the same number of shots $k \sim \mathrm{Uniform}(1, 50)$ is used for all classes within that task. The number of validation (or query) samples is kept at 15 samples per class.

Similar to the experiments carried out in Section 5.1, TOW demonstrates a higher performance compared to the three baselines: exploitation, exploration and uniform along the training process, as shown in Figure 2. In general, TOW can achieve the state-of-the-art results when evaluating on 3,000 testing tasks formed from mini-ImageNet, as shown in Table 3, compared to common meta-learning methods. To further evaluate TOW, we follow the same setting and use the models trained on mini-ImageNet to test on 50 classes of bird images split from CUB dataset. The results in the last column of Table 3 show that TOW can also work well on out-of-distribution tasks formed from CUB compared to most of the methods in the literature. The main reason that explains the worse performance of TOW, compared with Bayesian TAML, is that the meta-learning based methods used by TOW are MAML and Protonet, which have a smaller number of meta-parameters to model tasks than Bayesian TAML.

### 5.3 Weight visualisation

To further understand how the weight $\mathbf{u}_t$ varies along the training process, we conduct a study to monitor the weights $\mathbf{u}_t$ of a set of pre-defined Omniglot tasks. In the first setting, we fix $T = 5$ mini-batches of tasks, each consisting of $M = 5$ tasks formed from the *same* alphabet. In the second case, the configuration is similar, but each mini-batch contains $M = 5$ tasks formed from 5 *different* alphabets. For each case, we

**Table 3:** Prediction results on any-shot classification evaluated on 3,000 testing tasks with 95 percent confident interval; the bold numbers denote the results that are statistically significant. The results of previous methods are reported in (Lee et al., 2020).

| Training set | mini-ImageNet | |
|---|---|---|
| Testing set | mini-ImageNet | CUB |
| MAML (Finn et al., 2017) | 66.64 ±0.22 | 65.77 ±0.24 |
| Meta-SGD (Li et al., 2017) | 69.95 ±0.20 | 65.94 ±0.22 |
| MT-net (Lee & Choi, 2018) | 67.63 ±0.23 | 66.09 ±0.23 |
| ABML (Ravi & Beatson, 2019) | 56.91 ±0.19 | 57.88 ±0.20 |
| Protonet (Snell et al., 2017) | 69.11 ±0.19 | 60.80 ±0.19 |
| Proto-MAML (Triantafillou et al., 2020) | 68.96 ±0.18 | 61.77 ±0.19 |
| Bayesian TAML (Lee et al., 2020) | 71.46 ±0.19 | **71.71 ±0.21** |
| TOW-MAML | 70.02 ±0.24 | 68.34 ±0.25 |
| TOW-Protonet | **72.12 ±0.21** | 64.79 ±0.25 |

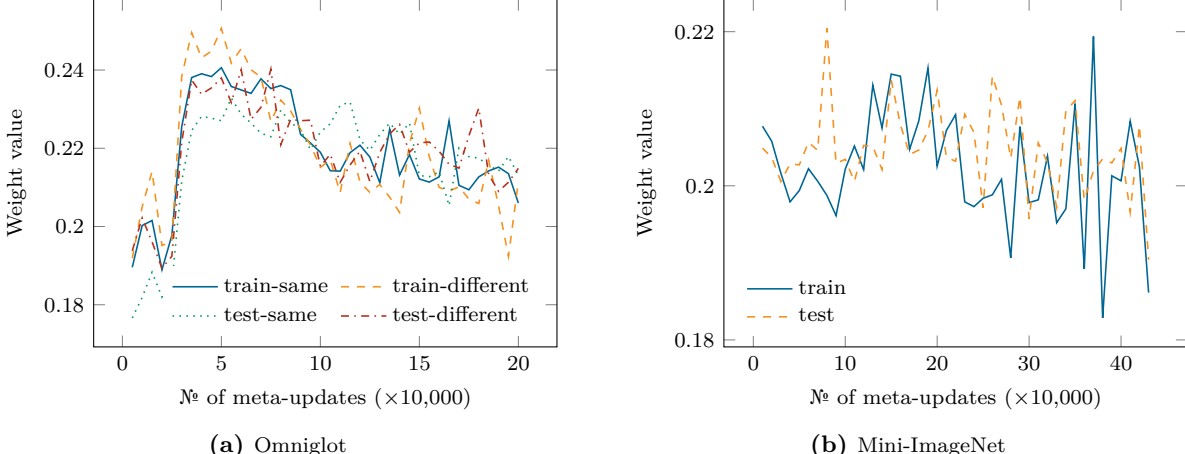

**(a)** Omniglot  **(b)** Mini-ImageNet

**Figure 3:** Visualisation of the weight values associated with (a) Omniglot and (b) mini-ImageNet. For the Omniglot dataset, the tasks are drawn from the same Omniglot alphabet (either training or testing set), and the notation *same* means that all tasks in a mini-batch are formed from one alphabet, while *different* indicates the mini-batch consists of tasks formed from different alphabets.

also run with tasks drawn from training and testing sets. Our desire is to observe how the weight changes and whether there is a difference between training and testing tasks. We plot the weight for a single task belonging to the "controlled" 5-task mini-batch of interest in Figure 3. The results show the variation of the weights of the tasks of interest with a common trend among all the tasks considered, in which the weights are large at the beginning and gradually reduce when training progresses. This is, indeed, expected since most tasks are unfamiliar to the model at the early state, and gradually becomes more familiar. In addition, we observe that the weights for testing tasks are slightly larger than the ones for training tasks.

We also conduct the same ablation study for mini-ImageNet tasks. Since in mini-ImageNet, we do not have any information regarding to the hierarchical structure of classes, we carry out the experiment on training and testing tasks only. One could, of course, utilise the word-net structure to categorise classes into "alphabet" as Omniglot, but for the sake of simplicity, we proceed without including such information. Figure 3b shows the weight values of some tasks at each training checkpoint, which also has a similar but noisier trend as the ones in Omniglot.

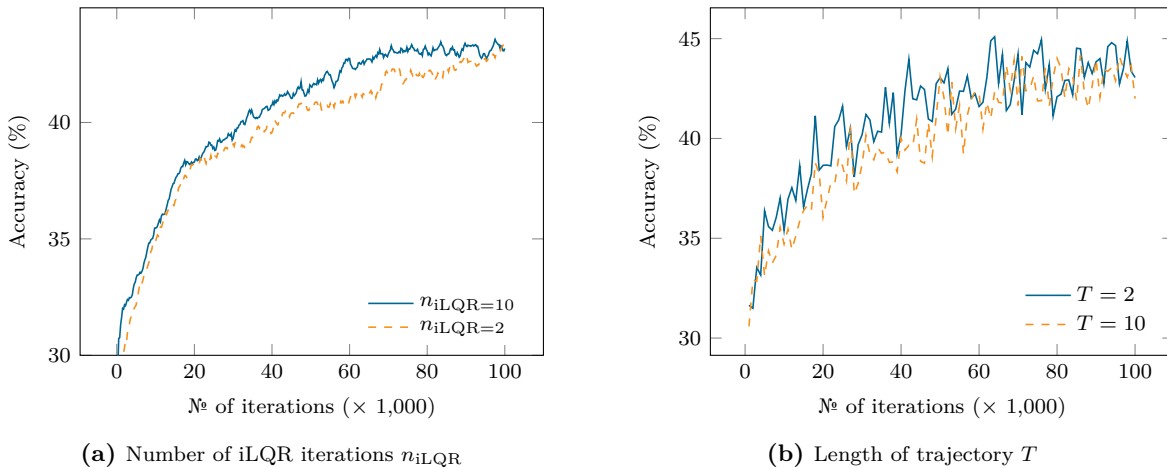

**(a)** Number of iLQR iterations $n_{\text{iLQR}}$        **(b)** Length of trajectory $T$

**Figure 4:** Ablation studies on 5-way 1-shot mini-ImageNet with different values of (a) number of iterations $n_{\text{iLQR}}$ in iLQR (with smoothing factor 0.1), and (b) the length of trajectory $T$ (without smoothing).

## 5.4 Ablation studies

We carry out ablation studies to analyse the effect of each hyper-parameter to the performance of the proposed method, TOW. In particular, we follow the experiment setting of 5-way 1-shot tasks formed from the mini-ImageNet dataset as shown in Section 5.1. Specifically, the model is the 4 convolutional neural network, where the mini-batch size $M$ is set to 5, the number of iLQR iterations $n_{\text{iLQR}}$ is set to 2, $T = 2$, $\beta_u = 10$ and $\mu_u = 1/M$.

### 5.4.1 Number of iLQR iterations

We carry out an ablation study about the effect of the number of iterations used in iLQR, $n_{\text{iLQR}}$, by analysing the validation accuracy on two different settings: one with 2 iterations and the other with 10 iterations. The qualitative result between the two settings in Figure 4a agrees with our intuition that the larger the number of iterations in iLQR, the more accurate the weighting and hence, the larger the validation accuracy. However, the trade-off is the significant increase in terms of training time, which is approximately 6 times larger than the one with 2 iterations.

### 5.4.2 The length of the trajectory $T$

As the proposed method, TOW, is based on iLQR, it also shares some of the properties of iLQR. According to the iLQR paper (Tassa et al., 2012, Section IV. A), the value of the horizon $T$ is a problem-dependent quantity which must be found by trial-and-error. The optimal value of $T$ is, therefore, chosen by further fine-tuning. However, as our aim is to connect the trajectory optimisation and task-weighting in meta-learning and analyse the convergence of the proposed method theoretically, we select the value of $T$ specified in the paper for demonstration purposes. As shown in Figure 4b, increasing $T$ might not always result in the best performance. It is due to the approximation nature of iLQR where the transition dynamics $f$ and the cost function $c$ are linearised and quadraticised via Taylor series. Such approximations are, however, only accurate for a short trajectory (small $T$). For a long trajectory, the future states predicted by iLQR would be deviated further away, making the optimisation sub-optimal. Even if we assume that we have enough computational power to treat the entire training set of tasks as one trajectory and optimise that by iLQR, the result might not be better than a short trajectory.

### 5.4.3 Hyper-parameters of the prior of $\mathbf{u}$

We also present two ablation studies to understand the impact of the Gaussian prior of $\mathbf{u}$ on the performance of TOW. In this case, we vary either $\beta_u$ or $\mu_u$ introduced in Eq. (5), while fixing other hyper-parameters.

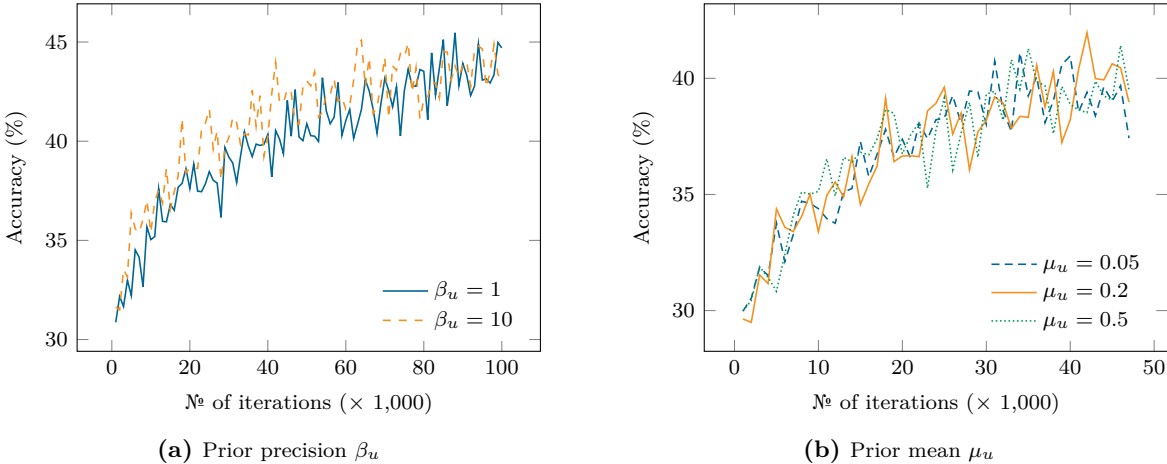

**(a)** Prior precision $\beta_u$

**(b)** Prior mean $\mu_u$

**Figure 5:** Ablation studies on the prior of the weighting vector $\mathbf{u}$ introduced in Eq. (5) (without smoothing).

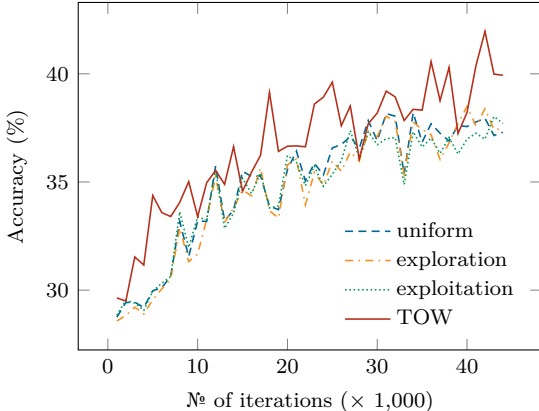

**Figure 6:** Comparison between TOW and the three baselines where each of the baselines is trained 5 times more than TOW, where the additional tasks used to train the baselines are randomly sampled from the training task distribution.

For $\beta_u$, we run on two different values: $\beta_u = 1$ and $\beta_u = 10$ and show the result in Figure 5a. For $\mu_u$, we run an experiment with $\mu_u = 0.05$ and $\mu_u = 0.5$ with a mini-batch size of 5 tasks (corresponding to $\mathbf{u}_{\text{uniform}} = 0.2 \times \mathbf{1}$) and show the result in Figure 5b.

According to this ablation study, a smaller $\beta_u$ results in a slightly better performance compared to one with a larger $\beta_u$, while the signal in Figure 5b is unclear. In the case of $\beta_u$, we hypothesise that a larger value would place a stronger regularisation, forcing the weighting value to be close to a uniform one. Reducing $\beta_u$ would allow the weighting values to fluctuate and easily overfit. In case of $\mu_u$, TOW seems to work well with different values of $\mu_u$.

### 5.4.4 Train the baselines more

To provide a fairer comparison against competing approaches, instead of performing one iteration for each baseline, we increase the number of iterations when training the baselines. In particular, each of the baselines is trained on five times more tasks compared with the number of tasks for training TOW. To compensate for this larger number iterations, we also reduce by five folds the learning rate, so the comparison is fair. The results in Figure 6 show that TOW qualitatively out-performs the three baselines in this fairer setting.

# 6 Related work

Our study directly relates to re-weighting tasks in meta-learning. One recent work is TR-MAML (Collins et al., 2020) which places higher weights on tasks with larger validation losses to optimise performance for worst-case scenarios. However, when the number of training tasks is large, e.g., $\binom{1000}{5} \approx 8.25 \times 10^{12}$ 5-way classification tasks formed from 1000 characters in Omniglot dataset (Lake et al., 2015), learning weight for each training task is intractable. TR-MAML circumvents such issue by clustering tasks into a small number of clusters based on some ad-hoc intuition and learn the weight for each cluster. This, however, reduces the practicability of TR-MAML. Another work, $\alpha$-MAML (Cai et al., 2020), provides an upper-bound on the distance between the weighted risk evaluated on training tasks to the expected risk on testing tasks. The re-weight factors can then be obtained to minimise that upper-bound, reducing the variance between training and testing tasks. Note that $\alpha$-MAML follows a transductive setting where labelled data of testing tasks are available in the training phase, while we follow the inductive setting where no information about testing is available during training. In reinforcement learning (RL), MWL-MAML (Xu et al., 2021) is recently proposed to employ meta-learning to learn the local optimal re-weight factor of each trajectory using a few gradient descent steps. The downside of MWL-MAML is the need of validation trajectories (or validation tasks in meta-learning) that are representative enough to learn those weights. In addition, MWL-MAML also requires to store the weight value of each task as in TR-MAML, making the approach intractable in our setting. Furthermore, TR-MAML, $\alpha$-MAML and MWL-MAML rely on a single mini-batch of tasks to determine the weights without considering the effect of sequence of mini-batches when training a meta-model, potentially rendering sub-optimal solutions. In contrast, our proposed method does not need to cluster tasks nor require additional set of validation tasks. In addition, our proposed method automates the calculation of task-weighting through an optimisation over a sequence of mini-batches, allowing to obtain better local-optimal solutions outside of a single mini-batch of tasks. There are also other studies about task balancing, such as Learn to Balance (L2B) (Lee et al., 2020). However, L2B introduces additional parameters in the task adaptation step (inner-loop), while our method explicitly introduces a weighting vector at the meta-parameter update step (outer-loop).

Our study is also similar to task-weighting in multi-task learning (Chen et al., 2018; Sener & Koltun, 2018; Guo et al., 2018; Liu et al., 2021) where the goal is to obtain an optimal re-weighting vector $\mathbf{u}$ for all tasks. Such modelling can, therefore, work well with a small number of tasks, but potentially fall short when the number of tasks is very large, e.g. in the magnitude of $10^{12}$ training tasks for 5-way Omniglot classification, due to the poor scalability of the computational and storage complexities of that modelling. In comparison, our proposed approach does not explicitly learn the weighting vector for all training tasks, but determines the weighting vector for tasks in current and some following mini-batches via a trajectory optimisation technique. In a loose sense, the multi-task learning approaches can be considered as an analogy to a "batch" learning setting w.r.t. the weighting vector $\mathbf{u}$, while ours is analogous to an "online" learning setting which can scale well to the number of training tasks. There is also a concurrent work – Auto-Lambda (Liu et al., 2022) – that is designed to use meta-learning to learn how to weight tasks in the multi-task setting. However, Auto-Lambda is similar to TR-MAML, which is designed for a fixed number of tasks in the multi-task learning, while being intractable when the number of tasks is large. Furthermore, Auto-Lambda is also similar to MWL-MAML since Auto-Lambda employs validation subsets of training tasks to meta-learn the weighting of tasks.

This paper is motivated from the observation of large variation in terms of prediction performance made by meta-learning algorithms on various testing tasks (Dhillon et al., 2019, Figure 1), implying that the trained meta-model may be biased toward certain training tasks. Such observation may be rooted in task relatedness or task similarity which is a growing research topic in the field of transfer learning. Existing works include task-clustering using k-nearest neighbours (Thrun & O'Sullivan, 1996) or using convex optimisation (Jacob et al., 2009), learning task relationship through task covariance matrices (Zhang & Yeung, 2012), or theoretical guarantees to learn similarity between tasks (Shui et al., 2019). Recently, a large-scale empirical study, known as Taskonomy (Zamir et al., 2018), investigated the relationship between 26 computer vision tasks. Another promising direction to quantify task similarity is to employ task representation, notably Task2Vec (Achille et al., 2019), which is based on Fisher Information matrix to embed tasks into a latent space. One commonality among those studies is that learning from certain training tasks may be beneficial

to generalise to unseen tasks. This suggests the design of a mechanism to re-weight the contribution of each training task to improve the performance of the meta-model of interest.

## 7 Discussion and conclusion

We propose a principled approach based on trajectory optimisation to mitigate the issue of non-uniform distribution of training tasks in meta-learning. The idea is to model the training process in meta-learning by trajectory optimisation with state as meta-parameter and action as the weights of training tasks. The local optimal weights obtained from iLQR – a trajectory optimiser are then used to re-weight tasks to train the meta-parameter of interest. We demonstrate that the proposed approach converges with less number of training tasks and has a final prediction accuracy that out-performs some common hand-crafted task-weighting baselines.

Our proposed method also has some limitations that could be addressed in future work. TOW relies on iLQR which is not ideal for large-scale systems with high dimensional state space such as deep neural networks. Despite the approximation of Hessian matrices to use only diagonals as mentioned in Section 3.1, the linearisation of the state-transition dynamics and quadraticisation of the cost function are still time-consuming, and consequently, reduce TOW's efficiency. Future work might find a faster approximation to optimise the running time for TOW as well as evaluate on large-scaled datasets, such as meta-dataset (Triantafillou et al., 2020).

Furthermore, our method is local in nature due to the Taylor's series approximation about a nominal trajectory used in iLQR. One way to improve further is to define a "global" or "stationary" policy $\pi_\theta(\mathbf{x}_t, \mathbf{u}_t)$, which is similar to Guided Policy Search (Levine & Koltun, 2013; 2014). This policy can then be trained on multiple local optimal trajectories obtained from iLQR. While this approach may offer a superior generalisation for the policy, scalability is an issue since the policy needs to process the high-dimensional state $\mathbf{x}_t$. As a result, a very large model may be required to implement such policy.

### Acknowledgments

This work was funded by the Australian Research Council through grant FT190100525. The computation performed in this paper was managed and supported by the Deep Purple service at the Australian Institute for Machine Learning and the Phoenix HPC service at the University of Adelaide.

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

## Appendix A   Approximation of state-transition dynamics and cost function

### A.1   Linearise the state-transition dynamics

In this section, we present the explicit form of the linearisation for the state-transition dynamics $f$. In particular, the main purpose is to obtain the first gradient of the state-transition dynamics $f$ w.r.t. $\mathbf{x}$ and $\mathbf{u}$ defined in Eq. (2). We consider two cases of $f$: stochastic gradient descent (SGD) and Adam.

#### A.1.1   Stochastic gradient descent (SGD)

The transition dynamics is given as:

$$\mathbf{x}_{t+1} = f(\mathbf{x}_t, \mathbf{u}_t) = \mathbf{x}_t - \alpha \boldsymbol{\nabla}_{\mathbf{x}_t} \left[ \mathbf{u}_t^\top \mathsf{L}(\mathbf{x}_t) \right]. \tag{9}$$

Applying Taylor's expansion to the first order around a state-action pair $(\hat{\mathbf{x}}_t, \hat{\mathbf{u}}_t)$ gives:

$$\mathbf{x}_{t+1} = \hat{\mathbf{x}}_{t+1} + \left( \mathbf{I}_D - \alpha \boldsymbol{\nabla}_{\mathbf{x}}^2 \left[ \hat{\mathbf{u}}_t^\top \mathsf{L}(\hat{\mathbf{x}}_t) \right] \right) (\mathbf{x}_t - \hat{\mathbf{x}}_t) - \alpha \boldsymbol{\nabla}_{\mathbf{x}}^\top \left[ \mathsf{L}(\hat{\mathbf{x}}_t) \right] (\mathbf{u}_t - \hat{\mathbf{u}}_t). \tag{10}$$

If $\delta \mathbf{x}_t = \mathbf{x}_t - \hat{\mathbf{x}}_t$ (similar for $\delta \mathbf{u}_t$), then:

$$\delta \mathbf{x}_{t+1} = \left( \mathbf{I}_D - \alpha \boldsymbol{\nabla}_{\mathbf{x}}^2 \left[ \hat{\mathbf{u}}_t^\top \mathsf{L}(\hat{\mathbf{x}}_t) \right] \right) \delta \mathbf{x}_t + \left( -\alpha \boldsymbol{\nabla}_{\mathbf{x}}^\top \left[ \mathsf{L}(\hat{\mathbf{x}}_t) \right] \right) \delta \mathbf{u}_t. \tag{11}$$

Hence, the coefficient matrices of the Taylor's series for the state-transition dynamics following the SGD update can be expressed as:

$$\boxed{\begin{aligned} \mathbf{F}_{\mathbf{x}_t} &= \mathbf{I}_D - \alpha \boldsymbol{\nabla}_{\mathbf{x}}^2 \left[ \hat{\mathbf{u}}_t^\top \mathsf{L}(\hat{\mathbf{x}}_t) \right] \\ \mathbf{F}_{\mathbf{u}_t} &= -\alpha \boldsymbol{\nabla}_{\mathbf{x}}^\top \left[ \mathsf{L}(\hat{\mathbf{x}}_t) \right]. \end{aligned}}$$

$$\tag{12a}$$
$$\tag{12b}$$

#### A.1.2   Adam

The gradient-based update for the parameter of interest using Adam keeps track of the mean and variance:

$$\begin{cases} \mathbf{m}_t &= \beta_1 \mathbf{m}_{t-1} + (1 - \beta_1) \mathbf{J}_t^\top \mathbf{u}_t \\ \mathbf{v}_t &= \beta_2 \mathbf{v}_{t-1} + (1 - \beta_2) \left( \mathbf{J}_t^\top \mathbf{u}_t \right) \odot \left( \mathbf{J}_t^\top \mathbf{u}_t \right), \end{cases} \tag{13}$$

where:

- $\mathbf{m}_0 = \mathbf{0}$, $\mathbf{v}_0 = \mathbf{0}$,

- $\mathbf{J}_t$ is the Jacobian matrix of the validation losses for tasks in a minibatch $t$-th:

$$\mathbf{J}_t = \boldsymbol{\nabla}_{\mathbf{x}} \left[ \mathsf{L}(\mathbf{x}_t) \right]. \tag{14}$$

- $\odot$ is the elementwise multiplication.

The corrected-bias estimators are then defined as:

$$\begin{cases} \hat{\mathbf{m}}_t &= \dfrac{\mathbf{m}_t}{1 - \beta_1^t} \\ \hat{\mathbf{v}}_t &= \dfrac{\mathbf{v}_t}{1 - \beta_2^t}. \end{cases} \tag{15}$$

The update or state-transition dynamics is then given as:

$$\mathbf{x}_{t+1} = f(\mathbf{x}_t, \mathbf{u}_t) = \mathbf{x}_t - \alpha \frac{\hat{\mathbf{m}}_t}{\sqrt{\hat{\mathbf{v}}_t} + \epsilon}. \tag{16}$$

The update for a new state of the model parameter can also be written by substitution:

$$\mathbf{x}_{t+1} = \mathbf{x}_t - \alpha \frac{\mathbf{m}_t}{1 - \beta_1^t} \frac{1}{\sqrt{\frac{\mathbf{v}_t}{1-\beta_2^t}} + \epsilon}. \tag{17}$$

The approximation using Taylor's expansion up to the first order will result with the following matrices:

$$\mathbf{F}_{\mathbf{x}_t} = \mathbf{I}_D - \frac{\alpha}{1 - \beta_1^t} \left. \boldsymbol{\nabla}_{\mathbf{x}_t} \left[ \frac{\mathbf{m}_t}{\sqrt{\frac{\mathbf{v}_t}{1-\beta_2^t}} + \epsilon} \right] \right|_{\hat{\mathbf{x}}_t, \hat{\mathbf{u}}_t} \tag{18a}$$

$$\mathbf{F}_{\mathbf{u}_t} = -\frac{\alpha}{1 - \beta_1^t} \left. \boldsymbol{\nabla}_{\mathbf{u}_t} \left[ \frac{\mathbf{m}_t}{\sqrt{\frac{\mathbf{v}_t}{1-\beta_2^t}} + \epsilon} \right] \right|_{\hat{\mathbf{x}}_t, \hat{\mathbf{u}}_t}. \tag{18b}$$

In the following, we determine the gradient terms in each of the expressions for $\mathbf{F}_{\mathbf{x}_t}$ and $\mathbf{F}_{\mathbf{u}_t}$. For simplicity, we assume that $\mathbf{m}_{t-1}$ and $\mathbf{v}_{t-1}$ are constant w.r.t. both $\mathbf{x}_{t-1}$ and $\mathbf{u}_{t-1}$.

The gradient w.r.t. $\mathbf{u}_t$ on $\mathbf{m}_t$ and $\mathbf{v}_t$ can be expressed as:

$$\boldsymbol{\nabla}_{\mathbf{u}_t} [\mathbf{m}_t] = (1 - \beta_1)\mathbf{J}_t^\top$$
$$\boldsymbol{\nabla}_{\mathbf{u}_t} [\mathbf{v}_t] = 2(1 - \beta_2)\mathbf{J}_t^\top \odot \left( \mathbf{J}_t^\top \mathbf{u}_t \right). \tag{19}$$

Note that given two functions $f$ (a notation for a general function, not the state-transition dynamic) and $g$

$$\boldsymbol{\nabla}_{\mathbf{u}_t} \left[ \frac{f}{g} \right] = \boldsymbol{\nabla}_{\mathbf{u}_t} \left[ f \odot \frac{1}{g} \right]$$

$$= \boldsymbol{\nabla}_{\mathbf{u}_t} [f] \odot \frac{1}{g} + f \odot \boldsymbol{\nabla}_{\mathbf{u}_t} \left[ \frac{1}{g} \right]$$

$$= \boldsymbol{\nabla}_{\mathbf{u}_t} [f] \odot \frac{1}{g} - f \odot \frac{1}{g^2} \odot \boldsymbol{\nabla}_{\mathbf{u}_t} [g]. \tag{20}$$

Hence:

$$\boldsymbol{\nabla}_{\mathbf{u}_t} \left[ \frac{\mathbf{m}_t}{\sqrt{\frac{\mathbf{v}_t}{1-\beta_2^t}} + \epsilon} \right] = \frac{\boldsymbol{\nabla}_{\mathbf{u}_t} [\mathbf{m}_t]}{\sqrt{\frac{\mathbf{v}_t}{1-\beta_2^t}} + \epsilon} - \frac{\mathbf{m}_t}{\left( \sqrt{\frac{\mathbf{v}_t}{1-\beta_2^t}} + \epsilon \right)^2} \odot \frac{\boldsymbol{\nabla}_{\mathbf{u}_t} [\mathbf{v}_t]}{2\sqrt{(1 - \beta_2^t)\mathbf{v}_t}}. \tag{21}$$

To calculate the gradient w.r.t. $\mathbf{x}_t$, the update of Adam is rewritten as:

$$\begin{cases} \mathbf{m}_t &= \beta_1 \mathbf{m}_{t-1} + (1 - \beta_1)\boldsymbol{\nabla}_{\mathbf{x}_t} \left[ \mathbf{u}_t^\top \mathsf{L}(\mathbf{x}_t) \right] \\ \mathbf{v}_t &= \beta_2 \mathbf{v}_{t-1} + (1 - \beta_2)\boldsymbol{\nabla}_{\mathbf{x}_t} \left[ \mathbf{u}_t^\top \mathsf{L}(\mathbf{x}_t) \right] \odot \boldsymbol{\nabla}_{\mathbf{x}_t} \left[ \mathbf{u}_t^\top \mathsf{L}(\mathbf{x}_t) \right]. \end{cases} \tag{22}$$

The gradient w.r.t. $\mathbf{x}_t$ on $\mathbf{m}_t$ and $\mathbf{v}_t$ can be expressed as:

$$\boldsymbol{\nabla}_{\mathbf{x}_t} [\mathbf{m}_t] = (1 - \beta_1)\boldsymbol{\nabla}_{\mathbf{x}_t}^2 \left[ \mathbf{u}_t^\top \mathsf{L}(\mathbf{x}_t) \right]$$
$$\boldsymbol{\nabla}_{\mathbf{x}_t} [\mathbf{v}_t] = 2(1 - \beta_2)\boldsymbol{\nabla}_{\mathbf{x}_t} \left[ \mathbf{u}_t^\top \mathsf{L}(\mathbf{x}_t) \right] \odot \boldsymbol{\nabla}_{\mathbf{x}_t}^2 \left[ \mathbf{u}_t^\top \mathsf{L}(\mathbf{x}_t) \right]. \tag{23}$$

Hence:

$$\boldsymbol{\nabla}_{\mathbf{x}_t} \left[ \frac{\mathbf{m}_t}{\sqrt{\frac{\mathbf{v}_t}{1-\beta_2^t}} + \epsilon} \right] = \frac{\boldsymbol{\nabla}_{\mathbf{x}_t} [\mathbf{m}_t]}{\sqrt{\frac{\mathbf{v}_t}{1-\beta_2^t}} + \epsilon} - \frac{\mathbf{m}_t}{\left( \sqrt{\frac{\mathbf{v}_t}{1-\beta_2^t}} + \epsilon \right)^2} \odot \frac{\boldsymbol{\nabla}_{\mathbf{x}_t} [\mathbf{v}_t]}{2\sqrt{(1 - \beta_2^t)\mathbf{v}_t}}. \tag{24}$$

### A.2 Quadraticise cost function

The cost function consists of two terms: validation loss on the query subset and penalisation of the action $\mathbf{u}_t$. Since the cost function itself is already in the second-order function w.r.t. the re-weighting vector $\mathbf{u}$, we only need to quadraticise the function using Taylor's series w.r.t. the meta-parameter $\mathbf{x}$.

#### A.2.1 Quadraticise validation loss

The loss term in (5) can be approximated to second order as:

$$\mathbf{1}_M^\top \mathsf{L}(\mathbf{x}_t) \approx \mathbf{1}_M^\top \mathsf{L}(\hat{\mathbf{x}}_t) + (\mathbf{x}_t - \hat{\mathbf{x}}_t)^\top \boldsymbol{\nabla}_{\mathbf{x}_t} \left[\mathbf{1}_M^\top \mathsf{L}(\hat{\mathbf{x}}_t)\right] + \frac{1}{2}(\mathbf{x}_t - \hat{\mathbf{x}}_t)^\top \boldsymbol{\nabla}_{\mathbf{x}_t}^2 \left[\mathbf{1}_M^\top \mathsf{L}(\hat{\mathbf{x}}_t)\right](\mathbf{x}_t - \hat{\mathbf{x}}_t). \tag{25}$$

#### A.2.2 Quadraticise the penalisation of the action $\mathbf{u}_t$

The penalisation term is already in the quadratic form. Here, it is rewritten following the Taylor's series as follows:

$$\begin{aligned}
\frac{\beta_u}{2}(\mathbf{u}_t - \mu_u \mathbf{1}_M)^\top (\mathbf{u}_t - \mu_u \mathbf{1}_M) &= \frac{\beta_u}{2}\left[(\mathbf{u}_t - \hat{\mathbf{u}}_t) + \hat{\mathbf{u}}_t - \mu_u \mathbf{1}_M\right]^\top \left[(\mathbf{u}_t - \hat{\mathbf{u}}_t) + \hat{\mathbf{u}}_t - \mu_u \mathbf{1}_M\right] \\
&= \frac{1}{2}(\mathbf{u}_t - \hat{\mathbf{u}}_t)^\top (\beta_u \mathbf{I}_M)(\mathbf{u}_t - \hat{\mathbf{u}}_t) + (\mathbf{u}_t - \hat{\mathbf{u}}_t)^\top \beta_u (\hat{\mathbf{u}}_t - \mu_u \mathbf{1}_M). \quad (26)
\end{aligned}$$

Given the locally-quadratic approximation of the cost function in (25) and (26), the coefficient matrices and vector of the quadratic form of the cost function can be written as:

$$\begin{aligned}
\mathbf{C}_{\mathbf{x}_t,\mathbf{x}_t} &= \boldsymbol{\nabla}_{\mathbf{x}_t}^2 \left[\mathbf{1}_M^\top \mathsf{L}(\hat{\mathbf{x}}_t)\right] && \text{(27a)} \\
\mathbf{C}_{\mathbf{x}_t,\mathbf{u}_t} &= \mathbf{C}_{\mathbf{u}_t,\mathbf{x}_t} = \mathbf{0} && \text{(27b)} \\
\mathbf{C}_{\mathbf{u}_t,\mathbf{u}_t} &= \beta_u \mathbf{I}_M && \text{(27c)} \\
\mathbf{c}_{\mathbf{x}_t} &= \boldsymbol{\nabla}_{\mathbf{x}_t} \left[\mathbf{1}_M^\top \mathsf{L}(\hat{\mathbf{x}}_t)\right] && \text{(27d)} \\
\mathbf{c}_{\mathbf{u}_t} &= \beta_u (\hat{\mathbf{u}}_t - \mu_u \mathbf{1}_M). && \text{(27e)}
\end{aligned}$$

## Appendix B   Derivation of iLQR

This section presents the derivation for one iteration of iLQR adopted from the original iLQR papers (Todorov & Li, 2005; Tassa et al., 2012). One difference is at the backtracking line-search where we employ a similar acceptance criterion as in DDP.

For brevity, the finite-horizon discrete-time optimal control problem in (1) is restated as follows:

$$\{\mathbf{u}_t^*\}_{t=1}^T = \arg \min_{\{\mathbf{u}_t\}_{t=1}^T} \sum_{t=1}^T c(\mathbf{x}_t, \mathbf{u}_t) \quad \text{s.t. } \mathbf{x}_{t+1} = f(\mathbf{x}_t, \mathbf{u}_t), \tag{1}$$

Let $Q_t$ be the *cost-to-go* defined as:

$$Q_t(\mathbf{x}_{t:T}, \mathbf{u}_{t:T}) = \sum_{j=t}^T c(\mathbf{x}_j, \mathbf{u}_j), \tag{28}$$

and $V_t$ be the *value* function at time step $t$. In other words, $V_t$ is the cost-to-go given the local optimal action sequence:

$$V_t(\mathbf{x}_t) = \min_{\mathbf{u}_{t:T}} Q_t(\mathbf{x}_{t:T}, \mathbf{u}_{t:T}). \tag{29}$$

According to the Dynamic Programming Principle, the objective function that minimises the cost function over an entire sequence of actions $\mathbf{u}_{t:T}$ can be reduced to a sequence of minimisation over a single action $\mathbf{u}_t$, proceeding backwards in time:

$$
\begin{aligned}
V_t(\mathbf{x}_t) &= \min_{\mathbf{u}_{t:T}} \sum_{j=t}^T c(\mathbf{x}_j, \mathbf{u}_j) \quad \text{(according to the definition of } Q \text{ defined in Eq. (28))} \\
&= \min_{\mathbf{u}_t} \left[ c(\mathbf{x}_t, \mathbf{u}_t) + \min_{\mathbf{u}_{(t+1):T}} \sum_{j=t+1}^T c(\mathbf{x}_j, \mathbf{u}_j) \right] \\
&= \min_{\mathbf{u}_t} \left[ c(\mathbf{x}_t, \mathbf{u}_t) + V(\mathbf{x}_{t+1}) \right] \\
&= \min_{\mathbf{u}_t} \left[ c(\mathbf{x}_t, \mathbf{u}_t) + V(f(\mathbf{x}_t, \mathbf{u}_t)) \right].
\end{aligned}
\tag{30}
$$

By applying the Taylor's series to the first order for the state-transition dynamics $f$ and to the second order for the cost function $c$ about a nominal trajectory $(\hat{\mathbf{x}}_t, \hat{\mathbf{u}}_t)$, the term inside the minimisation in (30) can be approximated to:

$$c(\mathbf{x}_t, \mathbf{u}_t) + V(f(\mathbf{x}_t, \mathbf{u}_t)) \approx \frac{1}{2} \begin{bmatrix} 1 \\ \delta\mathbf{x}_t \\ \delta\mathbf{u}_t \end{bmatrix}^\top \begin{bmatrix} 0 & \mathbf{q}_{\mathbf{x}_t}^\top & \mathbf{q}_{\mathbf{u}_t}^\top \\ \mathbf{q}_{\mathbf{x}_t} & \mathbf{Q}_{\mathbf{x}_t, \mathbf{x}_t} & \mathbf{Q}_{\mathbf{x}_t, \mathbf{u}_t} \\ \mathbf{q}_{\mathbf{u}_t} & \mathbf{Q}_{\mathbf{u}_t, \mathbf{x}_t} & \mathbf{Q}_{\mathbf{u}_t, \mathbf{u}_t} \end{bmatrix} \begin{bmatrix} 1 \\ \delta\mathbf{x}_t \\ \delta\mathbf{u}_t \end{bmatrix}, \tag{31}$$

where:

$$\delta\mathbf{x}_t = \mathbf{x}_t - \hat{\mathbf{x}}_t, \quad \delta\mathbf{u}_t = \mathbf{u}_t - \hat{\mathbf{u}}_t, \tag{32}$$

and

$$
\begin{aligned}
\mathbf{q}_{\mathbf{x}_t} &= \mathbf{c}_{\mathbf{x}_t} + \mathbf{F}_{\mathbf{x}_t}^\top \mathbf{v}_{t+1} & \mathbf{q}_{\mathbf{u}_t} &= \mathbf{c}_{\mathbf{u}_t} + \mathbf{F}_{\mathbf{u}_t}^\top \mathbf{v}_{t+1} \\
\mathbf{Q}_{\mathbf{x}_t, \mathbf{x}_t} &= \mathbf{C}_{\mathbf{x}_t, \mathbf{x}_t} + \mathbf{F}_{\mathbf{x}_t}^\top \mathbf{V}_{t+1} \mathbf{F}_{\mathbf{x}_t} & \mathbf{Q}_{\mathbf{u}_t, \mathbf{u}_t} &= \mathbf{C}_{\mathbf{u}_t, \mathbf{u}_t} + \mathbf{F}_{\mathbf{u}_t}^\top \mathbf{V}_{t+1} \mathbf{F}_{\mathbf{u}_t} \\
\mathbf{Q}_{\mathbf{x}_t, \mathbf{u}_t} &= \mathbf{Q}_{\mathbf{u}_t, \mathbf{x}_t}^\top = \mathbf{C}_{\mathbf{x}_t, \mathbf{u}_t} + \mathbf{F}_{\mathbf{x}_t}^\top \mathbf{V}_{t+1} \mathbf{F}_{\mathbf{u}_t},
\end{aligned}
\tag{33}
$$

with $\mathbf{F}_{\mathbf{x}_t}$ and $\mathbf{F}_{\mathbf{u}_t}$ are the first gradients of the state-transtition dynamics $f$ w.r.t. the variable in the subscripts, $\mathbf{c}_{(.)}$ and $\mathbf{C}_{(.)}$ are the first and second gradients of the cost function $c$ w.r.t. the variables in the subscripts (please refer to Eq. (2) for details), and $\mathbf{v}_{t+1}$ and $\mathbf{V}_{t+1}$ are the first and second gradients of the value function $V$ w.r.t. the state $\mathbf{x}_{t+1}$.

Minimising the quadratic problem in (31) w.r.t. $\delta \mathbf{u}_t$ results in:

$$\delta \mathbf{u}_t^* = \mathbf{K}_t \delta \mathbf{x}_t + \mathbf{k}_t, \tag{34}$$

where:

$$\mathbf{K}_t = -\mathbf{Q}_{\mathbf{u}_t, \mathbf{u}_t}^{-1} \mathbf{Q}_{\mathbf{u}_t, \mathbf{x}_t} \qquad \mathbf{k}_t = -\mathbf{Q}_{\mathbf{u}_t, \mathbf{u}_t}^{-1} \mathbf{q}_{\mathbf{u}_t}, \tag{35}$$

Substituting the action $\delta \mathbf{u}_t$ obtained in (34) into (31) gives a quadratic model:

$$V_t(\mathbf{x}_t) \approx \frac{1}{2} \delta \mathbf{x}_t^\top \mathbf{V}_t \delta \mathbf{x}_t + \delta \mathbf{x}_t^\top \mathbf{v}_t + \text{const.}, \tag{36}$$

where $\mathbf{V}_t$ and $\mathbf{v}_t$ are the second and first gradients of the value function $V_t$ w.r.t. the state $\mathbf{x}_t$, which can be expressed as:

$$\mathbf{V}_t = \mathbf{Q}_{\mathbf{x}_t, \mathbf{x}_t} + \mathbf{Q}_{\mathbf{x}_t, \mathbf{u}_t} \mathbf{K}_t \qquad\qquad \mathbf{v}_t = \mathbf{q}_{\mathbf{x}_t} + \mathbf{Q}_{\mathbf{x}_t, \mathbf{u}_t} \mathbf{k}_t. \tag{37}$$

Hence, one can recursively calculate the local quadratic model $\{\mathbf{V}_t, \mathbf{v}_t\}$ of the value function $V_t$, and the linear controller $\{\mathbf{K}_t, \mathbf{k}_t\}$ backward through time. Once it is completed, a new trajectory can be calculated using the forward pass with the general non-linear state-transition dynamics $f$ as follows:

$$\begin{aligned} \hat{\mathbf{x}}_1 &= \mathbf{x}_1 \\ \mathbf{u}_t &= \mathbf{K}_t \left( \mathbf{x}_t - \hat{\mathbf{x}}_t \right) + \mathbf{k}_t + \hat{\mathbf{u}}_t \\ \mathbf{x}_{t+1} &= f \left( \mathbf{x}_t, \mathbf{u}_t \right). \end{aligned} \tag{38}$$

Although the trajectory obtained in (38) converges to a local minimum for the approximate model of the value function $V_t$, it does not guarantee the convergence for general non-linear models such as the one in (1). The reason is that the new trajectory might deviate too far from the nominal trajectory, resulting in a poor Taylor approximation of the true model. To overcome, a backtracking linear-search with parameter $\varepsilon \in (0, 1]$ is introduced:

$$\mathbf{u}_t = \mathbf{K}_t \left( \mathbf{x}_t - \hat{\mathbf{x}}_t \right) + \varepsilon \mathbf{k}_t + \hat{\mathbf{u}}_t. \tag{39}$$

The criterion to accept the trajectory produced in the iteration with backtracking line-search is similar to the one in DDP:

$$J(\mathbf{u}_{1:T}) - J(\hat{\mathbf{u}}_{1:T}) < \frac{1}{2} \varepsilon \theta_1, \tag{40}$$

where

$$J(\mathbf{u}_{1:T}) = \sum_{t=1}^{T} c(\mathbf{x}_t, \mathbf{u}_t) \tag{41}$$

$$\theta_t = \theta_{t+1} - \mathbf{q}_{\mathbf{u}_t}^\top \mathbf{Q}_{\mathbf{u}_t, \mathbf{u}_t} \mathbf{q}_{\mathbf{u}_t}. \tag{42}$$

Hence, in the worst case when the new trajectory strays too far from the model's region of validity, then $\varepsilon \to 0$ and the trajectory is the same as the nominal trajectory. The procedure of one iLQR iteration is outlined in Algorithm 2 in Appendix D. In addition, the convergence proof for iLQR adapted from DDP (Yakowitz & Rutherford, 1984) is provided in Appendix C to complete the analysis.

## Appendix C  Convergence of iLQR

To prove the convergence of iLQR algorithm, we rely on Theorem 3 in (Polak, 1971, p. 14), which is re-stated in Theorem 3 in Appendix C.1

### C.1  Auxiliary to prove convergence

**Definition 1** (Algorithm model). *Given $a : \mathcal{T} \to \mathcal{T}$ is an algorithm, and $c : \mathcal{T} \to \mathcal{R}$ is a function used as stopping criterion, the algorithm model can be described as:*

1. *Compute a $z_0 \in \mathcal{T}$.*

2. *Set $i = 0$.*

3. *Compute $a(z_i)$ .*

4. *Set $z_{i+1} = a(z_i)$.*

5. *If $c(z_{i+1}) \geq c(z_i)$, stop;[3] else set $i = i + 1$ and go to step 3.*

Theorem 3 will show what such an algorithm will compute.

**Theorem 3** ((Polak, 1971, Theorem 3, p. 14)). *Suppose that:*

*(i) $c(.)$ is either continuous at all non-desirable points $z \in \mathcal{T}$, or else $c(z)$ is bounded from below for $z \in \mathcal{T}$;*

*(ii) for every $z \in \mathcal{T}$ which is not desirable, there exist an $\varepsilon(z) > 0$ and a $\delta(z) < 0$ such that:*

$$c(a(z')) - c(z') \leq \delta(z) < 0, \forall z' \in B(z, \varepsilon(z)) = \{z \in \mathcal{T} : \|z' - z\|_{\mathcal{B}} \leq \varepsilon(z)\}. \tag{43}$$

*Then, either the sequence $\{z_i\}$ constructed by algorithm defined in Definition 1 is finite and its next to last element is desirable, or else it is infinite and every accumulation point of $\{z_i\}$ is desirable.*

### C.2  Proof of iLQR convergence

Since iLQR is an algorithm model as described in Definition 1, to prove its convergence property, one needs to assert that the 2 conditions in Theorem 3 are met.

First condition is satisfied since the cost $J(\mathbf{u}_{1:T})$ is continuous, and twice differentiable. We will prove that iLQR satisfies the second condition of Theorem 3.

From (39) and (42), the sequence of actions obtained from iLQR $\mathbf{u}_{1:T}$ and $\theta_1$ depend on the nominal trajectory. Hence, we explicitly denote them as $\mathbf{u}(\varepsilon, \hat{\mathbf{u}})$ and $\theta_1(\hat{\mathbf{u}})$. In addition, since the loss function $\ell(.)$ is assumed to be twice differentiable, the cost function and state-transition dynamics are continuous. Hence, both $\mathbf{u}(\varepsilon, \hat{\mathbf{u}})$ and $\theta_1(\hat{\mathbf{u}})$ are also continuous.

The convergence proof for iLQR is presented in Appendix C.2.2. The proof requires some auxiliary lemmas in Appendix C.2.1.

#### C.2.1  Auxiliary lemmas

**Lemma 4.** *If $\mathbf{u}(\varepsilon, \bar{\mathbf{u}})$ is a non-stationary sequence of actions, then $\theta_1 < 0$.*

---

[3]A direct test for determining if $z_i$ is desirable may be substituted for the test $c(z_{i+1}) \geq c(z_i)$.

*Proof.* It is apparent from the derivation of iLQR that $\mathbf{q}_{\mathbf{u}_t} = 0, \forall t \in \{1, \dots, T\}$ if and only if $\hat{\mathbf{u}}$ is a stationary sequence of actions. Hence, by negation, if $\hat{\mathbf{u}}$ is a non-stationary sequence of actions, $\mathbf{q}_{\mathbf{u}_t} \neq 0$ for some time steps $t \in \{1, \dots, T\}$.

Also note that, if $\mathbf{u}(\varepsilon, \bar{\mathbf{u}})$ is non-stationary, then $\hat{\mathbf{u}}$ is also non-stationary. In addition, if the Hessian matrix w.r.t. action $\mathbf{u}$ is assumed to be positive-definite (PSD), then from the construction, $\mathbf{Q}_{\mathbf{u}_t,\mathbf{u}_t}$ and its inverse are also PSD.

Th PSD of $\mathbf{Q}_{\mathbf{u}_t,\mathbf{u}_t}^{-1}$ combining with (42) leads to the fact that $\theta_1(\hat{\mathbf{u}}) < 0$ for any non-stationary sequence of actions $\hat{\mathbf{u}}$. $\qquad\square$

**Lemma 5.** *If $\mathbf{u}(\varepsilon, \bar{\mathbf{u}})$ is a non-stationary sequence of actions, then:*

$$J(\mathbf{u}(\varepsilon, \bar{\mathbf{u}})) - J(\bar{\mathbf{u}}) = \varepsilon\theta_1(\bar{\mathbf{u}}) + \mathcal{O}(\varepsilon^2). \tag{44}$$

*Proof.* If we define $\Delta Q_t$ as the resulting incremental change in the cost-to-go $Q$ from the perturbation $\Delta \mathbf{u}_t$ using Taylor expansion, then:

$$\Delta Q_t = Q_t(\mathbf{x}_t, \mathbf{u}_t + \Delta \mathbf{u}_t) - Q_t(\mathbf{x}_t, \mathbf{u}_t) = \mathbf{q}_{\mathbf{u}_t}^\top \Delta \mathbf{u}_t + \mathcal{O}(\|\Delta \mathbf{u}_t\|^2). \tag{45}$$

Note that:

$$J(\mathbf{u}(\varepsilon, \hat{\mathbf{u}})) - J(\hat{\mathbf{u}}) = J(\mathbf{u}_{1:N} + \Delta \mathbf{u}_{1:N}) - J(\mathbf{u}_{1:N})$$

$$= \sum_{t=1}^{T} \Delta Q_t = \sum_{t=1}^{N} \mathbf{q}_{\mathbf{u}_t}^\top \Delta \mathbf{u}_t + \mathcal{O}\left(\|\Delta \mathbf{u}_{1:N}\|^2\right). \tag{46}$$

Taking the derivative w.r.t. $\varepsilon$ gives:

$$\frac{\partial J(\mathbf{u}(\varepsilon))}{\partial \varepsilon} = \sum_{t=1}^{T} \mathbf{q}_{\mathbf{u}_t}^\top \frac{\mathrm{d}\mathbf{u}_t}{\mathrm{d}\varepsilon}. \tag{47}$$

From (39), $\mathbf{u}_t$ is a linear function w.r.t. $\varepsilon$, resulting in:

$$\frac{\mathrm{d}\mathbf{u}_t}{\mathrm{d}\varepsilon} = \mathbf{k}_t = -\mathbf{Q}_{\mathbf{u}_t,\mathbf{u}_t}\mathbf{q}_{\mathbf{u}_t}. \tag{48}$$

Hence:

$$\frac{\partial J(\mathbf{u}(\varepsilon))}{\partial \varepsilon} = -\sum_{t=1}^{T} \mathbf{q}_{\mathbf{u}_t}^\top \mathbf{Q}_{\mathbf{u}_t,\mathbf{u}_t}\mathbf{q}_{\mathbf{u}_t} = \theta_1(\hat{\mathbf{u}}), \tag{49}$$

which implies (44). $\qquad\square$

**Lemma 6.** *If $\mathbf{u}(\varepsilon, \bar{\mathbf{u}})$ is a non-stationary sequence of actions, then there exists $\varepsilon_1$ such that:*

$$J(\mathbf{u}(\varepsilon, \bar{\mathbf{u}})) - J(\hat{\mathbf{u}}) \leq \eta\varepsilon\theta_1(\hat{\mathbf{u}}), \forall \varepsilon \in [0, \varepsilon_1], \forall \eta \in (0.5, 1). \tag{50}$$

*Proof.* The result in Lemma 5 can be re-written as:

$$J(\mathbf{u}(\varepsilon, \hat{\mathbf{u}})) - J(\hat{\mathbf{u}}) = \eta\varepsilon\theta_1(\hat{\mathbf{u}}) + (1 - \eta)\varepsilon\theta_1(\hat{\mathbf{u}}) + \mathcal{O}(\varepsilon^2), \forall \eta \in (0.5, 1). \tag{51}$$

Hence, there must exist a value $\varepsilon = \varepsilon_1 \geq 0$ such that: $(1 - \eta)\theta_1(\hat{\mathbf{u}}) + \mathcal{O}(\varepsilon^2) < 0$. This leads to:

$$J(\mathbf{u}(\varepsilon, \hat{\mathbf{u}})) - J(\hat{\mathbf{u}}) \leq \eta\varepsilon_1\theta_1(\hat{\mathbf{u}}) \leq \eta\varepsilon\theta_1(\hat{\mathbf{u}}), \forall \varepsilon \in [0, \varepsilon_1]. \tag{52}$$

Note that the second inequality holds due to the negativity of $\theta_1$ proved in Lemma 4. This completes the proof. $\qquad\square$

**Lemma 7.** *Given that $\theta_1$ and any action sequence $\mathbf{u}$ are continuous, and for any $\delta > 0$ such that $\|\hat{\mathbf{u}} - \mathbf{u}\| < \delta$, there exists $\varepsilon_2 > 0$ such that:*

$$\theta_1(\mathbf{u}) - \varepsilon_2 < \theta_1(\hat{\mathbf{u}}) < \theta_1(\mathbf{u}) + \varepsilon_2.$$

*Proof.* Since $\theta_1$ is assumed to be continuous, and the variable $\mathbf{u}$ is also continuous, we can employ the definition of continuity of function $\theta_1$ at $\mathbf{u}$ to obtain the following:

$$\forall \varepsilon_2 > 0 \implies \exists \delta > 0 : |\theta_1(\hat{\mathbf{u}}) - \theta_1(\mathbf{u})| < \varepsilon_2, \forall \hat{\mathbf{u}} \in \{\hat{\mathbf{u}} \in \mathcal{U} : \|\hat{\mathbf{u}} - \mathbf{u}\| < \delta\}. \tag{53}$$

The negation form of the statement in (53) can be expressed as:

$$\forall \delta > 0 \implies \exists \varepsilon_2 > 0 : |\theta_1(\hat{\mathbf{u}}) - \theta_1(\mathbf{u})| < \varepsilon_2, \forall \hat{\mathbf{u}} \in \{\hat{\mathbf{u}} \in \mathcal{U} : \|\hat{\mathbf{u}} - \mathbf{u}\| < \delta\}. \tag{54}$$

Solving the inequation above completes the proof. $\qquad\square$

### C.2.2 Convergence of iLQR

**Theorem 8.** *Let the state-transition dynamics $f$ and the cost function $c$ have continuous second partial derivatives w.r.t the continuous state $\mathbf{x}$ and action $\mathbf{u}$. If $\mathbf{u}(\varepsilon, \hat{\mathbf{u}})$ denotes the successive application of iLQR, then any accumulation point of $\mathbf{u}(\varepsilon, \hat{\mathbf{u}})$ is stationary w.r.t the finite-horizon cost $J(\mathbf{u}_{1:T})$.*

*Proof.* We first determine the condition that $\mathbf{u}(\varepsilon, \hat{\mathbf{u}})$ calculated in (39) is the successor of iLQR at $\hat{\mathbf{u}}$. According to iLQR algorithm, $\mathbf{u}(\varepsilon, \hat{\mathbf{u}}) = \text{iLQR}(\hat{\mathbf{u}})$ only if:

$$J(\mathbf{u}(\varepsilon, \hat{\mathbf{u}})) - J(\hat{\mathbf{u}}) \leq \frac{\varepsilon}{2} \theta_1(\hat{\mathbf{u}}). \tag{55}$$

Note that from Lemma 6:

$$J(\mathbf{u}(\varepsilon, \hat{\mathbf{u}})) - J(\hat{\mathbf{u}}) \leq \eta \varepsilon \theta_1(\hat{\mathbf{u}}) < \frac{\varepsilon}{2} \theta_1(\hat{\mathbf{u}}), \forall \eta \in (0.5, 1), \forall \varepsilon \in [0, \varepsilon_1] \tag{56}$$

which indicates that $\mathbf{u}(\varepsilon, \hat{\mathbf{u}})$ is a successor of iLQR when $\varepsilon \in [0, \varepsilon_1]$.

If $\mathbf{u}(\varepsilon, \hat{\mathbf{u}})$ is a successor of iLQR, the acceptance criterion combining with the result in Lemma 7 leads to:

$$J(\text{iLQR}(\hat{\mathbf{u}})) - J(\hat{\mathbf{u}}) \leq \frac{\varepsilon}{2} \theta_1(\hat{\mathbf{u}}) < \frac{\varepsilon}{2} \left[ \theta_1(\mathbf{u}) + \varepsilon_2 \right], \forall \hat{\mathbf{u}} \in \{\hat{\mathbf{u}} \in \mathcal{U} : \|\hat{\mathbf{u}} - \mathbf{u}\| < \delta\}. \tag{57}$$

Note that if $\delta$ is set to be small enough, then $\varepsilon_2$ is also very small, resulting in $\theta_1(\mathbf{u}) + \varepsilon_2 < 0$. If we set $\delta(\hat{\mathbf{u}}) = \delta$ and $\varepsilon(\hat{\mathbf{u}}) = \frac{\varepsilon}{2} \left[ \theta_1(\mathbf{u}) + \varepsilon_2 \right]$, then iLQR satisfies the second condition in Theorem 3. $\qquad\square$

## Appendix D    Trajectory optimisation algorithm(s)

---

**Algorithm 2** Implementation of iLQR backward to determine the controller of interest.

---

1: **procedure** ILQRBACKWARD($\{\hat{\mathbf{x}}_t, \hat{\mathbf{u}}_t\}_{t=1}^T$)
2:     $\mathbf{V}_{T+1} = \mathbf{0}$, and $\mathbf{v}_{T+1} = \mathbf{0}$            ▷ *Quadratic matrix and vector of cost-to-go*
3:     $\theta_{T+1} = 0$            ▷ *Stopping criterion for a nominal trajectory*
4:     **for** $t = T : 1$ **do**            ▷ *Backward through time*
5:         $\mathbf{F}_{\mathbf{x}_t}, \mathbf{F}_{\mathbf{u}_t}$ = linearise dynamics            ▷ *see Appendix A.1*
6:         $\mathbf{C}_{\mathbf{x}_t,\mathbf{x}_t}, \mathbf{C}_{\mathbf{x}_t,\mathbf{u}_t}, \mathbf{C}_{\mathbf{u}_t,\mathbf{x}_t}, \mathbf{C}_{\mathbf{u}_t,\mathbf{u}_t}, \mathbf{c}_{\mathbf{x}_t}, \mathbf{c}_{\mathbf{u}_t}$ = quadraticise cost function     ▷ *see Appendix A.2*

7:         $\mathbf{Q}_{\mathbf{x}_t,\mathbf{x}_t} = \mathbf{C}_{\mathbf{x}_t,\mathbf{x}_t} + \mathbf{F}_{\mathbf{x}_t}^\top \mathbf{V}_{t+1} \mathbf{F}_{\mathbf{x}_t}$            ▷ *2nd derivatives of cost-to-go*
8:         $\mathbf{Q}_{\mathbf{x}_t,\mathbf{u}_t} = \mathbf{F}_{\mathbf{x}_t}^\top \mathbf{V}_{t+1} \mathbf{F}_{\mathbf{u}_t}$
9:         $\mathbf{Q}_{\mathbf{u}_t,\mathbf{x}_t} = \mathbf{F}_{\mathbf{u}_t}^\top \mathbf{V}_{t+1} \mathbf{F}_{\mathbf{x}_t}$
10:        $\mathbf{Q}_{\mathbf{u}_t,\mathbf{u}_t} = \mathbf{C}_{\mathbf{u}_t,\mathbf{u}_t} + \mathbf{F}_{\mathbf{u}_t}^\top \mathbf{V}_{t+1} \mathbf{F}_{\mathbf{u}_t}$

11:        $\mathbf{q}_{\mathbf{x}_t} = \mathbf{c}_{\mathbf{x}_t} + \mathbf{F}_{\mathbf{x}_t}^\top \mathbf{v}_{t+1}$            ▷ *1st derivatives of cost-to-go*
12:        $\mathbf{q}_{\mathbf{u}_t} = \mathbf{c}_{\mathbf{u}_t} + \mathbf{F}_{\mathbf{u}_t}^\top \mathbf{v}_{t+1}$

13:        $\mathbf{K}_t = -\mathbf{Q}_{\mathbf{u}_t,\mathbf{u}_t}^{-1} \mathbf{Q}_{\mathbf{u}_t,\mathbf{x}_t}$            ▷ *Linear controller*
14:        $\mathbf{k}_t = -\mathbf{Q}_{\mathbf{u}_t,\mathbf{u}_t}^{-1} \mathbf{q}_{\mathbf{u}_t}$

15:        $\mathbf{V}_t = \mathbf{Q}_{\mathbf{x}_t,\mathbf{x}_t} + \mathbf{Q}_{\mathbf{x}_t,\mathbf{u}_t} \mathbf{K}_t$            ▷ *2nd derivatives of value function*
16:        $\mathbf{v}_t = \mathbf{q}_{\mathbf{x}_t} + \mathbf{Q}_{\mathbf{x}_t,\mathbf{u}_t} \mathbf{k}_t$            ▷ *1st derivatives of value function*

17:        $\theta_t = \theta_{t+1} - \mathbf{q}_{\mathbf{u}_t}^\top \mathbf{Q}_{\mathbf{u}_t,\mathbf{u}_t} \mathbf{q}_{\mathbf{u}_t}$
18:     **return** $\{\mathbf{K}_t, \mathbf{k}_t\}_{t=1}^T, \theta_1$

---

## Appendix E    Complexity analysis

Before analysing the complexity of the proposed method presented in Algorithm 1, we provide a detail description of the annotations used for brevity in Table 4.

**Table 4:** Detailed description of the notations used in the paper.

| Notation | Description |
|---|---|
| $\mathcal{O}((T_0))$ | Time complexity to train a meta-learning method following a uniform weighting strategy |
| $D$ | Dimension of meta-parameter $\mathbf{x}$ |
| $M$ | Number of tasks within a "meta" mini-batch |
| $m_0 = \max_{i \in \{1,...,M\}} m_i^{(q)}$ | Total number of validation samples within a task |
| $n_{\mathrm{iLQR}}$ | Number of iterations used in iLQR to calculate the re-weighting vector $\mathbf{u}$ |
| $n_{\mathrm{ls}}$ | Number of iterations performing line-search |
| $\eta$ | Number of arithmetic operations in the model of interest |

The downside of TOW is the overhead due to the linearisation and quadraticisation for the state-transition dynamics and cost function, and the calculation to obtain the controller $\mathbf{K}_t$ and $\mathbf{k}_t$ shown in Algorithm 1. If $\mathcal{O}(T_0)$ is the time complexity to train a meta-learning method following a uniform weighting strategy, then the time complexity required by TOW will consist of the following:

- nominal trajectory to obtain $(\hat{\mathbf{x}}_t, \hat{\mathbf{u}}_t)$: $\mathcal{O}(T_0)$

- linearisation and quadraticisation using Gauss-Newton matrices: $\mathcal{O}(n_{\mathrm{iLQR}} m_0 \eta D)$

- iLQR backward: $\mathcal{O}(n_{\mathrm{iLQR}} M D)$

- iLQR forward with back-tracking line search: $\mathcal{O}(n_{\mathrm{iLQR}} n_{\mathrm{ls}} T_0)$.

Thus, the final complexity of TOW is: $\mathcal{O}((n_{\mathrm{iLQR}} n_{\mathrm{ls}} + 1)T_0 + n_{\mathrm{iLQR}}(m_0 \eta + M)D))$ comparing to $\mathcal{O}(T_0)$ in the conventional meta-learning.

## Appendix F  Convergence analysis for trajectory optimisation based task weighting meta-learning

### F.1  Notations

The following notations are used throughout the paper:

- $\|\mathbf{x}\|$ is the L2 norm of a vector $\mathbf{x} \in \mathbb{R}^D$, e.g. $\sqrt{\mathbf{x}^\top \mathbf{x}}$

- $\|\mathbf{A}\|$ is the matrix norm of a matrix $\mathbf{A} \in \mathbb{R}^{M \times D}$ induced by a vector norm:

$$\|\mathbf{A}\| = \sup \frac{\|\mathbf{A}\mathbf{x}\|}{\|\mathbf{x}\|}, \forall \mathbf{x} \in \mathbf{R}^D : \|\mathbf{x}\| \neq 0.$$

Given the vector and matrix norm, two common inequalities used in this section are:

- Triangle inequality: $\|\mathbf{A} + \mathbf{B}\| \leq \|\mathbf{A}\| + \|\mathbf{B}\|, \forall \mathbf{A}, \mathbf{B} \in \mathbb{R}^{M \times D}$

- Sub-multiplication: $\|\mathbf{A}\mathbf{x}\| \leq \|\mathbf{A}\|\|\mathbf{x}\|$.

### F.2  Auxiliary lemmas

#### F.2.1  Boundedness of action (or weighting vector) $\mathbf{u}$

**Lemma 1.** *If $\mathbf{u}_t$ is a stationary action of a nominal action $\hat{\mathbf{u}}_t$ obtained from iLQR, then:*

$$\exists \delta > 0 : \|\mathbf{u}_t - \hat{\mathbf{u}}_t\| \leq \delta.$$

*Proof.* According to the procedure of iLQR shown in (39):

$$\mathbf{u}_t - \hat{\mathbf{u}}_t = \mathbf{K}_t \left( \mathbf{x}_t - \hat{\mathbf{x}}_t \right) + \varepsilon \mathbf{k}_t. \tag{58}$$

Since the matrix $\mathbf{K}$, vectors $\mathbf{k}$, $\mathbf{x}$ and $\hat{\mathbf{x}}$ are well-defined, the norm of $\mathbf{u} - \hat{\mathbf{u}}$ is also well-defined.

In addition, $\delta$ is implicitly related to the Gaussian prior $\mathcal{N}(\mathbf{u}_t; \mu_u \mathbf{1}, 1/\beta_u \mathbf{I}_M)$. A larger value of $\beta_u$ in (5) would result in a smaller value for $\delta$. □

**Corollary 9.** *If $\mathbf{u}_{t_1}$ and $\mathbf{u}_{t_2}$ are stationary actions of two nominal actions $\hat{\mathbf{u}}_{t_1} = \hat{\mathbf{u}}_{t_2} = \hat{\mathbf{u}}_{\mathrm{uniform}} = 1/M \mathbf{1}$ obtained from iLQR at time steps $t_1$ and $t_2$, respectively, then the followings hold:*

$$\|\mathbf{u}_{t_1} - \mathbf{u}_{t_2}\| \leq 2\delta \qquad \textit{(bounded L2 norm of difference between 2 actions)} \tag{59a}$$

$$\|\mathbf{u}_{t_1}\| \leq \delta + \frac{1}{\sqrt{M}} \qquad \textit{(bounded L2 norm)} \tag{59b}$$

$$\mathbf{1}^\top \mathbf{u}_{t_1} \leq \delta \sqrt{M} + 1. \qquad \textit{(bounded L1 norm)} \tag{59c}$$

*Proof.* The first inequality (59a) can be proved by simply applying triangle inequality on the $\ell_2$ norm and employing the result in Lemma 1:

$$\begin{aligned}
\|\mathbf{u}_{t_1} - \mathbf{u}_{t_2}\| &= \|(\mathbf{u}_{t_1} - \hat{\mathbf{u}}_{\mathrm{uniform}}) + (\hat{\mathbf{u}}_{\mathrm{uniform}} - \mathbf{u}_{t_2})\| \\
&\leq \|\mathbf{u}_{t_1} - \hat{\mathbf{u}}_{\mathrm{uniform}}\| + \|\mathbf{u}_{t_2} - \hat{\mathbf{u}}_{\mathrm{uniform}}\| \\
&\leq \|\mathbf{u}_{t_1} - \hat{\mathbf{u}}_{t_1}\| + \|\mathbf{u}_{t_2} - \hat{\mathbf{u}}_{t_2}\| \\
&\leq 2\delta.
\end{aligned} \tag{60}$$

The second inequality (59b) can similarly be proved using triangle inequality:

$$\|\mathbf{u}_{t_1}\| = \|(\mathbf{u}_{t_1} - \hat{\mathbf{u}}_{\mathrm{uniform}}) + \hat{\mathbf{u}}_{\mathrm{uniform}}\| \leq \|\mathbf{u}_{t_1} - \hat{\mathbf{u}}_{\mathrm{uniform}}\| + \|\hat{\mathbf{u}}_{\mathrm{uniform}}\| \leq \delta + \frac{1}{\sqrt{M}}. \tag{61}$$

The last inequality (59c) can be proved by the Cauchy-Schwarz inequality:

$$\mathbf{1}^\top \mathbf{u}_{t_1} \leq \left|\mathbf{1}^\top \mathbf{u}_{t_1}\right| \leq \sqrt{M}\|\mathbf{u}_{t_1}\| \quad \text{(Cauchy-Schwarz inequality)}$$
$$\leq \delta\sqrt{M} + 1. \quad \text{(due to Ineq. (59b))} \tag{62}$$

$\square$

### F.2.2  Bounded variances of the loss' gradient and the weighted validation loss

The bounded gradient norm in (6) (see Assumption 1) also implies that the variance of gradient of the loss function w.r.t. training samples is bounded as shown in Lemma 10.

**Lemma 10** (Bounded variance of loss' gradient). *If Assumption 1 holds, then the variance of gradient of the loss function is $\sigma^2$-bounded:*

$$\exists \sigma > 0 : \mathbb{E}_{(\mathbf{s}_{ij},y) \sim \mathcal{D}_i}\left[\left\|\boldsymbol{\nabla}_{\mathbf{x}}\ell(\mathbf{s}_{ij}, y_{ij}; \mathbf{x}) - \mathbb{E}_{(\mathbf{s}_{ij},y) \sim \mathcal{D}_i}[\boldsymbol{\nabla}_{\mathbf{x}}\ell(\mathbf{s}_{ij}, y_{ij}; \mathbf{x})]\right\|^2\right] \leq \sigma^2, \ \forall \mathbf{x} \in \mathbb{R}^D$$

*Proof.* The term inside the expectation in the left-hand side can be written as:

$$\left\|\boldsymbol{\nabla}_{\mathbf{x}}\ell(\mathbf{s}_{ij}, y_{ij}; \mathbf{x}) - \mathbb{E}_{(\mathbf{s}_{ij},y) \sim \mathcal{D}_i}[\boldsymbol{\nabla}_{\mathbf{x}}\ell(\mathbf{s}_{ij}, y_{ij}; \mathbf{x})]\right\|$$
$$\leq \|\boldsymbol{\nabla}_{\mathbf{x}}\ell(\mathbf{s}_{ij}, y_{ij}; \mathbf{x})\| + \left\|\mathbb{E}_{(\mathbf{s}_{ij},y) \sim \mathcal{D}_i}[\boldsymbol{\nabla}_{\mathbf{x}}\ell(\mathbf{s}_{ij}, y_{ij}; \mathbf{x})]\right\| \quad \text{(triangle inequality)}$$
$$\leq \|\boldsymbol{\nabla}_{\mathbf{x}}\ell(\mathbf{s}_{ij}, y_{ij}; \mathbf{x})\| + \mathbb{E}_{(\mathbf{s}_{ij},y) \sim \mathcal{D}_i}[\|\boldsymbol{\nabla}_{\mathbf{x}}\ell(\mathbf{s}_{ij}, y_{ij}; \mathbf{x})\|] \quad \text{(Jensen's inequality)}$$
$$\leq 2L \quad \text{(Boundedness of gradient in (6), which is proved in Lemma 14).} \tag{63}$$

Selecting $\sigma = 2L$ completes the proof. $\square$

The result in Lemma 10 also leads to the boundedness of the variance of the weighted validation loss as shown in Lemma 11.

**Lemma 11** (Bounded variance of weighted loss). *Given Lemma 10, the variance of $\boldsymbol{\nabla}_{\mathbf{x}}\left[\mathbf{u}_t^\top \mathsf{L}(\mathbf{x}_t)\right]$ is bounded above by $\widetilde{\sigma}^2 = \sigma^2 \left(\delta + M^{-0.5}\right)^2$.*

*Proof.* We use the following well-known inequality for variance as a part of the proof.

If $X_i, \forall i \in \{1, \ldots, n\}$, are random variables with finite variance: $\mathrm{Var}(X_i) < +\infty$, then:

$$\mathrm{Var}\left(\sum_{i=1}^n X_i\right) \leq n \sum_{i=1}^n \mathrm{Var}(X_i).$$

Note that the validation loss $\mathsf{L}_i(\mathbf{x}_t)$ defined in (3) is the empirical expected values of loss $\ell$ evaluated on task $i$-th. Hence, applying the above inequality for variance gives:

$$\mathrm{Var}(\boldsymbol{\nabla}_{\mathbf{x}}\mathsf{L}_i(\mathbf{x}_t)) = \mathrm{Var}\left(\frac{1}{m_q}\sum_{j=1}^{m_q}\boldsymbol{\nabla}_{\mathbf{x}}\ell\left(\mathbf{s}_{ij}^{(q)}, y_{ij}^{(q)}; \mathbf{x} - \frac{\gamma}{m_s}\sum_{k=1}^{m_s}\boldsymbol{\nabla}_{\mathbf{x}}\left[\ell\left(\mathbf{s}_{ik}^{(s)}, y_{ik}^{(s)}; \mathbf{x}_t\right)\right]\right)\right)$$
$$\leq \frac{1}{m_q}\sum_{j=1}^{m_q}\mathrm{Var}\left(\boldsymbol{\nabla}_{\mathbf{x}}\ell\left(\mathbf{s}_{ij}^{(q)}, y_{ij}^{(q)}; \mathbf{x} - \frac{\gamma}{m_s}\sum_{k=1}^{m_s}\boldsymbol{\nabla}_{\mathbf{x}}\left[\ell\left(\mathbf{s}_{ik}^{(s)}, y_{ik}^{(s)}; \mathbf{x}_t\right)\right]\right)\right)$$
$$\leq \sigma^2. \quad \text{(due to Lemma 10)} \tag{64}$$

The variance of the weighted validation loss $\boldsymbol{\nabla}_{\mathbf{x}}\mathbf{u}_t^\top \mathsf{L}(\mathbf{x}_t)$ can be written as:

$$\mathrm{Var}\left(\boldsymbol{\nabla}_{\mathbf{x}}\mathbf{u}_t^\top \mathsf{L}(\mathbf{x}_t)\right) = \mathrm{Var}\left(\sum_{i=1}^M \mathbf{u}_{ti}\boldsymbol{\nabla}_{\mathbf{x}}\mathsf{L}_i(\mathbf{x}_t)\right) = \sum_{i=1}^M \mathbf{u}_{ti}^2 \mathrm{Var}\left(\boldsymbol{\nabla}_{\mathbf{x}}\mathsf{L}_i(\mathbf{x}_t)\right)$$

$$\leq \sigma^2 \sum_{i=1}^M \mathbf{u}_{ti}^2 = \sigma^2\|\mathbf{u}_t\|^2 \quad \text{(due to the result in (64))}$$

$$\leq \sigma^2\left(\delta + M^{-0.5}\right)^2 \quad \text{(Corollary 9).} \tag{65}$$

$\square$

### F.2.3  Smoothness of validation loss

In this subsubsection, we prove Lemma 12 about the smoothness of validation loss. To make the subsubsection self-contained, we re-state the definition of the task-specific parameter $\phi_i(\mathbf{x})$ and the true validation loss $\bar{\mathsf{L}}_i(\mathbf{x})$ as follows:

$$\phi_i(\mathbf{x}) = \mathbf{x} - \frac{\gamma}{m_i^{(s)}} \sum_{k=1}^{m_i^{(s)}} \nabla_{\mathbf{x}}\left[\ell\left(\mathbf{s}_{ik}^{(s)}, y_{ik}^{(s)}; \mathbf{x}\right)\right] \tag{66}$$

$$\bar{\mathsf{L}}_i(\mathbf{x}) = \mathbb{E}_{\left(\mathbf{s}_{ij}^{(q)}, y_{ij}^{(q)}\right)\sim\mathcal{D}_i^{(q)}}\left[\ell\left(\mathbf{s}_{ij}^{(q)}, y_{ij}^{(q)}; \phi(\mathbf{x})\right)\right]. \tag{7}$$

Note that in (3), the task-specific parameter is defined as:

$$\phi_i^*(\mathbf{x}) \in \arg\min_{\phi_i} \frac{1}{m_i^{(s)}} \sum_{k=1}^{m_i^{(s)}}\left[\ell\left(\mathbf{s}_{ik}^{(s)}, y_{ik}^{(s)}; \phi_i(\mathbf{x})\right)\right].$$

Here, we approximate that lower-level optimisation by one step of gradient descent as shown in Eq. (66).

Lemma 12 and its proof are shown as follows:

**Lemma 12.** *If the conditions in Assumptions 1 to 3 hold, then the gradient of the true validation loss $\bar{\mathsf{L}}_i(\mathbf{x})$ defined in Eq. (7) is $\widetilde{S}$-Lipschitz, where: $\widetilde{S} = S(1+\gamma S)^2 + \gamma\rho L$.*

*Proof.* Before starting the proof, we exploit the notation of gradient of the loss function at a point $\mathbf{x} = \mathbf{v}$ as follows:

$$\boldsymbol{\nabla}_{\mathbf{x}}\ell(\mathbf{s}, y; \mathbf{v}) = \boldsymbol{\nabla}_{\mathbf{x}}\ell(\mathbf{s}, y; \mathbf{x})\bigg|_{\mathbf{x}=\mathbf{v}}. \tag{67}$$

Given the definition of the true validation loss in Eq. (7), its gradient w.r.t. $\mathbf{x}$ can be calculated using chain rule as follows:

$$\boldsymbol{\nabla}_{\mathbf{x}}\bar{\mathsf{L}}_i(\mathbf{x}) = \boldsymbol{\nabla}_{\mathbf{x}}\mathbb{E}_{\left(\mathbf{s}_{ij}^{(q)}, y_{ij}^{(q)}\right)\sim\mathcal{D}_i^{(q)}}\left[\ell\left(\mathbf{s}_{ij}^{(q)}, y_{ij}^{(q)}; \phi_i(\mathbf{x})\right)\right]$$

$$= \mathbb{E}_{\left(\mathbf{s}_{ij}^{(q)}, y_{ij}^{(q)}\right)\sim\mathcal{D}_i^{(q)}}\left[\boldsymbol{\nabla}_{\mathbf{x}}\phi_i(\mathbf{x}) \times \boldsymbol{\nabla}_{\mathbf{x}}\ell\left(\mathbf{s}_{ij}^{(q)}, y_{ij}^{(q)}; \phi_i(\mathbf{x})\right)\right]. \tag{68}$$

And since $\phi_i(\mathbf{x})$ obtained by gradient descent does not depends on validation (or query) samples, we can, therefore, rewrite the above gradient as:

$$\boldsymbol{\nabla}_{\mathbf{x}}\bar{\mathsf{L}}_i(\mathbf{x}) = \boldsymbol{\nabla}_{\mathbf{x}}\phi_i(\mathbf{x}) \times \mathbb{E}_{\left(\mathbf{s}_{ij}^{(q)}, y_{ij}^{(q)}\right)\sim\mathcal{D}_i^{(q)}}\left[\boldsymbol{\nabla}_{\mathbf{x}}\ell\left(\mathbf{s}_{ij}^{(q)}, y_{ij}^{(q)}; \phi_i(\mathbf{x})\right)\right]$$

$$= \left\{\mathbf{I} - \gamma\mathbb{E}_{\left(\mathbf{s}_{ij}^{(s)}, y_{ij}^{(s)}\right)\sim\mathcal{S}_i^{(s)}}\left[\boldsymbol{\nabla}_{\mathbf{x}}^2\ell\left(\mathbf{s}_{ik}^{(s)}, y_{ik}^{(s)}; \mathbf{x}\right)\right]\right\} \times \mathbb{E}_{\left(\mathbf{s}_{ij}^{(q)}, y_{ij}^{(q)}\right)\sim\mathcal{D}_i^{(q)}}\left[\boldsymbol{\nabla}_{\mathbf{x}}\ell\left(\mathbf{s}_{ij}^{(q)}, y_{ij}^{(q)}; \phi_i(\mathbf{x})\right)\right]. \tag{69}$$

Note that we abuse the notation and use $\mathbb{E}_{\left(\mathbf{s}_{ij}^{(s)}, y_{ij}^{(s)}\right) \sim \mathcal{S}_i^{(s)}}$ to indicate the average evaluated on all data points in set $\mathcal{S}_i^{(s)}$.

In the following, we omit sample $(\mathbf{s}, y)$ from the expectation to simplify the notations. In particular, the above gradient can be re-written as:

$$\boldsymbol{\nabla}_{\mathbf{x}} \bar{L}_i(\mathbf{x}) = \left\{\mathbf{I} - \gamma \mathbb{E}_{\mathcal{S}_i^{(s)}} \left[\boldsymbol{\nabla}_{\mathbf{x}}^2 \ell\left(\mathbf{s}_{ik}^{(s)}, y_{ik}^{(s)}; \mathbf{x}\right)\right]\right\} \mathbb{E}_{\mathcal{D}_i^{(q)}} \left[\boldsymbol{\nabla}_{\mathbf{x}} \ell\left(\mathbf{s}_{ij}^{(q)}, y_{ij}^{(q)}; \phi_i(\mathbf{x})\right)\right]. \tag{70}$$

Thus, we can calculate the difference of the gradient evaluated on the same task $\mathcal{T}_i$ but with two different meta-parameters:

$$
\begin{aligned}
&\left\|\boldsymbol{\nabla}_{\mathbf{x}} \bar{L}_i\left(\bar{\mathbf{x}}\right) - \boldsymbol{\nabla}_{\mathbf{x}} \bar{L}_i\left(\widetilde{\mathbf{x}}\right)\right\| \\
&= \left\|\left\{\mathbf{I} - \gamma \mathbb{E}_{\mathcal{S}_i^{(s)}} \left[\boldsymbol{\nabla}_{\mathbf{x}}^2 \ell\left(\mathbf{s}_{ik}^{(s)}, y_{ik}^{(s)}; \bar{\mathbf{x}}\right)\right]\right\} \mathbb{E}_{\mathcal{D}_i^{(q)}} \left[\boldsymbol{\nabla}_{\mathbf{x}} \ell\left(\mathbf{s}_{ik}^{(q)}, y_{ik}^{(q)}; \phi_i(\bar{\mathbf{x}})\right)\right]\right. \\
&\quad \left. - \left\{\mathbf{I} - \gamma \mathbb{E}_{\mathcal{S}_i^{(s)}} \left[\boldsymbol{\nabla}_{\mathbf{x}}^2 \ell\left(\mathbf{s}_{ik}^{(s)}, y_{ik}^{(s)}; \widetilde{\mathbf{x}}\right)\right]\right\} \mathbb{E}_{\mathcal{D}_i^{(q)}} \left[\boldsymbol{\nabla}_{\mathbf{x}} \ell\left(\mathbf{s}_{ik}^{(q)}, y_{ik}^{(q)}; \phi_i(\widetilde{\mathbf{x}})\right)\right]\right\| \\
&= \left\|\left\{\mathbf{I} - \gamma \mathbb{E}_{\mathcal{S}_i^{(s)}} \left[\boldsymbol{\nabla}_{\mathbf{x}}^2 \ell\left(\mathbf{s}_{ik}^{(s)}, y_{ik}^{(s)}; \bar{\mathbf{x}}\right)\right]\right\} \mathbb{E}_{\mathcal{D}_i^{(q)}} \left[\boldsymbol{\nabla}_{\mathbf{x}} \ell\left(\mathbf{s}_{ik}^{(q)}, y_{ik}^{(q)}; \phi_i(\bar{\mathbf{x}})\right)\right]\right. \\
&\quad - \left\{\mathbf{I} {\color{green}- \gamma \mathbb{E}_{\mathcal{S}_i^{(s)}} \left[\boldsymbol{\nabla}_{\mathbf{x}}^2 \ell\left(\mathbf{s}_{ik}^{(s)}, y_{ik}^{(s)}; \bar{\mathbf{x}}\right)\right] + \gamma \mathbb{E}_{\mathcal{S}_i^{(s)}} \left[\boldsymbol{\nabla}_{\mathbf{x}}^2 \ell\left(\mathbf{s}_{ik}^{(s)}, y_{ik}^{(s)}; \bar{\mathbf{x}}\right)\right]} \right. \\
&\quad \left. \left. - \gamma \mathbb{E}_{\mathcal{S}_i^{(s)}} \left[\boldsymbol{\nabla}_{\mathbf{x}}^2 \ell\left(\mathbf{s}_{ik}^{(s)}, y_{ik}^{(s)}; \widetilde{\mathbf{x}}\right)\right]\right\} \times \mathbb{E}_{\mathcal{D}_i^{(q)}} \left[\boldsymbol{\nabla}_{\mathbf{x}} \ell\left(\mathbf{s}_{ik}^{(q)}, y_{ik}^{(q)}; \phi_i(\widetilde{\mathbf{x}})\right)\right]\right\| \quad \text{(adding and substract the {\color{green}green terms})} \\
&= \left\|\left\{\mathbf{I} - \gamma \mathbb{E}_{\mathcal{S}_i^{(s)}} \left[\boldsymbol{\nabla}_{\mathbf{x}}^2 \ell\left(\mathbf{s}_{ik}^{(s)}, y_{ik}^{(s)}; \bar{\mathbf{x}}\right)\right]\right\} \mathbb{E}_{\mathcal{D}_i^{(q)}} \left[\boldsymbol{\nabla}_{\mathbf{x}} \ell\left(\mathbf{s}_{ik}^{(q)}, y_{ik}^{(q)}; \phi_i(\bar{\mathbf{x}})\right) - \boldsymbol{\nabla}_{\mathbf{x}} \ell\left(\mathbf{s}_{ik}^{(q)}, y_{ik}^{(q)}; \phi_i(\widetilde{\mathbf{x}})\right)\right]\right. \\
&\quad \left. - \gamma \mathbb{E}_{\mathcal{S}_i^{(s)}} \left[\boldsymbol{\nabla}_{\mathbf{x}}^2 \ell\left(\mathbf{s}_{ik}^{(s)}, y_{ik}^{(s)}; \bar{\mathbf{x}}\right) - \boldsymbol{\nabla}_{\mathbf{x}}^2 \ell\left(\mathbf{s}_{ik}^{(s)}, y_{ik}^{(s)}; \widetilde{\mathbf{x}}\right)\right] \mathbb{E}_{\mathcal{D}_i^{(q)}} \left[\boldsymbol{\nabla}_{\mathbf{x}} \ell\left(\mathbf{s}_{ik}^{(q)}, y_{ik}^{(q)}; \phi_i(\widetilde{\mathbf{x}})\right)\right]\right\|.
\end{aligned}
\tag{71}
$$

Applying the triangle inequality gives:

$$
\begin{aligned}
&\left\|\boldsymbol{\nabla}_{\mathbf{x}} \bar{L}_i\left(\bar{\mathbf{x}}\right) - \boldsymbol{\nabla}_{\mathbf{x}} \bar{L}_i\left(\widetilde{\mathbf{x}}\right)\right\| \\
&\leq \left\|\left\{\mathbf{I} - \gamma \mathbb{E}_{\mathcal{S}_i^{(s)}} \left[\boldsymbol{\nabla}_{\mathbf{x}}^2 \ell\left(\mathbf{s}_{ik}^{(s)}, y_{ik}^{(s)}; \bar{\mathbf{x}}\right)\right]\right\} \mathbb{E}_{\mathcal{D}_i^{(q)}} \left[\boldsymbol{\nabla}_{\mathbf{x}} \ell\left(\mathbf{s}_{ik}^{(q)}, y_{ik}^{(q)}; \phi_i(\bar{\mathbf{x}})\right) - \boldsymbol{\nabla}_{\mathbf{x}} \ell\left(\mathbf{s}_{ik}^{(q)}, y_{ik}^{(q)}; \phi_i(\widetilde{\mathbf{x}})\right)\right]\right\| \\
&\quad + \left\|\gamma \mathbb{E}_{\mathcal{S}_i^{(s)}} \left[\boldsymbol{\nabla}_{\mathbf{x}}^2 \ell\left(\mathbf{s}_{ik}^{(s)}, y_{ik}^{(s)}; \bar{\mathbf{x}}\right) - \boldsymbol{\nabla}_{\mathbf{x}}^2 \ell\left(\mathbf{s}_{ik}^{(s)}, y_{ik}^{(s)}; \widetilde{\mathbf{x}}\right)\right] \mathbb{E}_{\mathcal{D}_i^{(q)}} \left[\boldsymbol{\nabla}_{\mathbf{x}} \ell\left(\mathbf{s}_{ik}^{(q)}, y_{ik}^{(q)}; \phi_i(\widetilde{\mathbf{x}})\right)\right]\right\|.
\end{aligned}
\tag{72}
$$

Next, we upper-bound the two terms in the right-hand side of Ineq. (72). The first term can be upper-bounded as:

$$
\begin{aligned}
\text{First term} &= \left\|\left\{\mathbf{I} - \gamma \mathbb{E}_{\mathcal{S}_i^{(s)}} \left[\boldsymbol{\nabla}_{\mathbf{x}}^2 \ell\left(\mathbf{s}_{ik}^{(s)}, y_{ik}^{(s)}; \bar{\mathbf{x}}\right)\right]\right\} \mathbb{E}_{\mathcal{D}_i^{(q)}} \left[\boldsymbol{\nabla}_{\mathbf{x}} \ell\left(\mathbf{s}_{ik}^{(q)}, y_{ik}^{(q)}; \phi_i(\bar{\mathbf{x}})\right) - \boldsymbol{\nabla}_{\mathbf{x}} \ell\left(\mathbf{s}_{ik}^{(q)}, y_{ik}^{(q)}; \phi_i(\widetilde{\mathbf{x}})\right)\right]\right\| \\
&\leq \left\|\mathbf{I} - \gamma \mathbb{E}_{\mathcal{S}_i^{(s)}} \left[\boldsymbol{\nabla}_{\mathbf{x}}^2 \ell\left(\mathbf{s}_{ik}^{(s)}, y_{ik}^{(s)}; \bar{\mathbf{x}}\right)\right]\right\| \cdot \left\|\mathbb{E}_{\mathcal{D}_i^{(q)}} \left[\boldsymbol{\nabla}_{\mathbf{x}} \ell\left(\mathbf{s}_{ik}^{(q)}, y_{ik}^{(q)}; \phi_i(\bar{\mathbf{x}})\right) - \boldsymbol{\nabla}_{\mathbf{x}} \ell\left(\mathbf{s}_{ik}^{(q)}, y_{ik}^{(q)}; \phi_i(\widetilde{\mathbf{x}})\right)\right]\right\|
\end{aligned}
\tag{73}
$$

Applying Jensen's inequality on the L2 norm of the expectation in the right-hand side of the above inequality to bring the expectation outside of the L2 norm, then employing the smoothness of $\ell$ in Assumption 2 to obtain the following:

$$
\begin{aligned}
\text{First term} &\leq \left\|\mathbf{I} - \gamma \mathbb{E}_{\mathcal{S}_i^{(s)}} \left[\boldsymbol{\nabla}_{\mathbf{x}}^2 \ell\left(\mathbf{s}_{ik}^{(s)}, y_{ik}^{(s)}; \bar{\mathbf{x}}\right)\right]\right\| \times \mathbb{E}_{\mathcal{D}_i^{(q)}} \left\|\boldsymbol{\nabla}_{\mathbf{x}} \ell\left(\mathbf{s}_{ik}^{(q)}, y_{ik}^{(q)}; \phi_i(\bar{\mathbf{x}})\right) - \boldsymbol{\nabla}_{\mathbf{x}} \ell\left(\mathbf{s}_{ik}^{(q)}, y_{ik}^{(q)}; \phi_i(\widetilde{\mathbf{x}})\right)\right\| \\
&\leq \left\|\mathbf{I} - \gamma \mathbb{E}_{\mathcal{S}_i^{(s)}} \left[\boldsymbol{\nabla}_{\mathbf{x}}^2 \ell\left(\mathbf{s}_{ik}^{(s)}, y_{ik}^{(s)}; \bar{\mathbf{x}}\right)\right]\right\| \times S \left\|\phi_i(\bar{\mathbf{x}}) - \phi_i(\widetilde{\mathbf{x}})\right\|. \quad (\boldsymbol{\nabla}_{\mathbf{x}} \ell \text{ is } S\text{-Lipschitz – Assumption 2})
\end{aligned}
\tag{74}
$$

Given the definition of $\phi(\mathbf{x})$ in Eq. (66), we can obtain the following:

$$
\begin{aligned}
\|\phi_i(\bar{\mathbf{x}}) - \phi_i(\widetilde{\mathbf{x}})\| &= \left\| (\bar{\mathbf{x}} - \widetilde{\mathbf{x}}) - \gamma \mathbb{E}_{\mathcal{S}_i^{(s)}} \left[ \boldsymbol{\nabla}_{\mathbf{x}} \ell \left( \mathbf{s}_{ik}^{(s)}, y_{ij}^{(s)}; \bar{\mathbf{x}} \right) - \boldsymbol{\nabla}_{\mathbf{x}} \ell \left( \mathbf{s}_{ik}^{(s)}, y_{ij}^{(s)}; \widetilde{\mathbf{x}} \right) \right] \right\| \\
&\leq \|\bar{\mathbf{x}} - \widetilde{\mathbf{x}}\| + \gamma \left\| \mathbb{E}_{\mathcal{S}_i^{(s)}} \left[ \boldsymbol{\nabla}_{\mathbf{x}} \ell \left( \mathbf{s}_{ik}^{(s)}, y_{ij}^{(s)}; \bar{\mathbf{x}} \right) - \boldsymbol{\nabla}_{\mathbf{x}} \ell \left( \mathbf{s}_{ik}^{(s)}, y_{ij}^{(s)}; \widetilde{\mathbf{x}} \right) \right] \right\| \quad \text{(triangle inequality)} \\
&\leq \|\bar{\mathbf{x}} - \widetilde{\mathbf{x}}\| + \gamma \mathbb{E}_{\mathcal{S}_i^{(s)}} \left\| \boldsymbol{\nabla}_{\mathbf{x}} \ell \left( \mathbf{s}_{ik}^{(s)}, y_{ij}^{(s)}; \bar{\mathbf{x}} \right) - \boldsymbol{\nabla}_{\mathbf{x}} \ell \left( \mathbf{s}_{ik}^{(s)}, y_{ij}^{(s)}; \widetilde{\mathbf{x}} \right) \right\| \quad \text{(Jensen's inequality)} \\
&\leq \|\bar{\mathbf{x}} - \widetilde{\mathbf{x}}\| + \gamma S \|\bar{\mathbf{x}} - \widetilde{\mathbf{x}}\| \quad \text{(Assumption 2)}
\end{aligned}
\tag{75}
$$

Thus, one can upper-bound further Ineq. (74) as follows:

$$
\text{First term} \leq S(1 + \gamma S) \left\| \mathbf{I} - \gamma \mathbb{E}_{\mathcal{S}_i^{(s)}} \left[ \boldsymbol{\nabla}_{\mathbf{x}}^2 \ell \left( \mathbf{s}_{ik}^{(s)}, y_{ik}^{(s)}; \bar{\mathbf{x}} \right) \right] \right\| \cdot \|\bar{\mathbf{x}} - \widetilde{\mathbf{x}}\|.
\tag{76}
$$

If $\{\lambda_d\}_{d=1}^D$ are the eigenvalues of the Hessian matrix $\mathbb{E}_{\mathcal{S}_i^{(s)}} \left[ \boldsymbol{\nabla}_{\mathbf{x}}^2 \ell \left( \mathbf{s}_{ik}^{(s)}, y_{ik}^{(s)}; \bar{\mathbf{x}} \right) \right]$, then due to Lemma 13, the eigenvalues of $\mathbf{I} - \gamma \mathbb{E}_{\mathcal{S}_i^{(s)}} \left[ \boldsymbol{\nabla}_{\mathbf{x}}^2 \ell \left( \mathbf{s}_{ik}^{(s)}, y_{ik}^{(s)}; \bar{\mathbf{x}} \right) \right]$ are $\{1 - \gamma \lambda_d\}_{d=1}^D$. In addition, since $\mathbf{I} - \gamma \mathbb{E}_{\mathcal{S}_i^{(s)}} \left[ \boldsymbol{\nabla}_{\mathbf{x}}^2 \ell \left( \mathbf{s}_{ik}^{(s)}, y_{ik}^{(s)}; \bar{\mathbf{x}} \right) \right]$ is symmetric and positive semi-definite, its norm equals to the largest eigenvalue (refer to Lemma 15):

$$
\left\| \mathbf{I} - \gamma \mathbb{E}_{\mathcal{S}_i^{(s)}} \left[ \boldsymbol{\nabla}_{\mathbf{x}}^2 \ell \left( \mathbf{s}_{ik}^{(s)}, y_{ik}^{(s)}; \bar{\mathbf{x}} \right) \right] \right\| = \max_d |1 - \gamma \lambda_d| \leq 1 + \gamma \max_d |\lambda_d|.
\tag{77}
$$

According to Assumption 2 and the result in (Bubeck et al., 2015, section 3.2, page 266), the eigenvalues of the Hessian matrix $\mathbb{E}_{\mathcal{S}_i^{(s)}} \left[ \boldsymbol{\nabla}_{\mathbf{x}}^2 \ell \left( \mathbf{s}_{ik}^{(s)}, y_{ik}^{(s)}; \bar{\mathbf{x}} \right) \right]$ are smaller than $S$, implying:

$$
\left\| \mathbf{I} - \gamma \mathbb{E}_{\mathcal{S}_i^{(s)}} \left[ \boldsymbol{\nabla}_{\mathbf{x}}^2 \ell \left( \mathbf{s}_{ik}^{(s)}, y_{ik}^{(s)}; \bar{\mathbf{x}} \right) \right] \right\| \leq 1 + \gamma S.
\tag{78}
$$

Therefore, the first term on the right-hand side of (72) is upper-bounded by:

$$
\text{First term} \leq S (1 + \gamma S)^2 \|\bar{\mathbf{x}} - \widetilde{\mathbf{x}}\|.
\tag{79}
$$

The second term in the right-hand side of (72) can be upper-bounded as:

$$
\begin{aligned}
\text{Second term} &= \left\| \gamma \mathbb{E}_{\mathcal{S}_i^{(s)}} \left[ \boldsymbol{\nabla}_{\mathbf{x}}^2 \ell \left( \mathbf{s}_{ik}^{(s)}, y_{ik}^{(s)}; \bar{\mathbf{x}} \right) - \boldsymbol{\nabla}_{\mathbf{x}}^2 \ell \left( \mathbf{s}_{ik}^{(s)}, y_{ik}^{(s)}; \widetilde{\mathbf{x}} \right) \right] \mathbb{E}_{\mathcal{D}_i^{(q)}} \left[ \boldsymbol{\nabla}_{\mathbf{x}} \ell \left( \mathbf{s}_{ik}^{(q)}, y_{ik}^{(q)}; \phi_i(\widetilde{\mathbf{x}}) \right) \right] \right\| \\
&\leq \left\| \gamma \mathbb{E}_{\mathcal{S}_i^{(s)}} \left[ \boldsymbol{\nabla}_{\mathbf{x}}^2 \ell \left( \mathbf{s}_{ik}^{(s)}, y_{ik}^{(s)}; \bar{\mathbf{x}} \right) - \boldsymbol{\nabla}_{\mathbf{x}}^2 \ell \left( \mathbf{s}_{ik}^{(s)}, y_{ik}^{(s)}; \widetilde{\mathbf{x}} \right) \right] \right\| \cdot \left\| \mathbb{E}_{\mathcal{D}_i^{(q)}} \left[ \boldsymbol{\nabla}_{\mathbf{x}} \ell \left( \mathbf{s}_{ik}^{(q)}, y_{ik}^{(q)}; \phi_i(\widetilde{\mathbf{x}}) \right) \right] \right\| \\
&\leq \gamma \mathbb{E}_{\mathcal{S}_i^{(s)}} \left\| \boldsymbol{\nabla}_{\mathbf{x}}^2 \ell \left( \mathbf{s}_{ik}^{(s)}, y_{ik}^{(s)}; \bar{\mathbf{x}} \right) - \boldsymbol{\nabla}_{\mathbf{x}}^2 \ell \left( \mathbf{s}_{ik}^{(s)}, y_{ik}^{(s)}; \widetilde{\mathbf{x}} \right) \right\| \times \mathbb{E}_{\mathcal{D}_i^{(q)}} \left\| \boldsymbol{\nabla}_{\mathbf{x}} \ell \left( \mathbf{s}_{ik}^{(q)}, y_{ik}^{(q)}; \phi_i(\widetilde{\mathbf{x}}) \right) \right\| \\
&\quad \text{(Jensen's inequality)} \\
&\leq \gamma \rho L \|\bar{\mathbf{x}} - \widetilde{\mathbf{x}}\| \quad \text{(Eq. (6) and Assumption 3).}
\end{aligned}
\tag{80}
$$

Combining the results in (72), (79) and (80) gives:

$$
\left\| \boldsymbol{\nabla}_{\mathbf{x}} \bar{\mathsf{L}}_i (\bar{\mathbf{x}}) - \boldsymbol{\nabla}_{\mathbf{x}} \bar{\mathsf{L}}_i (\widetilde{\mathbf{x}}) \right\| \leq \left[ S(1 + \gamma S)^2 + \gamma \rho L \right] \|\bar{\mathbf{x}} - \widetilde{\mathbf{x}}\|.
\tag{81}
$$

This completes the proof. $\qquad\square$

### F.3 Convergence of TOW

**Theorem 2** (Main result). *If Assumptions 1 to 3 hold, the learning rate $\alpha < 2/\widetilde{S}(\delta\sqrt{M}+1)$, and $\mathbf{z}$ is randomly sampled from $\{\mathbf{x}_t\}_{t=1}^{T_{\mathrm{iter}}}$ returned by Algorithm 1, then:*

$$\mathbb{E}_{\mathbf{z}\sim\{\mathbf{x}_t\}_{t=1}^{T_{\mathrm{iter}}}}\left[\mathbb{E}_{\left(\mathcal{D}_{1:t}^{(q)}\right)^M}\left[\left\|\boldsymbol{\nabla}_{\mathbf{z}}\mathbf{u}_t^\top\bar{\mathsf{L}}_{1:M}\left(\mathbf{z}\right)\right\|^2\right]\right] \leq \epsilon_0 + \frac{\kappa}{T_{\mathrm{iter}}},$$

*where:*

$$\epsilon_0 = \frac{4\delta B\sqrt{M} + \alpha^2\widetilde{\sigma}^2\widetilde{S}\left(\delta\sqrt{M}+1\right)}{\alpha\left[2 - \alpha\widetilde{S}\left(\delta\sqrt{M}+1\right)\right]} > 0, \quad \kappa = \frac{2\mathbf{u}_1^\top\bar{\mathsf{L}}_{1:M}\left(\mathbf{x}_1\right)}{\alpha\left[2 - \alpha\widetilde{S}\left(\delta\sqrt{M}+1\right)\right]},$$

*with $T_{\mathrm{iter}}$ as the number of gradient-update for the meta-parameter, and $\mathbb{E}_{\left(\mathcal{D}_{1:t}^{(q)}\right)^M}$ as the expectation taken over all query data sampled from $t$ mini-batches.*

*Proof.* According to Eq. (7), $\bar{\mathsf{L}}_i(\mathbf{x}) \in \mathbb{R}$ is the expected validation loss of task $i$-th using meta-parameter $\mathbf{x}$. In addition, the notation $\left(\mathcal{D}_t^{(q)}\right)^M$ indicates the data probability that generates query data pairs for $M$ tasks in a mini-batch at time step $t$.

For convenience, we also denote $\bar{\mathsf{L}}(x) \in \mathbb{R}^M$ is the vector consisting of expected validation losses on $M$ tasks in a mini-batch of tasks:

$$\bar{\mathsf{L}}(x) = \begin{bmatrix} \bar{\mathsf{L}}_1(x) & \bar{\mathsf{L}}_2(x) & \dots & \bar{\mathsf{L}}_M(x) \end{bmatrix}^\top. \tag{82}$$

From Lemma 12, the gradient of the validation loss $\bar{\mathsf{L}}_i\left(\mathbf{x}\right)$ is $\widetilde{S}$-Lipschitz continuous. Hence, applying Taylor's theorem for the function $\bar{\mathsf{L}}_i(\mathbf{x}_{t+1})$ at point $\mathbf{x}_t$ gives:

$$\bar{\mathsf{L}}_i\left(\mathbf{x}_{t+1}\right) \leq \bar{\mathsf{L}}_i\left(\mathbf{x}_t\right) + \boldsymbol{\nabla}_{\mathbf{x}}^\top\bar{\mathsf{L}}_i\left(\mathbf{x}_t\right)\left(\mathbf{x}_{t+1} - \mathbf{x}_t\right) + \frac{\widetilde{S}}{2}\|\mathbf{x}_{t+1} - \mathbf{x}_t\|^2. \tag{83}$$

Note that $\mathbf{u}_{ti}$ is constrained to be non-negative as mentioned in Section 3.1 or in step 16 of Algorithm 1. Hence, one can multiply both sides by $\mathbf{u}_{ti} \geq 0, t \in \{1,\dots,T\}, i \in \{1,\dots,M\}$ and sum side-by-side to obtain:

$$\mathbf{u}_t^\top\bar{\mathsf{L}}\left(\mathbf{x}_{t+1}\right) \leq \mathbf{u}_t^\top\bar{\mathsf{L}}\left(\mathbf{x}_t\right) + \boldsymbol{\nabla}_{\mathbf{x}}^\top\left[\mathbf{u}_t\bar{\mathsf{L}}\left(\mathbf{x}_t\right)\right]\left(\mathbf{x}_{t+1} - \mathbf{x}_t\right) + \frac{\widetilde{S}}{2}\|\mathbf{x}_{t+1} - \mathbf{x}_t\|^2\mathbf{1}_M^\top\mathbf{u}_t. \tag{84}$$

For simplicity, we assume that the state-transition dynamics is SGD as mentioned in (5). Thus:

$$\mathbf{x}_{t+1} - \mathbf{x}_t = -\alpha\boldsymbol{\nabla}_{\mathbf{x}}\left[\mathbf{u}_t^\top\mathsf{L}\left(\mathbf{x}_t\right)\right]. \tag{85}$$

Thus, one can simplify to:

$$\mathbf{u}_t^\top\bar{\mathsf{L}}\left(\mathbf{x}_{t+1}\right) \leq \mathbf{u}_t^\top\bar{\mathsf{L}}\left(\mathbf{x}_t\right) - \alpha\boldsymbol{\nabla}_{\mathbf{x}}^\top\left[\mathbf{u}_t\bar{\mathsf{L}}\left(\mathbf{x}_t\right)\right]\boldsymbol{\nabla}_{\mathbf{x}}\left[\mathbf{u}_t^\top\mathsf{L}\left(\mathbf{x}_t\right)\right] + \frac{\widetilde{S}}{2}\left\|\boldsymbol{\nabla}_{\mathbf{x}}\left[\mathbf{u}_t^\top\mathsf{L}\left(\mathbf{x}_t\right)\right]\right\|^2\mathbf{1}_M^\top\mathbf{u}_t. \tag{86}$$

The above inequality contains the true validation loss $\bar{\mathsf{L}}(\mathbf{x})$ and the empirical validation loss $\mathsf{L}(\mathbf{x})$. The relation between them can be written as: $\bar{\mathsf{L}}(\mathbf{x}) = \mathbb{E}_{\left(\mathcal{D}^{(q)}\right)^M}\left[\mathsf{L}(\mathbf{x})\right]$. Thus, we take the expectation of validation data (expectation over the $t$-th mini-batch, each consists of $M$ tasks) on both sides to obtain:

$$\mathbb{E}_{\left(\mathcal{D}_t^{(q)}\right)^M}\left[\mathbf{u}_t^\top\bar{\mathsf{L}}\left(\mathbf{x}_{t+1}\right)\right] \leq \mathbb{E}_{\left(\mathcal{D}_t^{(q)}\right)^M}\left[\mathbf{u}_t^\top\bar{\mathsf{L}}\left(\mathbf{x}_t\right) - \alpha\boldsymbol{\nabla}_{\mathbf{x}}^\top\left[\mathbf{u}_t\bar{\mathsf{L}}\left(\mathbf{x}_t\right)\right]\boldsymbol{\nabla}_{\mathbf{x}}\left[\mathbf{u}_t^\top\mathsf{L}\left(\mathbf{x}_t\right)\right] + \frac{\widetilde{S}}{2}\left\|\boldsymbol{\nabla}_{\mathbf{x}}\left[\mathbf{u}_t^\top\mathsf{L}\left(\mathbf{x}_t\right)\right]\right\|^2\mathbf{1}_M^\top\mathbf{u}_t\right], \tag{87}$$

where $\left(\mathcal{D}_t^{(q)}\right)^M$ is the probability distribution of task data in the mini-batch of $M$ tasks at time step $t$.

Note that one can use Eq. (7) to imply that:

$$\boldsymbol{\nabla}_{\mathbf{x}}\left[\mathbf{u}_t\bar{\mathsf{L}}\left(\mathbf{x}_t\right)\right] = \mathbb{E}_{\left(\mathcal{D}_t^{(q)}\right)^M}\left[\boldsymbol{\nabla}_{\mathbf{x}}\left[\mathbf{u}_t^\top\mathsf{L}\left(\mathbf{x}_t\right)\right]\right],$$

which also implies:

$$\boldsymbol{\nabla}_{\mathbf{x}}\left[\mathbf{u}_t\bar{\mathsf{L}}\left(\mathbf{x}_t\right)\right] = \mathbb{E}_{\left(\mathcal{D}_t^{(q)}\right)^M}\left[\boldsymbol{\nabla}_{\mathbf{x}}\left[\mathbf{u}_t^\top\mathsf{L}\left(\mathbf{x}_t\right)\right]\right]. \tag{88}$$

Thus, the above inequality can be written as:

$$\mathbb{E}_{\left(\mathcal{D}_t^{(q)}\right)^M}\left[\mathbf{u}_t^\top\bar{\mathsf{L}}\left(\mathbf{x}_{t+1}\right)\right] \le \mathbf{u}_t^\top\bar{\mathsf{L}}\left(\mathbf{x}_t\right) - \alpha\left\|\boldsymbol{\nabla}_{\mathbf{x}}\left[\mathbf{u}_t\bar{\mathsf{L}}\left(\mathbf{x}_t\right)\right]\right\|^2$$

$$+ \frac{\alpha^2\widetilde{S}}{2}\left[\mathrm{Var}\left[\boldsymbol{\nabla}_{\mathbf{x}}\left[\mathbf{u}_t^\top\mathsf{L}\left(\mathbf{x}_t\right)\right]\right] + \left\|\boldsymbol{\nabla}_{\mathbf{x}}\left[\mathbf{u}_t\bar{\mathsf{L}}\left(\mathbf{x}_t\right)\right]\right\|^2\right]\mathbf{1}_M^\top\mathbf{u}_t$$

$$(\text{since }\mathbb{E}[X^2] = \mathrm{Var}[X] + (\mathbb{E}[X])^2\ )$$

$$\le \mathbf{u}_t^\top\bar{\mathsf{L}}\left(\mathbf{x}_t\right) - \alpha\left(1 - \frac{\alpha\widetilde{S}}{2}\mathbf{1}_M^\top\mathbf{u}_t\right)\left\|\boldsymbol{\nabla}_{\mathbf{x}}\left[\mathbf{u}_t\bar{\mathsf{L}}\left(\mathbf{x}_t\right)\right]\right\|^2$$

$$+ \frac{\alpha^2\widetilde{S}}{2}\mathrm{Var}\left[\boldsymbol{\nabla}_{\mathbf{x}}\left[\mathbf{u}_t^\top\mathsf{L}\left(\mathbf{x}_t\right)\right]\right]\mathbf{1}_M^\top\mathbf{u}_t. \tag{89}$$

Since:

$$\begin{cases}\mathbf{1}_M^\top\mathbf{u}_t & \le \delta\sqrt{M} + 1 \quad (\text{Corollary 9 in Appendix F.2.1})\\ \mathrm{Var}\left[\boldsymbol{\nabla}_{\mathbf{x}}\left[\mathbf{u}_t^\top\mathsf{L}\left(\mathbf{x}_t\right)\right]\right] & \le \widetilde{\sigma}^2 \qquad\quad (\text{Lemma 11})\end{cases}$$

then:

$$\mathbb{E}_{\left(\mathcal{D}_t^{(q)}\right)^M}\left[\mathbf{u}_t^\top\bar{\mathsf{L}}\left(\mathbf{x}_{t+1}\right)\right] \le \mathbf{u}_t^\top\bar{\mathsf{L}}\left(\mathbf{x}_t\right) - \alpha\left[1 - \frac{\alpha\widetilde{S}}{2}\left(\delta\sqrt{M} + 1\right)\right]\left\|\boldsymbol{\nabla}_{\mathbf{x}}\left[\mathbf{u}_t\bar{\mathsf{L}}\left(\mathbf{x}_t\right)\right]\right\|^2 + \frac{\alpha^2\widetilde{\sigma}^2\widetilde{S}}{2}\left(\delta\sqrt{M} + 1\right). \tag{90}$$

Re-arranging the gradient norm of the weighted validation loss to the left-hand side gives:

$$\alpha\left[1 - \frac{\alpha\widetilde{S}}{2}\left(\delta\sqrt{M} + 1\right)\right]\left\|\boldsymbol{\nabla}_{\mathbf{x}}\left[\mathbf{u}_t\bar{\mathsf{L}}\left(\mathbf{x}_t\right)\right]\right\|^2 \le \mathbf{u}_t^\top\bar{\mathsf{L}}\left(\mathbf{x}_t\right) - \mathbb{E}_{\left(\mathcal{D}_t^{(q)}\right)^M}\left[\mathbf{u}_t^\top\bar{\mathsf{L}}\left(\mathbf{x}_{t+1}\right)\right] + \frac{\alpha^2\widetilde{\sigma}^2\widetilde{S}}{2}\left(\delta\sqrt{M} + 1\right). \tag{91}$$

The right-hand-side, excluding the constant term at the end, can be written as:

$$\mathbf{u}_t^\top\bar{\mathsf{L}}\left(\mathbf{x}_t\right) - \mathbb{E}_{\left(\mathcal{D}_t^{(q)}\right)^M}\left[\mathbf{u}_t^\top\bar{\mathsf{L}}\left(\mathbf{x}_{t+1}\right)\right]$$

$$= \left[\mathbf{u}_t^\top\bar{\mathsf{L}}\left(\mathbf{x}_t\right) - \mathbb{E}_{\left(\mathcal{D}_t^{(q)}\right)^M}\left[\mathbf{u}_{t+1}^\top\bar{\mathsf{L}}\left(\mathbf{x}_{t+1}\right)\right]\right] + \left[\mathbb{E}_{\left(\mathcal{D}_t^{(q)}\right)^M}\left[\mathbf{u}_{t+1}^\top\bar{\mathsf{L}}\left(\mathbf{x}_{t+1}\right)\right] - \mathbb{E}_{\left(\mathcal{D}_t^{(q)}\right)^M}\left[\mathbf{u}_t^\top\bar{\mathsf{L}}\left(\mathbf{x}_{t+1}\right)\right]\right] \tag{92}$$

The second part in the right-hand side of the above expression can be upper-bounded as:

$$\mathbb{E}_{\left(\mathcal{D}_t^{(q)}\right)^M}\left[\mathbf{u}_{t+1}^\top\bar{\mathsf{L}}\left(\mathbf{x}_{t+1}\right)\right] - \mathbb{E}_{\left(\mathcal{D}_t^{(q)}\right)^M}\left[\mathbf{u}_t^\top\bar{\mathsf{L}}\left(\mathbf{x}_{t+1}\right)\right]$$

$$= \mathbb{E}_{\left(\mathcal{D}_t^{(q)}\right)^M}\left[\left(\mathbf{u}_{t+1} - \mathbf{u}_t\right)^\top\bar{\mathsf{L}}\left(\mathbf{x}_{t+1}\right)\right]$$

$$\le \mathbb{E}_{\left(\mathcal{D}_t^{(q)}\right)^M}\left[\left\|\mathbf{u}_{t+1} - \mathbf{u}_t\right\|\sqrt{\sum_{i=1}^M\bar{\mathsf{L}}_i^2\left(\mathbf{x}_{t+1}\right)}\right] \quad (\text{Cauchy-Schwarz inequality})$$

$$\le B\sqrt{M}\,\mathbb{E}_{\left(\mathcal{D}_t^{(q)}\right)^M}\left\|\mathbf{u}_{t+1} - \mathbf{u}_t\right\| \quad (\ell \text{ is } B\text{-bounded, and hence, } \bar{\mathsf{L}}_i \text{ is } B\text{-bounded})$$

$$\le 2\delta B\sqrt{M} \quad (\text{Corollary 9}). \tag{93}$$

Hence, one can upper-bound further the right-hand side of (91), resulting in:

$$\alpha\left[1-\frac{\alpha\widetilde{S}}{2}\left(\delta\sqrt{M}+1\right)\right]\left\|\boldsymbol{\nabla}_{\mathbf{x}}\left[\mathbf{u}_t\bar{\mathsf{L}}\left(\mathbf{x}_t\right)\right]\right\|^2 \leq \mathbf{u}_t^{\top}\bar{\mathsf{L}}\left(\mathbf{x}_t\right)-\mathbb{E}_{\left(\mathcal{D}_t^{(q)}\right)^M}\left[\mathbf{u}_{t+1}^{\top}\bar{\mathsf{L}}\left(\mathbf{x}_{t+1}\right)\right]+2\delta B\sqrt{M}$$
$$+\frac{\alpha^2\widetilde{\sigma}^2\widetilde{S}}{2}\left(\delta\sqrt{M}+1\right). \tag{94}$$

We take the expectation over all the mini-batches used at time step $t, t-1, \ldots, 1$ on both sides of Ineq. (94) to obtain:

$$\alpha\left[1-\frac{\alpha\widetilde{S}}{2}\left(\delta\sqrt{M}+1\right)\right]\mathbb{E}_{\left(\mathcal{D}_{1:t}^{(q)}\right)^M}\left[\left\|\boldsymbol{\nabla}_{\mathbf{x}}\left[\mathbf{u}_t\bar{\mathsf{L}}\left(\mathbf{x}_t\right)\right]\right\|^2\right]$$
$$\leq \mathbb{E}_{\left(\mathcal{D}_{1:t}^{(q)}\right)^M}\left[\mathbf{u}_t^{\top}\bar{\mathsf{L}}\left(\mathbf{x}_t\right)\right]-\mathbb{E}_{\left(\mathcal{D}_{1:t}^{(q)}\right)^M}\left[\mathbf{u}_{t+1}^{\top}\bar{\mathsf{L}}\left(\mathbf{x}_{t+1}\right)\big|\mathbf{x}_1\right]+2\delta B\sqrt{M}+\frac{\alpha^2\widetilde{\sigma}^2\widetilde{S}}{2}\left(\delta\sqrt{M}+1\right), \tag{95}$$

where: $\mathbb{E}_{\left(\mathcal{D}_{1:t}^{(q)}\right)^M}[.]=\mathbb{E}_{\left(\mathcal{D}_1^{(q)}\right)^M}\mathbb{E}_{\left(\mathcal{D}_2^{(q)}\right)^M}\ldots\mathbb{E}_{\left(\mathcal{D}_t^{(q)}\right)^M}[.]$.

Summing (95) from $t=1$ to $T_{\text{iter}}$ gives:

$$\alpha\left[1-\frac{\alpha\widetilde{S}}{2}\left(\delta\sqrt{M}+1\right)\right]\sum_{t=1}^{T_{\text{iter}}}\mathbb{E}_{\left(\mathcal{D}_{1:t}^{(q)}\right)^M}\left[\left\|\boldsymbol{\nabla}_{\mathbf{x}}\left[\mathbf{u}_t\bar{\mathsf{L}}\left(\mathbf{x}_t\right)\right]\right\|^2\right]$$
$$\leq \mathbf{u}_1^{\top}\bar{\mathsf{L}}\left(\mathbf{x}_1\right)-\mathbb{E}_{\left(\mathcal{D}_{1:T_{\text{iter}}}^{(q)}\right)^M}\left[\mathbf{u}_{T_{\text{iter}}+1}^{\top}\bar{\mathsf{L}}\left(\mathbf{x}_{T_{\text{iter}}+1}\right)\right]+T_{\text{iter}}\left[2\delta B\sqrt{M}+\frac{\alpha^2\widetilde{\sigma}^2\widetilde{S}}{2}\left(\delta\sqrt{M}+1\right)\right]. \tag{96}$$

Note that $\mathbf{u}_{ti}\geq 0 \;\forall t\in\{1,\ldots,T_{\text{iter}}\}, i\in\{1,\ldots,M\}$, and $\bar{\mathsf{L}}_i\geq 0$ due to the non-negativity of the loss function $\ell$ assumed in (3). Hence, we can upper-bound further the right-hand side to:

$$\alpha\left[1-\frac{\alpha\widetilde{S}}{2}\left(\delta\sqrt{M}+1\right)\right]\sum_{t=1}^{T_{\text{iter}}}\mathbb{E}_{\left(\mathcal{D}_{1:t}^{(q)}\right)^M}\left[\left\|\boldsymbol{\nabla}_{\mathbf{x}}\left[\mathbf{u}_t\bar{\mathsf{L}}\left(\mathbf{x}_t\right)\right]\right\|^2\right]\leq \mathbf{u}_1^{\top}\bar{\mathsf{L}}\left(\mathbf{x}_1\right)+T_{\text{iter}}\left[2\delta B\sqrt{M}+\frac{\alpha^2\widetilde{\sigma}^2\widetilde{S}}{2}\left(\delta\sqrt{M}+1\right)\right]. \tag{97}$$

If the learning rate $\alpha$ is selected such that: $1-\alpha\widetilde{S}/2\left(\delta\sqrt{M}+1\right)>0$, then dividing both sides by $\alpha T_{\text{iter}}\left[1-\alpha\widetilde{S}/2\times\left(\delta\sqrt{M}+1\right)\right]$ gives:

$$\frac{1}{T_{\text{iter}}}\sum_{t=1}^{T_{\text{iter}}}\mathbb{E}_{\left(\mathcal{D}_{1:t}^{(q)}\right)^M}\left[\left\|\boldsymbol{\nabla}_{\mathbf{x}}\left[\mathbf{u}_t\bar{\mathsf{L}}\left(\mathbf{x}_t\right)\right]\right\|^2\right]\leq\frac{2\mathbf{u}_1^{\top}\bar{\mathsf{L}}\left(\mathbf{x}_1\right)+T_{\text{iter}}\left[4\delta B\sqrt{M}+\alpha^2\widetilde{\sigma}^2\widetilde{S}\left(\delta\sqrt{M}+1\right)\right]}{\alpha T_{\text{iter}}\left[2-\alpha\widetilde{S}\left(\delta\sqrt{M}+1\right)\right]}. \tag{98}$$

Next, we use a similar trick in the SVRG paper (Johnson & Zhang, 2013) to make the left-hand side term useful. The idea is to output some randomly chosen $\mathbf{x}_t$ rather than outputing $\mathbf{x}_{T_{\text{iter}}}$. For simplicity, let $\mathbf{z}=\mathbf{x}_t$ with a uniform probability for $t\in\{1,\ldots,T_{\text{iter}}\}$. The expectation of the gradient norm in this case can be expressed as:

$$\frac{1}{T_{\text{iter}}}\sum_{t=1}^{T_{\text{iter}}}\mathbb{E}_{\left(\mathcal{D}_{1:t}^{(q)}\right)^M}\left[\left\|\boldsymbol{\nabla}_{\mathbf{x}}\left[\mathbf{u}_t\bar{\mathsf{L}}\left(\mathbf{x}_t\right)\right]\right\|^2\right]=\mathbb{E}_{\mathbf{z}\sim\{\mathbf{x}_t\}_{t=1}^{T_{\text{iter}}}}\left[\mathbb{E}_{\left(\mathcal{D}_{1:t}^{(q)}\right)^M}\left[\left\|\boldsymbol{\nabla}_{\mathbf{x}}\left[\mathbf{u}_t\bar{\mathsf{L}}\left(\mathbf{x}_t\right)\right]\right\|^2\right]\right]. \tag{99}$$

This combining with (98) leads to:

$$\mathbb{E}_{\mathbf{z}\sim\{\mathbf{x}_t\}_{t=1}^{T_{\text{iter}}}}\left[\mathbb{E}_{\left(\mathcal{D}_{1:t}^{(q)}\right)^M}\left[\left\|\boldsymbol{\nabla}_{\mathbf{z}}\left[\mathbf{u}_t\bar{\mathsf{L}}\left(\mathbf{z}\right)\right]\right\|^2\right]\right]\leq\frac{2\mathbf{u}_1^{\top}\bar{\mathsf{L}}\left(\mathbf{x}_1\right)+T_{\text{iter}}\left[4\delta B\sqrt{M}+\alpha^2\widetilde{\sigma}^2\widetilde{S}\left(\delta\sqrt{M}+1\right)\right]}{\alpha T_{\text{iter}}\left[2-\alpha\widetilde{S}\left(\delta\sqrt{M}+1\right)\right]}, \tag{100}$$

which concludes the proof. $\qquad\square$

### F.4 Miscellaneous lemmas

**Lemma 13.** *If $\lambda$ is an eigenvalue of matrix $\mathbf{A}$, then $\lambda - 1$ is an eigenvalue of matrix $\mathbf{A} - \mathbf{I}$, where $\mathbf{I}$ is the identity matrix.*

*Proof.* According to the definition of eigenvalue, $\lambda$ is an eigenvalue of $\mathbf{A}$ if:

$$\det\left(\mathbf{A} - \lambda \mathbf{I}\right) = 0.$$

Rewriting the above equation gives:

$$\det\left((\mathbf{A} - \mathbf{I}) - (\lambda - 1)\mathbf{I}\right) = 0. \tag{101}$$

Hence, $\lambda - 1$ is an eigenvalue of $\mathbf{A} - \mathbf{I}$. $\qquad\square$

**Lemma 14** (Adpated from https://math.stackexchange.com/a/4303207/274798)**.**
*If a function $f : \mathbb{R}^n \to \mathbb{R}^m$, where $n, m \in \mathbb{N}$, is differentiable and $L$-Lipschitz, then its gradient norm is bounded by $L$.*

*Proof.* According to the definition of vector or matrix norm:

$$
\begin{aligned}
\|\boldsymbol{\nabla} f(x)\| &= \sup_v \frac{\|\boldsymbol{\nabla} f(x)\,v\|}{\|v\|}, \forall x, v \in \mathbb{R}^n : \|v\| \neq 0 \\
&= \sup_v \lim_{\lambda \to 0} \frac{|\lambda|\|\boldsymbol{\nabla} f(x)\,v\|}{|\lambda|\|v\|}, \ \lambda \in \mathbb{R}, \lambda \neq 0 \\
&= \sup_v \lim_{\lambda \to 0} \frac{\|\boldsymbol{\nabla} f(x)\,(\lambda v)\|}{\|\lambda v\|}.
\end{aligned} \tag{102}
$$

Applying the triangle inequality gives:

$$
\begin{aligned}
\|\boldsymbol{\nabla} f(x)\| &\leq \sup_v \lim_{\lambda \to 0} \frac{\|f(x + \lambda v) - f(x) - \boldsymbol{\nabla} f(x)\,(\lambda v)\|}{\|\lambda v\|} + \frac{\|f(x + \lambda v) - f(x)\|}{\|\lambda v\|} \\
&\leq \sup_v \lim_{\lambda \to 0} \frac{\|f(x + \lambda v) - f(x) - \boldsymbol{\nabla} f(x)\,(\lambda v)\|}{\|\lambda v\|} + \frac{L\|x + \lambda v - x\|}{\|\lambda v\|} \ (f \text{ is } L\text{-Lipschitz}) \\
&\leq L.
\end{aligned} \tag{103}
$$

The first term equals to 0 since it corresponds to the definition of Fréchet derivative. $\qquad\square$

**Lemma 15** (Adapted from https://math.stackexchange.com/a/2193914/274798)**.**
*If $A \in \mathbb{R}^{n \times n}$ is a positive definite symmetric matrix, then its largest eigenvalue is*

$$\lambda_{\max} = \max\left\{\frac{\|A\,x\|}{\|x\|} : x \neq 0\right\}. \tag{104}$$

*Proof.* According to the spectral theorem, if $A$ is a real and symmetric matrix, then there exists an orthonormal basis consisting of eigenvectors of $A$, denoted as $v_1, v_2, \dots, v_n$. Without loss of generality, let's assume that the corresponding eigenvalues are: $\lambda_1, \lambda_2, \dots, \lambda_n$.

For any non-zero vector $x$, we can represent it in this orthonormal basis as:

$$x = \sum_{i=1}^{n} c_i v_i, \ c_i \in \mathbb{R}. \tag{105}$$

Then, the function of interest can be calculated as follows:

$$
\begin{aligned}
\frac{\|A\,x\|}{\|x\|} &= \frac{\|\sum_{i=1}^{n} c_i \lambda_i v_i\|}{\sqrt{\sum_{i=1}^{n} c_i^2}} \\
&= \sqrt{\frac{\sum_{i=1}^{n} c_i^2 \lambda_i^2}{\sum_{i=1}^{n} c_i^2}} \quad (\{v_i\}_{i=1}^{n} \text{ is an orthonormal basis}) \\
&\leq \max_{i \in \{1,\ldots,n\}} \lambda_i.
\end{aligned}
\tag{106}
$$

The equality occurs when the vector $x$ equals to the corresponding eigenvector of the eigenvalue $\max_{i \in \{1,\ldots,n\}} \lambda_i$. $\qquad\square$

## Appendix G   Experimental settings

For all experiments, GD is used to obtain $\phi^*(\mathbf{x})$ with 5 iterations and a learning rate of 0.1 for Omniglot and 0.01 for mini-ImageNet. The learning rate for the meta-parameters, $\alpha$, is set at $10^{-4}$ for all the setting. The mini-batch size is $M = 10$ tasks for Omniglot and $M = 5$ tasks for mini-ImageNet. For the Dirichlet concentration of the prior in the *exploration* and *exploitation* baselines, we try three values of $\kappa \in \{0.2, 1.2, 5\}$, and found that a too small value of $\kappa$ leads to noisy learning since only the easiest or hardest task is selected, while too large value of $\kappa$ makes both the baselines identical to uniform weighting. Hence, we select $\kappa = 1.2$ that balances between these two strategies. Note that $\kappa = 1$ results in a random prior, leading to a trivial solution. For the trajectory optimiser iLQR, the state-transition dynamics $f$ follows the formula of Adam optimiser since Adam provides a less noisy training as in SGD. The nominal trajectory is, as mentioned in Section 3.1, selected with uniform actions: $\hat{\mathbf{u}}_{tj} = 1/M, \forall j \in \{1, \ldots, M\}, t \in \{1, \ldots, T\}$. The number of iterations used in iLQR is 2 to speed up the training, although higher number of iterations can be used to achieve better performance by trading off the running time. We also provide an ablation study with two different numbers of iterations in iLQR in Section 5.4, where the larger number of iterations in iLQR slightly improves the prediction accuracy on the validation set of mini-ImageNet. The number of time steps (or number of mini-batches) is $T = 10$ for Omniglot and 5 for mini-ImageNet. The parameters of the prior on the action $\mathbf{u}_t$ are $\mu_u = 1/M$ and $\beta_u = 10$. As we do not observe any major difference between different configuration of $M$ and $T$ used in this experiment, we report the result for the case $M = 10$ and $T = 5$. Each experiment is carried out on a single NVIDIA Tesla V100 GPU with 32 GB memory following the configuration of NVIDIA DGX-1. The experiments mentioned in Section 5.4 is performed on a single NIVIDIA A100 GPU with 40 GB memory.

## Appendix H    Additional results when calculating with full matrix $\mathbf{V}_t$

In this Appendix, we provide additional results of prediction accuracy on tasks formed from the evaluation sets of Omniglot and mini-ImageNet. These results are based on the naive implementation of iLQR where the auxiliary Hessian matrix $\mathbf{V}_t$ of the *cost-to-go* in (28) is exact without any approximation. We note that due to the quadratic complexity of running time, we cannot train the one on mini-ImageNet until convergence. The running time using the full matrix $\mathbf{V}_t$ is 160 GPU-hour for Omniglot and 184 GPU-hour for mini-ImageNet.

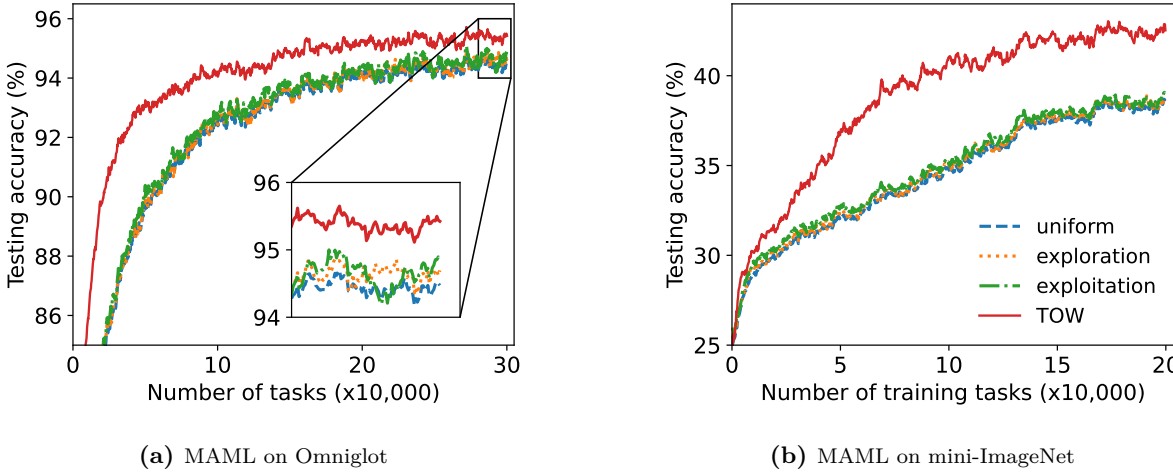

**(a)** MAML on Omniglot      **(b)** MAML on mini-ImageNet

**Figure 7:** Additional results of prediction accuracy on 100 random validation tasks using MAML when the matrix $\mathbf{V}_t$ is exact without any approximation.

# Appendix I  Examples of loss functions satisfying Assumptions 1 to 3

In this section, we analyse some common loss functions including mean square error (MSE) and cross-entropy (CE) to see if they satisfy Assumptions 1 to 3 made in Section 4.

**Table 5:** Examples of some common loss functions used with the assumption on the linearity in Eq. (107) that satisfy Assumptions 1 to 3 specified in Section 4.

| | **Loss** (Assumption 1) | | **Lipschitz gradient** | **Lipschitz Hessian** |
|---|---|---|---|---|
| | Bounded | Lipschitz | (Assumption 2) | (Assumption 3) |
| MSE | $\checkmark^4$ | $\checkmark^5$ | $\checkmark$ | $\checkmark$ |
| CE | | $\checkmark$ | $\checkmark$ | $\checkmark$ |

Before analysing the loss functions of interest, it is important to note that the objective function mostly depends on the model used. For example, if $g(\mathbf{s}; \mathbf{x})$ denotes the output of a model parameterised by $\mathbf{x}$ for the input $\mathbf{s}$, then the objective function of interest is $\ell(g(\mathbf{s}; \mathbf{x}), \mathbf{y})$. Depending on the model $g$ used, the objective function is different and would be very complicated if $g$ represents a deep neural network since it would result in a high-dimensional, non-linear, non-convex function. Analysing such general function is, however, beyond the scope of this paper. To simplify, we assume that $g$ is a linear function of $\mathbf{x}$:

$$g(\mathbf{s}; \mathbf{x}) = \mathbf{S}\mathbf{x}. \tag{107}$$

**Boundedness in Assumption 1**  both the loss functions of interest, MSE and CE, are theoretically unbounded, and thus, does not satisfy this assumption. However, we can simply clip the loss to ensure the satisfaction of the boundedness assumption.

## I.1  Mean square error

The loss function is defined as:

$$\ell(\mathbf{x}) = \|\mathbf{S}\mathbf{x} - \mathbf{y}\|_2^2. \tag{108}$$

**Lipschitz continuity**  In general, MSE does not satisfy the Lipschitz continuity property. However, when the vector norm of $\mathbf{x}$ is bounded: $\|\mathbf{x}\| \leq X_0$, then MSE is Lipschitz-continuous.

*Proof.* The Lipschitz continuity means:

$$\left| \|\mathbf{S}\mathbf{x}_1 - \mathbf{y}\|_2^2 - \|\mathbf{S}\mathbf{x}_2 - \mathbf{y}\|_2^2 \right| \leq \text{const.} \|\mathbf{x}_1 - \mathbf{x}_2\|. \tag{109}$$

---

[4]when the loss is clipped

[5]when the norm of parameter vector is bounded

The left hand side term can be expanded as follows:

$$
\begin{aligned}
&\left\| \|\mathbf{S}\mathbf{x}_1 - \mathbf{y}\|_2^2 - \|\mathbf{S}\mathbf{x}_2 - \mathbf{y}\|_2^2 \right\| \\
&= \left\| \mathbf{x}_1^\top \mathbf{S}^\top \mathbf{S}\mathbf{x}_1 - 2\left(\mathbf{S}\mathbf{x}_1\right)^\top \mathbf{y} - \mathbf{x}_2^\top \mathbf{S}^\top \mathbf{S}\mathbf{x}_2 + 2\left(\mathbf{S}\mathbf{x}_2\right)^\top \mathbf{y} \right\| \\
&= \left\| \mathbf{x}_1^\top \mathbf{S}^\top \mathbf{S}\mathbf{x}_1 - \mathbf{x}_2^\top \mathbf{S}^\top \mathbf{S}\mathbf{x}_2 - 2(\mathbf{x}_1 - \mathbf{x}_2)^\top \mathbf{S}^\top \mathbf{y} \right\| \\
&= \left\| (\mathbf{x}_1 - \mathbf{x}_2)^\top \mathbf{S}^\top \mathbf{S}\mathbf{x}_1 + \mathbf{x}_2^\top \mathbf{S}^\top \mathbf{S}(\mathbf{x}_1 - \mathbf{x}_2) - 2(\mathbf{x}_1 - \mathbf{x}_2)^\top \mathbf{S}^\top \mathbf{y} \right\| \\
&\quad \text{(triangle inequality)} \\
&\leq \left\| (\mathbf{x}_1 - \mathbf{x}_2)^\top \mathbf{S}^\top \mathbf{S}\mathbf{x}_1 \right\| + \left\| \mathbf{x}_2^\top \mathbf{S}^\top \mathbf{S}(\mathbf{x}_1 - \mathbf{x}_2) \right\| + 2\left\| (\mathbf{x}_1 - \mathbf{x}_2)^\top \mathbf{S}^\top \mathbf{y} \right\| \\
&\quad \text{(submultiplicative inequality)} \\
&\leq \left( \left\| \mathbf{S}^\top \mathbf{S}\mathbf{x}_1 \right\| + \left\| \mathbf{x}_2^\top \mathbf{S}^\top \mathbf{S} \right\| + 2\left\| \mathbf{S}^\top \mathbf{y} \right\| \right) \|\mathbf{x}_1 - \mathbf{x}_2\| \\
&\leq \left( \left\| \mathbf{S}^\top \mathbf{S} \right\|\|\mathbf{x}_1\| + \left\| \mathbf{S}^\top \mathbf{S} \right\|\|\mathbf{x}_2\| + 2\left\| \mathbf{S}^\top \mathbf{y} \right\| \right) \|\mathbf{x}_1 - \mathbf{x}_2\| \\
&\leq 2\left( X_0 \left\| \mathbf{S}^\top \mathbf{S} \right\| + \left\| \mathbf{S}^\top \mathbf{y} \right\| \right) \|\mathbf{x}_1 - \mathbf{x}_2\|.
\end{aligned}
\tag{110}
$$

where the norm is the operator norm if the argument is a matrix. $\qquad\square$

**Smoothness**  MSE is smooth, or its gradient is Lipschitz-continuous.

*Proof.* The gradient of MSE can be written as:

$$
\boldsymbol{\nabla}\ell(\mathbf{x}) = 2\mathbf{S}^\top\left(\mathbf{S}\mathbf{x} - \mathbf{y}\right).
\tag{111}
$$

Therefore, for any $\mathbf{x}_1$ and $\mathbf{x}_2$:

$$
\begin{aligned}
\|\boldsymbol{\nabla}\ell(\mathbf{x}_1) - \boldsymbol{\nabla}\ell(\mathbf{x}_2)\| &= 2\left\| \mathbf{S}^\top\left(\mathbf{S}\mathbf{x}_1 - \mathbf{y}\right) - \mathbf{S}^\top\left(\mathbf{S}\mathbf{x}_2 - \mathbf{y}\right) \right\| \\
&= 2\left\| \mathbf{S}^\top \mathbf{S}(\mathbf{x}_1 - \mathbf{x}_2) \right\| \\
&\leq 2\left\| \mathbf{S}^\top \mathbf{S} \right\|\|\mathbf{x}_1 - \mathbf{x}_2\|.
\end{aligned}
\tag{112}
$$

Thus, MSE in this setting is Lipschitz-continuous. $\qquad\square$

**Lipschitz-continuous Hessian**  Since the Hessian in this case is constant, it also satisfies the Lipschitz-continuity.

### I.2 Cross-entropy loss

The loss function can be defined as:

$$
\ell(\mathbf{x}) = -\mathbf{y}^\top \ln\left[\text{softmax}\left(\mathbf{S}\mathbf{x}\right)\right],
\tag{113}
$$

where:

$$
\text{softmax}\left(\mathbf{S}\mathbf{x}\right) = \frac{\exp\left(\mathbf{S}\mathbf{x}\right)}{\mathbf{1}^\top \exp\left(\mathbf{S}\mathbf{x}\right)}
\tag{114}
$$

is the softmax function.

**Lipschitz continuity**  The cross-entropy loss is Lipschitz-continuous.

*Proof.* To prove, we employ the mean-value theorem and then find the upper-bound of the gradient norm.

First, we calculate the gradient of the softmax function w.r.t. each element $x_i$ in $\mathbf{x}$:

$$
\begin{aligned}
\frac{\partial \mathrm{softmax}\left(\mathbf{Sx}\right)_j}{\partial x_i} &= \frac{\partial}{\partial x_i} \frac{\exp\left(\mathbf{Sx}\right)_j}{\mathbf{1}^\top \exp\left(\mathbf{Sx}\right)} \\
&= \frac{\partial}{\partial (\mathbf{Sx})_j} \frac{\exp\left(\mathbf{Sx}\right)_j}{\mathbf{1}^\top \exp\left(\mathbf{Sx}\right)} \times \frac{\partial(\mathbf{Sx})_j}{\partial x_i} \\
&= \mathrm{softmax}(\mathbf{Sx})_j \left[\mathbb{1}(j=i) - \mathrm{softmax}(\mathbf{Sx})_i\right] \mathbf{S}_{ij},
\end{aligned}
\tag{115}
$$

where the subscript denotes an element in the corresponding vector and $\mathbb{1}(.)$ denotes the indicator function.

Thus, the derivative of the cross-entropy loss w.r.t. the parameter of interest $\mathbf{x}$ can be written following the chain rule:

$$
\begin{aligned}
\frac{\partial \ell(\mathbf{x})}{\partial x_i} &= -\frac{\partial}{\partial x_i} \sum_{j=1}^{n_c} \mathbf{y}_j \ln\left[\mathrm{softmax}\left(\mathbf{Sx}\right)_j\right] \\
&= -\sum_{j=1}^{n_c} \frac{\mathbf{y}_j}{\mathrm{softmax}(\mathbf{Sx})_j} \times \frac{\partial \mathrm{softmax}(\mathbf{Sx})_j}{\partial x_i} \\
&= \sum_{j=1}^{n_c} \mathbf{y}_j \left[\mathbb{1}(j=i) - \mathrm{softmax}(\mathbf{Sx})_i\right] \mathbf{S}_{ij},
\end{aligned}
\tag{116}
$$

where $n_c$ is the number of classes.

Therefore, one can apply the triangle inequality to obtain the following:

$$
\begin{aligned}
\left\|\frac{\partial \ell(\mathbf{x})}{\partial x_i}\right\| &\leq \sum_{j=1}^{n_c} \left\|\mathbf{y}_j \left[\mathbb{1}(j=i) - \mathrm{softmax}(\mathbf{Sx})_i\right] \mathbf{S}_{ij}\right\| \\
&\leq \sum_{j=1}^{n_c} \underbrace{\|\mathbf{y}_j\|}_{\leq 1} \times \underbrace{\left\|\left[\mathbb{1}(j=i) - \mathrm{softmax}(\mathbf{Sx})_i\right]\right\|}_{\leq 2} \times \|\mathbf{S}_{ij}\| \\
&\leq 2 \sum_{j=1}^{n_c} \|\mathbf{S}_{ij}\|.
\end{aligned}
\tag{117}
$$

Since each element of the gradient is bounded, the gradient norm is also bounded.

By applying the mean-value theorem, we can obtain the following:

$$
\|\ell(\mathbf{x}_1) - \ell(\mathbf{x}_2)\| = \sup_{\mathbf{z} \in [\mathbf{x}_1, \mathbf{x}_2]} \|\boldsymbol{\nabla}_{\mathbf{z}} \ell(\mathbf{z})\| \times \|\mathbf{x}_1 - \mathbf{x}_2\|,
\tag{118}
$$

where: $\mathbf{z} \in [\mathbf{x}_1, \mathbf{x}_2]$ denotes a vector z contained in the set of points between $\mathbf{x}_1, \mathbf{x}_2 \in \mathbb{R}^D$.

Since the gradient norm is bounded, the mean-value theorem results in the Lipschitz continuity for the cross-entropy loss. $\qquad\square$

**Smoothness**    The cross-entropy loss in this case is smooth, or in other words, its gradient is Lipschitz-continuous.

*Proof.* According to (116), the difference of derivative between $\mathbf{x}_1$ and $\mathbf{x}_2$ can be written as:

$$
\begin{aligned}
\frac{\partial \ell(\mathbf{x}_1)}{\partial x_i} - \frac{\partial \ell(\mathbf{x}_2)}{\partial x_i} &= \sum_{j=1}^{n_c} \mathbf{y}_j \left[\mathrm{softmax}(\mathbf{Sx}_2)_i - \mathrm{softmax}(\mathbf{Sx}_1)_i\right] \mathbf{S}_{ij} \\
&= \left[\mathrm{softmax}(\mathbf{Sx}_2)_i - \mathrm{softmax}(\mathbf{Sx}_1)_i\right] \sum_{j=1}^{n_c} \mathbf{y}_j \mathbf{S}_{ij}.
\end{aligned}
\tag{119}
$$

Therefore:

$$\|\boldsymbol{\nabla}_{\mathbf{x}}\ell(\mathbf{x}_1) - \boldsymbol{\nabla}_{\mathbf{x}}\ell(\mathbf{x}_2)\| = \sqrt{\sum_{i=1}^{D}\left\{\left[\text{softmax}(\mathbf{S}\mathbf{x}_2)_i - \text{softmax}(\mathbf{S}\mathbf{x}_1)_i\right]\sum_{j=1}^{n_c}\mathbf{y}_j\mathbf{S}_{ij}\right\}^2}$$

$$\leq \max_{i\in\{1,\dots,D\}}\left|\sum_{j=1}^{n_c}\mathbf{y}_j\mathbf{S}_{ij}\right| \times \sqrt{\sum_{i=1}^{D}\left[\text{softmax}(\mathbf{S}\mathbf{x}_2)_i - \text{softmax}(\mathbf{S}\mathbf{x}_1)_i\right]^2}$$

$$\leq \max_{i\in\{1,\dots,D\}}\left|\sum_{j=1}^{n_c}\mathbf{y}_j\mathbf{S}_{ij}\right| \times \|\text{softmax}(\mathbf{S}\mathbf{x}_1) - \text{softmax}(\mathbf{S}\mathbf{x}_2)\|. \tag{120}$$

And since softmax function is Lipschitz-continuous[6], the gradient of cross-entropy loss, in this case, is also Lipschitz-continuous. $\qquad\square$

**Lipschitz continous Hessian**    The Hessian matrix of the cross-entropy loss is Lipschitz-continuous.

*Proof.* To prove the Lipschitz continuity of the Hessian matrix for the cross-entropy loss, we will prove that the norm of its third order derivative tensor is bounded. According to (Nesterov, 2003, Page 175), if the norm of the third order tensor, also known as *Terssian*, is bounded, then the Hessian matrix will satisfies the Lipschitz continuity. To do this, we calculate each element in the Terssian tensor as follows:

The second derivative of the cross-entropy loss w.r.t. an element $x_i$ can be written as:

$$\frac{\partial^2\ell(\mathbf{x})}{\partial x_i\partial x_k} = \frac{\partial}{\partial x_k}\sum_{j=1}^{n_c}\mathbf{y}_j\left[\mathbb{1}(j=i) - \text{softmax}(\mathbf{S}\mathbf{x})_i\right]\mathbf{S}_{ij}$$

$$= -\frac{\partial\text{softmax}(\mathbf{S}\mathbf{x})_i}{\partial x_k} \times \sum_{j=1}^{n_c}\mathbf{y}_j\mathbf{S}_{ij}$$

$$= -\text{softmax}(\mathbf{S}\mathbf{x})_k\left[\mathbb{1}(k=i) - \text{softmax}(\mathbf{S}\mathbf{x})_i\right]\mathbf{S}_{ik} \times \sum_{j=1}^{n_c}\mathbf{y}_j\mathbf{S}_{ij}. \tag{121}$$

Thus, the third order derivative can also be derived as:

$$\frac{\partial^3\ell(\mathbf{x})}{\partial x_i\partial x_k\partial x_l} = -\mathbf{S}_{ik} \times \left(\sum_{j=1}^{n_c}\mathbf{y}_j\mathbf{S}_{ij}\right) \times \frac{\partial}{\partial x_l}\left[\text{softmax}(\mathbf{S}\mathbf{x})_k\left[\mathbb{1}(k=i) - \text{softmax}(\mathbf{S}\mathbf{x})_i\right]\right]. \tag{122}$$

To this point, we can see that the third order derivative involves the derivative of softmax functions and some constants relating to input data $\mathbf{S}$ and its label $\mathbf{y}$. Also, as the derivative of softmax function shown in (115) consists of softmax and indicator functions, the derivative of softmax function is bounded. Thus, we can conclude that the third order derivative in (122) is also bounded with the bound depending on the norm of the input $\max_{i,j}|\mathbf{S}_{ij}|$ and label $\max_j|\mathbf{y}_j|$.

When the norm of each element in the Terssian is bounded, its norm is also bounded. Thus, according to the result in (Nesterov, 2003, Page 175), we can conclude that the Hessian is Lipschitz-continuous. $\qquad\square$

### I.3    Values of the Lipschitz continuity constants

The values of the constants used in Assumptions 1 to 3 heavily depend on some other parameters, including the maximum value of norm of the inputs and labels as well as the clipping norm value of the parameter $\mathbf{x}$

---

[6]see Proposition 4 in "On the properties of the softmax function with application in game theory and reinforcement learning", arXiv preprint arXiv:1704.00805.

(in case of MSE). Thus, we cannot give exact values in numbers for those constants, but only provide a few expressions to show the dependency of their values on the input and output.

**Table 6:** The values of some Lipschitz-continuity constants assumed in Assumptions 1 to 3.

| Loss function | $L$ | $S$ |
|---|:---:|:---:|
| MSE | $\sup_{\mathbf{S},\mathbf{y}} 2\left(X_0\lVert\mathbf{S}^\top\mathbf{S}\rVert + \lVert\mathbf{S}^\top\mathbf{y}\rVert\right)$ | $\sup_{\mathbf{S}} 2\lVert\mathbf{S}^\top\mathbf{S}\rVert$ |
| CE | $\sup_{\mathbf{S}} 2\lVert\mathbf{S}\rVert_1$ | - |

## Appendix J   Additional visualisation

We provide additional visualisation for the qualitative results reported in Section 5 with uncertainty in the evaluation. Conventionally, meta-learning methods are evaluated on a certain number of tasks where the accuracy reported is averaged over those testing tasks (Vinyals et al., 2016; Finn et al., 2017; Ravi & Larochelle, 2017; Snell et al., 2017; Ravi & Beatson, 2019; Nguyen et al., 2020). In this case, the uncertainty, represented by the confidence intervals, is calculated based on the accuracy values obtained on those testing tasks. In Section 5, we do not plot this uncertainty associated with the empirical results, e.g., Figures 1 and 2, to clarify the visualisation. Hence, we provide additional qualitative results with 95 percent confidence intervals in Figure 8, where Figure 8a is the accuracy on 100 validation 5-way 1-shot tasks formed from mini-ImageNet, and Figure 8b is the "zoom-in" version.

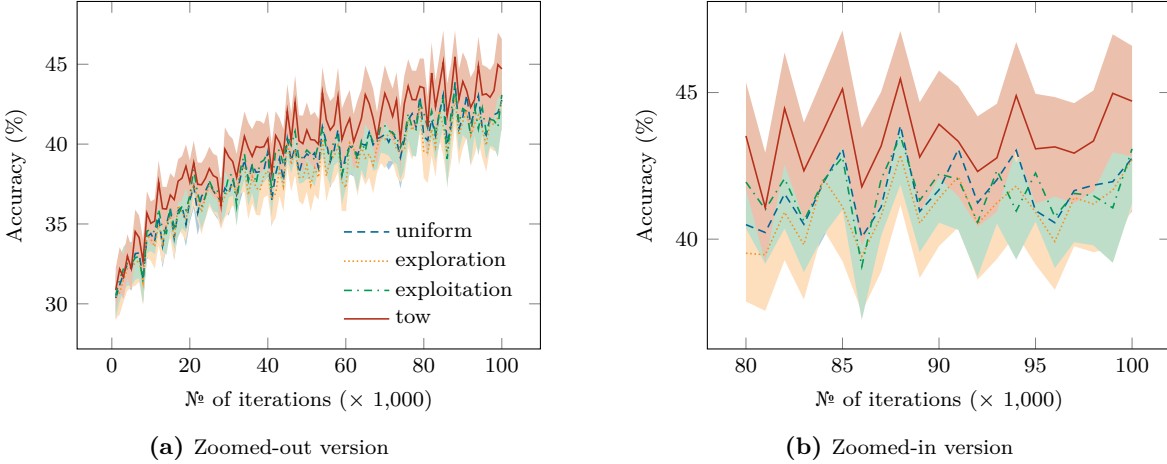

**(a)** Zoomed-out version          **(b)** Zoomed-in version

**Figure 8:** Additional visualisation of the prediction accuracy evaluated on 100 5-way 1-shot mini-ImageNet with 95 percent confidence intervals.

