# OpenReview forum: "Task Weighting in Meta-learning with Trajectory Optimisation"
_TMLR — Accepted by TMLR_

### Review · Reviewer_YmYi · 2023-05-29

**Summary Of Contributions:**

The paper proposes to use iLQR as part of an optimization algorithm for meta-learning. They frame the iterates of the outer-loop optimization problem as the states, the action space as weights on different sampled tasks in each minibatch, with the dynamics of the system as gradient descent on the weighted loss induced by the selected action and cost given by the unweighted loss over tasks.

The authors show improved results on several few shot learning benchmarks compared to not using any task weighting or simple heuristic task weighing schemes.They also provide convergence proofs for the introduced algorithm, as well as for iLQR, which apparently did not appear in prior works introducing it.

**Audience:**

Yes

**Broader Impact Concerns:**

no concerns

**Claims And Evidence:**

Yes

**Requested Changes:**

To major concerns regarding acceptance of the paper
- I do not see any analysis on the horizon $T$ of the trajectory optimization problem in the paper, and appears to be arbitrarily chosen to be 5 or 10 based on appendix G. Naturally, the proposed algorithm would have no impact with the trivial case of horizon 1, and it would be important to see how the quality benefits materialize as we increase the horizon. What would happen if we treat our *entire* training set of tasks as one trajectory optimized with iLQR?
- As mentioned above in the weaknesses section, comparisons against the baselines with extra compute. How does the method compare if we simply run through the minibatches with the baseline algorithm multiple times to compensate for the drastically increased compute costs?

Some other suggested changes to strengthen the work.
- For clarity: I would suggest introducing the structure of the proposed TOW algorithm earlier, in the end of the introduction. I think it would be very helpful to explicitly state early on we consider a sequence of meta-parameter iterates as a trajectory, and we optimize the average loss over the iterates as the cost.


**Strengths And Weaknesses:**

Strengths:
- The paper proposes an interesting connection by optimizing a "hyperparameter" like task weights via optimal control, and to my knowledge, the use of iLQR within the optimization procedure like this is an interesting novel application. It does not appear to me that the proposed trajectory optimization is specific to the metalearning setup with task weighting action; it could be interesting to consider applying this idea to generic optimization problems where the action can be any other hyperparameter for learning (for example, weighting of individual data examples or scaling factors for different coordinates of the learning rate).
- The paper shows promising improvements in benchmarks (albeit at the cost of much increased compute) and proof of convergence is provided under some typical assumptions.

Weaknesses:
- The iLQR loop seems quite computationally expensive to run, needing to reevaluate the trajectory of iterates many times for each update. Authors show that runtime for the proposed method can be 7-9x slower than the compared baselines. I suggest expanding the comparisons to include variants of the baselines with additional training (with the same # of tasks) to analyze how much benefit the optimized task reweighting over naively increasing the computation used to comparable amounts.
- I did find the presentation of the method a bit difficult to follow, in particular trying to understand where exactly the trajectory optimization algorithm fit in. In particular, it was not immediately clear that there were now three nested levels of iteration, one outer loop, one for the iLQR iterations, and one for the inner loop of the metalearning algorithm.

---

> ### Author Response · Authors · 2023-06-25
> **Analyse the effect of the horizon $T$**
>
> As the proposed method, TOW, is based on iLQR, it shares some of the properties of iLQR. According to the iLQR paper (Tassa et al., 2012, Section IV. A, the value of the horizon $T$ is a problem-dependent quantity which must be found by trial-and-error. The optimal value of $T$ is, therefore, chosen by further fine-tuning. However, as our aim is to connect the trajectory optimisation and task-weighting in meta-learning and analyse the convergence of the proposed method theoretically, we select the value of $T$ specified in the paper for demonstration purposes.
>
> In this revision, we run an ablation study where we vary the value of $T$ and show in Figure 4b in section 5.4.2 of the revised paper. Similar to the study in the iLQR paper, increasing $T$ might not always result in a better performance. It might be due to the approximation nature of iLQR where the transition dynamics $f$ and the cost function $c$ are linearised and quadraticised via Taylor series. Such approximations are, however, only accurate for a short trajectory (small $T$). For a long trajectory, the future states predicted by iLQR would be deviated further away, making the optimisation sub-optimal. Even if we assume that we have enough computational power to treat the entire training set of tasks as one trajectory and optimise that by iLQR, the result might not be better than a short trajectory. It is also worth noting that if the training set has a large number of classes, it would be intractable to optimise for all tasks, as explained in the section Related Work.

---

> ### Author Response · Authors · 2023-06-25
> **Comparisons against the baselines with extra compute**
>
> Thanks for suggesting this experiment.
>
> To provide a fairer comparison, instead of performing one iteration for each baseline, we train each of the baselines using five more mini-batches of tasks. Such a larger number of mini-batches require a reduction of the learning rate to enable a fair comparison, hence, we reduce the learning rate of the baselines by a factor of five. We note that the overhead caused by the much longer training of the baselines only allowed us to run this experiment given the short time frame of the discussion period. Nevertheless, the empirical results in Figure 6 show that TOW still outperforms the baselines on this new setting.

---

> ### Author Response · Authors · 2023-06-26
> **Introduce TOW earlier in the introduction**
>
> We thank the reviewer for the suggestion. We add a brief description about the proposed method at the end of the Section Introduction in Magenta colour to faciliate the understanding about our paper.

---

### Review · Reviewer_YUvV · 2023-05-31

**Summary Of Contributions:**

The authors propose the novel approach TOW for optimizing task weighting in the outer-loop of meta-learning based on a trajectory optimization approach. For this, they cast the task-weighting problem to a finite-horizon discrete-time trajectory optimisation and apply iterative linear quadratic regulator for optimizing task weighting. They also prove that under certain conditions the procedure converges. They show benefits of TOW compared to simpler baselines on few-shot classification on OmniGlot and Mini-ImageNet.

**Audience:**

Yes

**Broader Impact Concerns:**

There are no concerns from the reviewer

**Claims And Evidence:**

Yes

**Requested Changes:**

 - Having the related works section between methods and experiment sections is suboptimal as it disconnects these two parts. I would recommend to have related works either after the introduction or before the conclusion.
 - Section 3.1 is somewhat lengthy and not easy to digest. I think splitting it into subsubsection could help. Also, on page 5 it is not always clear which problems the authors want to address in the respective paragraphs.
 - Equation 5 has two hyperparameters ($\beta$ and $\mu_u$). How important is tuning these parameters in terms of preventing overfitting? It would be helpful to study this empirically.



**Strengths And Weaknesses:**

Strengths:
 - The paper motivates the problem setting well and provides the necessary background in Section 2.
 - The authors provide a novel approach (TOW) for solving a non-trivial problem (optimizing task weightings in meta-learning)
 - The authors provide a solid theoretical convergence analysis of the proposed procedure
 - The benefit of TOW over simpler baselines is convincingly demonstrated in the experiments.
 - The authors also discuss limitations of ROW, such as increased computational runtime.

Weaknesses:
 - Evaluation is limited to few-shot classification on relatively simple problems (OmniGlot, Mini-Imagenet). Evaluating also on other meta-learning problems would strengthen the empirical evidence.
 - TOW is not tested in conjunction with state-of-the-art meta-learning methods, like e.g. Bayesian TAML (Lee et al., 2020), but only with MAML and Protonet.
 - Runtime of TOW is relatively high (7-9x slower than baselines). It is not clear if this is a constant factor or increases with size of network of number of tasks. An analysis on the asymptotic behaviour of TOW in terms of large networks or large number of tasks would be helpful.

---

> ### Author Response · Authors · 2023-06-25
> **Evaluate on additional settings and other meta-learning methods**
>
> We agree that evaluating the proposed method on other meta-learning datasets with different  meta-learning baselines would help to strengthen the empirical evidence of the proposed TOW. However, please recall that the main goals of this paper are:
> 1. propose a new method that connects trajectory optimisation and meta-learning, and
> 2. introduce a new theoretical analysis of the convergence or our proposed TOW when integrating it into a base meta-learning method.
>
> Therefore, our paper aims to present a theoretical proof of such a concept rather than introducing a method that can empirically achieve the state-of-the-art results on a variety of meta-learning benchmarks. As we empirically demonstrate in the paper, our method can improve the performance of base meta-learning methods. Hence, we believe that the  evaluation currently present in the paper shows a reliable empirical analysis of the capability of TOW, providing enough evidence to support our claims. Another issue with the evaluation on more datasets using more base meta-learning methods lies in the need for a substantial amount of computational resources to produce many results that are unlikely to enable us to deduce much more beyond what we have in the paper. Instead of testing on additional benchmarks and methods, our future work is to focus on the scalability and efficiency of TOW to make it more practical.

---

> ### Author Response · Authors · 2023-06-25
> **Running time complexity of the proposed method TOW**
>
> We note that, given the setting specified in the paper, any meta-learning method is carried out following a stochastic approach where the meta-parameter is updated after every mini-batch of tasks. Thus, the complexity of meta-learning methods does not explicitly depend on the number of tasks, but only on the size of the mini-batch.
>
> In the paper, we explain the complexity of the proposed method at the end of Section 3.2 with details presented in Appendix E. We repeat the complexity analysis here to address the concern raised by the review. If a base meta-learning method (MAML or Proto-net) is assumed to have a complexity of $\mathcal{O}(T_{0})$, then the proposed method has a complexity of ${\mathcal{O}((n_{\mathrm{iLQR}} n_{\mathrm{ls}} + 1)T_{0} + n_{\mathrm{iLQR}} (m_{0} \eta + M) D)})$, where:
>
> - $\mathcal{O}(T_{0})$ is the time complexity to train a meta-learning method following a uniform weighting strategy,
> - $D$ is the dimension of meta-parameter $\mathbf{x}$ (or network size),
> - $M$ is the number of tasks within a \say{meta} mini-batch,
> - ${m_{0} = \max_{i \in \{1, \ldots, M\}} m_{i}^{(q)}}$ is the total number of validation samples within a task,
> - $n_{\mathrm{iLQR}}$ is the number of iterations used in iLQR to calculate the re-weighting vector $\mathbf{u}$,
> - $n_{\mathrm{ls}}$ is the number of iterations performing line-search,
> - $\eta$ is the number of arithmetic operations in the model of interest.
>
> In principle, $T_{0}$ is a function of $D$ and the type of function varies depending on which meta-learning method used. If we assume that $T_{0}$ is a linear function of $D$, for example: $T_{0} = \mathrm{const.} D$, then:
>
> - Conventional meta-learning: $\mathcal{O}(\mathrm{const.} D)$
> - TOW: $\mathcal{O}(((n_{\mathrm{iLQR}} n_{\mathrm{ls}} + 1) \mathrm{const.} + n_{\mathrm{iLQR}} (m_{0} \eta + M))D)$.
>
> Thus, TOW depends on the network size (the value of $D$) and will have a constant overhead compared to its corresponding base meta-learning method.

---

> ### Author Response · Authors · 2023-06-25
> **Ablation studies for the two hyper-parameters $\beta_{u}$ and $\mu_{u}$**
>
> The hyper-parameters $\beta_{u}$ and $\mu_{u}$ represent our belief or prior about the weighting vector $\mathbf{u}$. To this end, we carry out additional ablation studies to understand the influence of these hyper-parameters to the performance of TOW.
>
> For $\beta_{u}$, we carry out an experiment with $\beta_{u} = 1$ and $\beta_{u} = 10$, while freezing other hyper-parameters, and show the result in Figure 5a. Empirically, TOW performs slightly better for a larger $\beta_{u}$.
>
> For $\mu_{u}$, we conduct an experiment with three different values: 0.05, 0.2 (corresponding to 1/M or uniform weighting) and 0.5. The result in Figure 5b shows that the signal is not very clear since the performance of TOW is quite similar across three different values of $\mu_{u}$. Future work might need to investigate further to understand more about the effect of $\mu_{u}$.

---

> ### Author Response · Authors · 2023-06-26
> **Change paper structure to increase the readability and clarity**
>
> We thank the reviewer for the suggestions. We have made some changes in our revision as follows:
>
> - We move the Related Work to right before the Discussion and Conclusion.
> - We split the method into two subsections:
>     - Task-weighting meta-learning as trajectory optimisation: formulate the task-weighting meta-learning to a trajectory optimisation problem,
>     - Practical task-weighting method based on trajectory optimisation: propose a pratical approximation to implement the proposed method.
>
> We hope that these changes could increase the clarity and readability of our paper.

---

### Review · Reviewer_SZgw · 2023-06-19

**Summary Of Contributions:**

This paper proposes a new method for task weighting through trajectory optimization. They propose a method using iLQR which finds the optimal weightings over a trajectory of mini-batches. The paper proves convergence through this method to a stationary point of the policy. Finally, they provide several empirical examples using hand-crafted baselines. These baselines provide extreme versions of the exploration and exploitation their method should balance, and a uniform method which just uniformly weights the tasks. Finally, the paper compares favorably to the baseline performance found in (Lee et al, 2020).

**Audience:**

Yes

**Claims And Evidence:**

Yes

**Requested Changes:**

The weaknesses are ordered by importance. 1 and 2 are most critical to address while 3 is much more related to the clarity and writing of the paper. So the authors should focus on 1 and 2, using 3 to improve the paper in how they see fit.

**Strengths And Weaknesses:**

## Strengths:
- The motivation for the trajectory weighting is relatively clear from the perspective of prior work.
- The paper seems to do a good job going through the related work and clarifying how exactly their method/approach relates. Specifically, the introduction and section 4 have a lot of really nice explanations on prior work and their detriments.
-

## Weaknesses:
1. **BASELINES**: Given the rich set of potential baselines provided in the introduction and section 4, it is not clear why the only baselines provided were naive ones proposed by the authors. For instance, the potential for sub-optimal solutions is not a valid reason to not include the baseline in a scientific setting.
    - (1.1) I think $\alpha$-MAML and MWL-MAML (TR-MAML likely would be
  intractable) are great options for incorporating another baseline from the literature. While they are approaching the problem differently than here, this can be contextualized in the presented results more fully and discuss their trade-offs. This would also probably help understand what the computational trade-offs of your method over these would be. MWL does seem to only be tested w/ RL, but their approach should be applicable to the classification setting, no? Although, I'm not sure how the validation trajectories are a hurdle, could the authors go into more detail?
2. **The Study**: There are a few questions about the empirical study and some suggestions given the format of TMLR.
    - (a) I'm not as familiar with meta learning, but I'm not sure exactly why the results are presented without error shading in figure 1 and 2. This calls into question how the confidence intervals were calculated in table 2. Was this from multiple runs? If so how many? If not, is this standard in this literature?
    - (b) How does the TOW method compare computationally to the methods in Table 3? Is it still a large trade-off like the other baselines, or more reasonable?
    - (c) I think the studies in Appendix H and I have a place in the main paper to further understand the proposed algorithm. This also relates to the hypothesis discussed in section 5. These provide more evidence for your approach and suggest further study, imo.
    - (d) I think Appendix G should be included in the main paper, but that is more a style choice so the authors can decide.
3. **Writing**: I think the paper is very densly written and not very friendly to an audience who is not familiar with trajectory optimisation. There are also several parts of the text which could do with a re-organization to make things much clearer to the reader.
    - (a) I think section 3.1 should more directly relate back to algorithm 1 in a precise manner. Currently, the algorithm is a bit obfuscated on the precise details because of the writing in 3.1. For example, It is unclear if this approach uses Hessian diagonals or the Gauss-Newton diagonals. I believe the approach is using the Gauss-newton diagonals, but this is really unclear on first read.
    - (b) This also relates to section 4. While there is a lot of great information, I think the organization could have been done in such a way that it is not overly dense.

---

> ### Author Response · Authors · 2023-06-25
> **BASELINES: Why are alpha-MAML and MWL-MAML not included in the evaluation?**
>
> We do not use $\alpha$-MAML and MWL-MAML because their settings are different from the conventional meta-learning we are considering.
>
> In particular, $\alpha$-MAML follows the transductive setting where they assume the availability of testing tasks in the training phase. Such availability of testing tasks allows them to calculate the *distance* between weighted training tasks to those testing tasks to find an optimal weighting vector to train a meta-learning model. In contrast, we follow the inductive setting where there is no information regarding testing tasks available at training time. Our setting is quite common in the literature of meta-learning (Finn et al., 2017;
> Snell et al., 2017; Yoon et al., 2018). That is why we cannot use $\alpha$-MAML since $\alpha$-MAML needs extra information, potentially resulting in an unfair comparison.
>
> Regarding MWL-MAML, it is slightly similar to TR-MAML, except  that the weighting vector is learning via meta-learning. Thus, MWL-MAML also suffers difficulties similar to TR-MAML where the method requires to store a weight value for each task (see line 11 of Algorithm 2 in  (Xu et al., 2021). In our setting, such a requirement leads to an intractable solution because TR-MAML requires to store the weight value for each task.

---

> ### Author Response · Authors · 2023-06-25
> **How are the confidence intervals calculated in meta-learning? Why are the results in Figures 1 and 2 not presented with uncertainty?**
>
> In meta-learning, we benchmark the performance of a meta-learning method on a number of validation tasks. When evaluating on a task, we obtain the accuracy for that task, and the reported result is the average of those accuracies. Also, the 95\%  confidence interval is calculated based on those accuracies as in some previous meta-learning studies  (Vinyals et al., 2016; Finn et al., 2017; Ravi & Larochelle, 2017; Snell et al., 2017; Ravi & Beatson, 2019; Nguyen et al., 2020). However, there is another trend to calculate the performance by repeating the process multiple times  (Lee et al., 2020). In that case, the reported result is the average of averages and the confidence interval is simply measured from those multiple runs, not from the evaluation tasks.
>
> In this paper, we follow a traditional meta-learning evaluation protocol, where we report the confidence intervals based on the accuracy of each evaluation task. For Figures 1 and 2, we did not include such confidence intervals to ease the visualisation. Nevertheless, we provide a new visualisation with 95\% confidence intervals in Figure 8 in Appendix J.

---

> ### Author Response · Authors · 2023-06-25
> **Explain the computation of TOW presented in Table 3**
>
> The setting for the experiments in Table 3 is still very similar to the one in Table 1, except that the number of ways and shots are varied. In terms of computation, we also observe similar overheads as the ones presented in Table 2.

---

> ### Author Response · Authors · 2023-06-25
> **Writing and organisation structure of the paper**
>
> We thank the review to suggest several improvements in terms of writing and organising. We have made some changes as mentioned in the summary at the top of the discussion. We give some details of the changes here to follow more easily:
> - We split the section 3.1 into two subsections, where the first subsection presents the formulation of task-weighting in meta-learning to a trajectory optimisation, and the second subsection is to propose pratical approximation for implementation. The reviewer is right that we use Gauss-Newton diagonals to approximate the Hessian matrices.
> - For section 4, we move the auxilary lemmas to the Appendices to increase the readability of the presented analysis result. The remainings are: the boundedness of $\\mathbf{u}$, the assumptions made for the loss and its gradients and the main convergence result.

---

### Author Response · Authors · 2023-06-25
**Summary of the new revision submitted on June 25th 2023**

We thank the three reviewers for their constructive feedback. We introduce the following major changes to address the main points identified by the reviewers:
- We split the Section Method into two subsections (Section 3.1 Task-weighting as a trajectory optimisation, and Section 3.2 Practical task-weighting method based on trajectory optimisation)  for clarity.
- We move the auxiliary lemmas in section 4 (convergence analysis) to the Appendices to increase the readability of the paper.
- We move the Section Weight Visualisation from the Appendix to the main paper, making it a subsection of the Section Experiments.
- We add a new Section Ablation Studies, after the Section Experiments, to provide extensive analysis of the effect of some hyper-parameters used. In particular, we analyse the influence of the number of iLQR iterations, the length of the trajectory, the prior of the weighting vector $\mathbf{u}$ as well as training the three baselines more to have a fairer comparison.
- We add a brief description about the proposed method near the end of the Introduction section to increase the clarity of our paper.
- We clarify further some prior studies in the Related Work (highlighted in Magenta) and move the Related Work section to right before the Discussion and Conclusion section.
- We provide an additional visualisation of *shading plot* requested by reviewer SZgw in Appendix J.

The changes above are all highlighted or annotated to facilitate the next round of reviews.

---

### Author Response · Authors · 2023-06-30
**Discussion**

Dear the Action Editor and Reviewers,

We have provided a rebuttal to clarify and address concerns raised by the three reviewers. We are happy to discuss if there are any questions from the Reviewers and the Action Editor.

---

### Decision · Action_Editors · 2023-08-19

**Recommendation:** Accept as is

**Comment:**

This paper proposes an approach to perform task-weighting in the outer loop of the meta-learning process based on the trajectory optimization process.
The paper is clearly motivated and well-placed with respect to the state of the art. The problem the authors face is interesting, and the proposed approach is novel. The authors provide both theoretical analysis about the convergence of the proposed method and show empirically its effectiveness over simple baselines. The paper also highlights some limitations of the proposed approach.
During the discussion, the authors updated their paper to address some of the weaknesses pointed out by the reviewers. In particular, they have improved the clarity of the presentation, and the empirical analysis by adding some baselines and ablation studies.
The reviewers reached a consensus on accepting this paper.

**Audience:**

The paper is of interest to many researchers reading TMLR.

**Claims And Evidence:**

The claims made in the paper are supported by both theoretical and empirical evidence.